# Learning Interpretable Models Using an Oracle

## Abstract

We look at a specific aspect of model interpretability: models often need to be constrained in size for them to be considered interpretable, e.g., a decision tree of depth 5 is easier to interpret than one of depth 50. But smaller models also tend to have high bias. This suggests a trade-off between interpretability and accuracy. Our work addresses this by: (a) showing that learning a training distribution can often increase accuracy of small models, and therefore may be used as a strategy to compensate for small sizes, and (b) providing a *model-agnostic* algorithm to learn such training distributions. We also present a surprising artifact: the learned training distribution may be different from the test distribution.

We pose the distribution learning problem as one of optimizing parameters for an *Infinite Beta Mixture Model* based on a *Dirichlet Process*, so that the held-out accuracy of a model trained on a sample from this distribution is maximized. To make computation tractable, we project the training data onto one dimension: prediction uncertainty scores as provided by a highly accurate *oracle* model. A *Bayesian Optimizer* is used for learning the parameters.

Empirical results using multiple real world datasets, various oracles and interpretable models with different notions of model sizes, are presented. We observe significant relative improvements in the *F1-score* in most cases, occasionally seeing improvements greater than 100% over baselines.

Additionally we show that the proposed algorithm provides the following benefits: (a) its a framework which allows for flexibility in implementation, (b) it can be used across feature spaces, e.g., we show that the the text classification accuracy of a Decision Tree using character n-grams improves when using a Gated Recurrent Unit as an oracle, which uses a sequence of characters as its input, (c) it can be used to train models that have a non-differentiable training loss, e.g., Decision Trees, and (d) reasonable defaults exist for most parameters of the algorithm, which makes it convenient to use.

## 1  Introduction

In recent years, Machine Learning (ML) models have become increasingly pervasive in various real world systems. In many of these applications, such as movie and product recommendations, it is sufficient that the ML model is accurate. However, there is a growing emphasis on models to be *understandable* as well, especially in domains where the cost of being wrong is prohibitively high, e.g., medicine and healthcare (Caruana et al., 2015; Ustun & Rudin, 2016; J. Wang et al., 2022), defence applications (Gunning, 2016), law enforcement (Angwin et al., 2016; Larson et al., 2016) and banking (Castellanos & Nash, 2018).

This requirement is met by either (a) constructing models that are inherently interpretable, such as decision trees (Breiman et al., 1984; Quinlan, 1993; 2004; Hu et al., 2019) or decision sets (Lakkaraju et al., 2016), or (b) by allowing for post-hoc interpretability via model explanations, which may be simple models themselves, e.g., LIME (Ribeiro et al., 2016) or SHAP (Ribeiro et al., 2018). In either case, it would seem that model size plays a role in interpretability. Intuitively, a linear model with 10 terms, against one with 100 terms, or a decision tree (DT) of depth $= 5$ as opposed to one of depth $= 50$, is easier to parse by humans. This idea is supported by the following research:

- User studies indicate that small model size is one of a few important factors that makes a model interpretable: Lage et al. (2019) show in the context of decision sets that small model sizes, in conjunction with other properties, aid interpretability; Poursabzi-Sangdeh et al. (2021) find that smaller model sizes aid in certain tasks (for humans) where performance depends on understanding the workings of a model; Feldman (2000) notes that longer Boolean formulae are harder to learn by humans.

  While model size is important, Kulesza et al. (2013) caution against focusing on size in isolation, arguing smaller model sizes can be detrimental to understanding if they are too simplistic. Freitas (2014) highlights this aspect as well. Hence, size reductions that come at the cost of arbitrary reductions in accuracy are not helpful.

- This role of model size is variously acknowledged in the design and analysis of interpretable models: Herman (2017) refers to this as low *explanation complexity*, this is seen as important for *simulability* - ease of simulating the reasoning process of a model by a human (Lipton, 2018; Murdoch et al., 2019) - and is often listed as a desirable property in interpretable model representations. Some examples of the latter are small trees for cluster explanations (Moshkovitz et al., 2020; Laber et al., 2021), conciseness of decision sets in terms of number of rules and predicates per rule (Lakkaraju et al., 2016) and number of terms with non-zero coefficients in linear models (Ribeiro et al., 2016; Tibshirani, 1996).

Different algorithms constrain model size in a manner specific to their formulation, e.g., sparsity based loss for linear models and early stopping in decision trees. In this work we propose a *model-agnostic*[1] *technique to address this ubiquitous requirement of constraining model sizes while minimally trading off accuracy*. This provides the following practical value: instead of identifying an interpretable model family that is both (a) suited for a task in terms of representation, and (b) is sufficiently accurate for an acceptable size range, a practitioner now needs to just pick a suitable model family; the desired accuracy for their acceptable size range may be obtained using our technique.

Clearly the key challenge here is that to build small models one often needs to sacrifice accuracy, since size typically is inversely proportional to bias. We show that the accuracy of small models may be improved by *learning the training distribution* - thus mitigating this trade-off. This property forms the basis of our technique, and is shown to hold for multiple model families.

The parameters of the training distribution are learned using a form of *adaptive sampling*; they are iteratively modified to maximize held-out accuracy. In general, the number of parameters depends on the dimensionality of the data, which renders a naive execution of the strategy computationally expensive. We avoid this cost by representing the data using a *one-dimensional projection*: each training instance is represented by the uncertainty in its label prediction by an *oracle* model, a powerful probabilistic classifier[2]. The univariate space makes it feasible to define a rich search space of distributions using a *Dirichlet Process* with *Beta* mixture components. At a given iteration, the current distribution is used to sample training instances (with replacement) to create a new training dataset, which is then used to train a model of a given size. No synthetic instances are generated. Optimization over the space of distribution parameters is carried out by a *Bayeisan Optimizer*.

**Contributions**. These are the key contributions of this work:

1. We propose an algorithm[3] to find a training distribution that is optimal in terms of achieving high test accuracy, for a provided model family and model size. This algorithm is extensible in certain ways and therefore, may be seen as a *framework*.

---

[1] We adopt the common usage of the term (Ribeiro et al., 2016; Lundberg & Lee, 2017; Chen et al., 2018) to imply our technique is agnostic to model *families*.

[2] We'll often distinguish these two models using the terms (a) *interpretable model* - the size-constrained model that we are interested in, and (b) *oracle* - the model we use to obtain *uncertainty* scores. The oracle is not size-constrained and can belong to any model family; the only requirements are it should provide probabilistic scores for its predictions and it should be highly accurate.

[3] **NOTE**: We have made this available as Python package today, but we don't provide a link to preserve anonymity.

2. Based on extensive and rigorous experiments we report the counter-intuitive observation that *in general, the optimal training distribution is not the same as the test distribution, especially at small model sizes*[4].

Additionally, the following properties makes the proposed algorithm novel and practically valuable:

1. It is flexible in various ways: (a) it is **model-agnostic** in that the interpretable model and the oracle may belong to arbitrary model families, e.g., these can be *Linear Probability Model* and a *Random Forest* respectively, or even a decision tree and a *Gated Recurrent Network (GRU)*, (b) it **can be used with non-differentiable training loss functions**, and (c) it **admits a flexible notion of model size**, e.g., scalars such as depth of a decision tree or number of terms with non-zero coefficients in a linear model, and vectors such as both the number of trees and maximum depth per tree in a GBM model.

2. Although the algorithm has multiple hyperparameters that may be specified by a user, in practice only **one hyperparameter** needs to be set - reasonably good defaults exist for the rest.

3. The distribution is learned by solving an optimization problem that requires a **fixed number of seven optimization variables** irrespective of the dimensionality of the data.

The remainder of the paper covers the algorithm and various empirical results in detail, and is structured as follows: in Section 2, we illustrate results on a toy problem, introduce our notation, and discuss prior work. Section 3 discusses the algorithm in detail while Section 4 presents extensive experimental validation using real-world datasets. In Section 5 we discuss the results and their implications. Section 6 discusses directions for future work and Section 7 concludes the paper.

## 2 Overview

In this section, we first illustrate our technique on a toy problem to provide intuition. Then we establish terminology and notation, and provide a mathematical statement of the contributions. A discussion of related work concludes this section.

### 2.1 Toy Problem

Figure 1 illustrates the technique on a two-dimensional two-label dataset. The dataset is shown in Figure 1(a). Figure 1(b) visualizes the generalization learned by a *Gradient Boosted Model (GBM)* using this dataset. This serves as our oracle with an $F1$ score of $= 0.84$. Figure 1(c) shows what a *CART* (Breiman et al., 1984) decision tree of $depth = 5$ learns; here $F1 = 0.63$. Finally, Figure 1(d) shows what a CART decision tree of $depth = 5$ learns, when we supply the GBM as an oracle to our technique. There is a significant improvement with $F1 = 0.77$. Visually, we observe that the boundaries in Figure 1(d) approximate the ones learned by the oracle in Figure 1(b). We emphasize that both Figure 1(c) and (d) use the same training algorithm, data and constraint on model size - the change in accuracy is caused solely by the difference in their training distributions.

### 2.2 Terminology and Notation

We first define the notion of *model size* since it is critical for subsequent discussions. Model size is a model parameter with the following properties:

1. $model\_size \propto bias^{-1}$

2. The interpretability of a model decreases with increasing model size.

---

[4]This "small model effect" reaffirms the observations made in Ghose & Ravindran (2020).

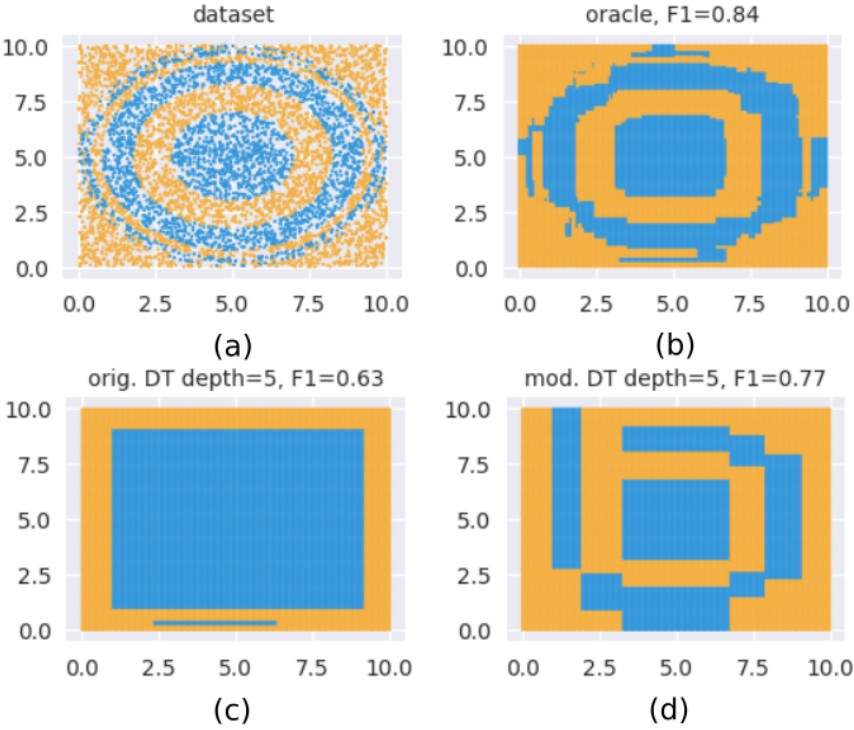

Figure 1: A demo of our technique using a GBM as an oracle. See text for explanation.

Only the first criteria above is required for using our technique. The second criteria specifies the practical context in which this technique is useful[5].

It must be noted that the notion of model size is subjective. Consider a GBM with DTs as base classifiers: here, the depth of the individual trees, or the number of trees, or both collectively may be seen as representing size. Even for a given notion of size, the value up to which a model is considered interpretable may be a matter of opinion. For example, some might consider a DT with $depth = 15$ to be interpretable, while some might decide $depth = 10$ to be the limit for interpretability. However, as long as the notion of size satisfies the criteria above, the discussion in this paper applies.

We now introduce the notations[6] used:

1. We denote a dataset, $D$, by a set of instance-label pairs, i.e.,
   $D = \{(x_1, y_1), (x_2, y_2), ..., (x_N, y_N)\}$, where $x_i$ is the feature vector representing an instance and $y_i$ is its corresponding label.

   We occasionally use *multisets* to represent data, when we want to draw attention to repeated instance-label pairs. Such usage is explicitly called out.

2. While we have referred earlier to the joint distribution of instances and labels, e.g., $p(X, Y)$ in Equation 1, this is understood to represent the dataset that we are actually given, in the form of a finite number of instance-label pairs.

3. We use the term *original*, as in *original distribution* or *original data* to denote the data that we are given. This is in contrast with samples we generate. The distribution of test datasets or held-out datasets is the original distribution for *all models discussed in this paper*.

---

[5]Another use-case for small models is for use on hardware-constrained low-powered devices, i.e., "tinyML" (Sanchez-Iborra & Skarmeta, 2020; Gupta et al., 2017); however, this use-case is beyond the scope of analysis here.

[6]Some of our notation is borrowed and extended from Ghose & Ravindran (2020) - their work solves a similar problem, and is discussed in Section 2.4, "Related Work".

4. Let $accuracy(M, p)$ be the classification accuracy of model $M$ on data represented by the joint distribution $p(X, Y)$ of instances $X$ and labels $Y$. The term "accuracy" is used in a generic sense to represent prediction accuracy; depending on the application, this might be *F1-score*, *AUC*, *lift*, etc.

   We'll often overload this function to accept a dataset instead of a distribution, e.g., $accuracy(M, D)$ denotes that the accuracy on dataset $D$.

5. Let $train_{\mathcal{F},f}(p, \eta)$ produce a model of size $\eta$ (for some pre-decided notion of size) from the model family $\mathcal{F}$ using a specific training algorithm $f$, e.g., $\mathcal{F}$ might represent DTs and $f$ might be the CART algorithm (Breiman et al., 1984), and the model size constraint might be *depth* = 5. $p$ represents the joint distribution $p(X, Y)$ of training instances $X$ and labels $Y$. $\eta = *$ denotes there are no limits placed on the model size.

   Here too, we'll overload the function to accept a dataset instead of a distribution, e.g., $train_{\mathcal{F},f}(D, \eta)$ implies dataset $D$ is used for training.

6. The terms *pdf* and *pmf* denote *probability density function* and *probability mass function* respectively. The term "probability distribution" may refer to either, and is made clear by the context.

## 2.3 Formal Statement

We now use the above terminology to rigorously state our claims. Let $\mathcal{I}$ and $\mathcal{O}$ represent specific interpretable and oracle model families respectively, and let $g$ and $h$ denote their corresponding training algorithms. Then, we claim, and *empirically* demonstrate, that *the interpretable model trained on the sample generated by our learned distribution is at least as accurate as one learned on the original training data, and is up to as accurate as the oracle*:

$$accuracy(train_{\mathcal{I},g}(p, \eta), p) \lesssim accuracy(train_{\mathcal{I},g}(q, \eta), p) \lesssim accuracy(train_{\mathcal{O},h}(p, *), p) \tag{1}$$

$p$ and $q$ both denote joint distributions of $X$ and $Y$. $p(X, Y)$ is the distribution we are provided, and *all* our models use this as the *test* distribution. $q(X, Y)$ is the distribution we learn over the uncertainty scores provided by the oracle; here, the oracle model is produced by $train_{\mathcal{O},h}(p, *)$. The use of the "$\lesssim$" symbol emphasizes these relationships are validated empirically using samples from the corresponding distributions $p$ and $q$.

Note that, typically, the train and test distributions are identical for a model, as in the first and third terms in Equation 1. However, for the middle term, the train and test distributions, $q$ and $p$ respectively, differ.

We also observe that the accuracy improvements from our technique diminish with increase in model size. In other words, Equation 1 may be deconstructed into two size-regimes: the interpretable model trained on the new sample is more accurate than the baseline model only until a model size $\eta^*$. At sizes greater than $\eta^*$ the model performances are equal:

$$accuracy(train_{\mathcal{I},g}(p, \eta), p) < accuracy(train_{\mathcal{I},g}(q, \eta), p), \text{ when } \eta \leq \eta^* \tag{2}$$
$$accuracy(train_{\mathcal{I},g}(p, \eta), p) = accuracy(train_{\mathcal{I},g}(q, \eta), p), \text{ when } \eta > \eta^* \tag{3}$$

## 2.4 Related Work

We present a brief review of related work here - a detailed version appears in the Appendix (Section A.1).

While there is precedent to using different train and test distributions, such as when there is class imbalance in the data (Japkowicz & Stephen, 2002; Chawla et al., 2002; He et al., 2008; Santhiappan et al., 2018), the only work we are aware of that applies the principle to learn size-constrained models is Ghose & Ravindran (2020). Their work uses a a specialized oracle, called *density trees*, to guide the learning phase. This work may be seen as a non-trivial extension: an oracle from an arbitrary model family may be used for guidance

(discussed in Section 3.1), resulting in greater flexibility and accuracy (Section 4.2.3). This also bestows an additional benefit: the feature spaces of the these models may be different (Section 4.3.1).

Knowledge Distillation (KD), Explainable AI (XAI), Active Learning and Transfer learning also have a notion of a teacher/oracle model. Relative to KD (Gou et al., 2021) and XAI, e.g., *TREPAN* (Craven & Shavlik, 1995), *Symbolic Metamodeling* (Alaa & van der Schaar, 2019), LIME, SHAP, the primary difference is that we ignore alignment/fidelity between the predictions of the oracle and the interpretable model. In fact, the oracle's label predictions are discarded. Also, the oracle is invoked exactly once to extract an one-time input (uncertainty scores) for our algorithm, unlike multiple invocations for generated instances as in LIME, SHAP or TREPAN. Essentially, the oracle plays a much more limited role here than in KD or XAI.

We operate within the traditional supervised setting for both the oracle and the interpretrable model, and they are presented with the same data. This differentiates us from Active Learning, where not all labels are available and there is an acquisition cost for them, and Transfer Learning, where a model is adapted to a different distribution/task.

$\square$

Next, we look at our methodology.

## 3 Methodology

We describe our methodology in this section. We begin with the intuition, and then look at the algorithm and various implementation details.

### 3.1 Intuition

Our intuition builds upon certain observations presented in Ghose & Ravindran (2020). There, to find an optimal training distribution:

1. A specific form of decision trees, referred to as *density trees* is learned to capture neighborhood information in the input space.

2. A node sampling distribution, defined over both internal nodes and leaves, is iteratively learned in the following manner: nodes are sampled based on the current distribution, and then data is sampled from within them. A classifier trained on this data is evaluated on a held-out set. This accuracy is used to modify the node sampling distribution, so it leads to greater accuracy in the next iteration.

   Thus, while the structure of the density trees are fixed, the flexibility of sampling from different regions of the input space is instrumented using various learned distributions that are defined over these trees.

It is shown that the learned distribution uses nodes from different depths. Since different depths represent varying levels of information about the location of class boundaries - information increases from the root towards the leaves - this indicates that a sampling distribution using this information *necessarily needs to be learned* as opposed to using simple heuristics like only sampling near class boundaries. This finding informs the key intuition behind this work: the class boundary information encoded in tree nodes is replaced by the much more general notion of *prediction uncertainty*, and we now learn a distribution over uncertainty scores instead, as provided by an oracle. An empirical comparison with density tree based sampling (Section 4.2) shows that, in addition to being more general, this formulation also performs better.

Algorithm 1 presents our intuition at a high level.

The training, validation and test data splits, denoted as $D_{train}, D_{val}, D_{test}$ respectively, are provided as input. The desired size of the interpretable model $\eta$, the training algorithm $train_{\mathcal{I},g}()$ and the oracle $M_O$ are also specified. Finally, the optimization budget, in terms of number of iterations, is specified as $T$. The training distribution is parameterized by $\Phi$, and our goal is to find the best parameters, denoted as $\Phi^*$, in

---

**Algorithm 1:** Algorithm sketch

**Data:** Data splits $D_{train}, D_{val}, D_{test}$, model size $\eta$, $train_{\mathcal{I},g}()$, oracle $M_O$, iterations $T$

**Result:** Optimal distribution parameters $\Phi^*$, test set accuracy $s_{test}$ at $\Phi^*$, and interpretable model $M^*$ at $\Phi^*$

**1** **for** $t \leftarrow 1$ **to** $T$ **do**

**2**     $\Phi_t \leftarrow suggest(s_1, ... s_{t-1}, \Phi_1, ..., \Phi_{t-1})$ `// randomly initialize ` $s_1, \Phi_1$

**3**     $D_s \leftarrow$ sample training instances, where the sampling probability of $(x_i, y_i) \in D_{train}$ is proportional to $p(u_{M_O}(x_i); \Phi_t)$

**4**     $M_t \leftarrow train_{\mathcal{I},g}(D_s, \eta)$

**5**     $s_t \leftarrow accuracy(M_t, D_{val})$

**6** **end**

**7** $t^* \leftarrow \arg\max_t \{s_1, s_2, ..., s_{T-1}, s_T\}$

**8** $\Phi^* \leftarrow \Phi_{t^*}$

**9** $M^* \leftarrow M_{t^*}$

**10** $s_{test} \leftarrow accuracy(M^*, D_{test})$

**11** **return** $\Phi^*$, $s_{test}$, $M^*$

---

$T$ iterations. Recall that the distribution is defined over prediction uncertainty values, as provided by $M_O$. The uncertainty score for an instance $x$ is denoted by $u_{M_O}(x)$.

The optimizer is denoted as $suggest()$, which, at iteration $t$, inspects the history of parameters explored, $\Phi_1, ..., \Phi_{t-1}$, and corresponding validation accuracies, $s_1, ... s_{t-1}$, to propose the next most promising parameter $\Phi_t$. This is used to create a training dataset $D_s$ by sampling instances from the provided training split $D_{train}$. The probability of sampling training instance $(x_i, y_i)$ is proportional to $p(u_{M_O}(x_i); \Phi_t)$. Model $M_t$ of size $\eta$ is trained using the dataset $D_s$, and its accuracy $s_t$ is calculated using the validation split $D_{val}$. Once the optimization budget is exhausted, i.e., $T$ iterations are complete, the model with the highest validation accuracy $M^*$, the corresponding distribution parameters $\Phi^*$ and test score $s_{test}$, are returned.

The next few sections further elaborate the various steps in Algorithm 1.

### 3.2 Measuring Uncertainty

Our technique critically depends on the measurement of uncertainty. We denote the uncertainty of prediction by a model $M$ on an instance $x$ by $u_M(x)$, where $u_M(x) \in [0, 1]$. A good uncertainty metric for our application (a) should not exclusively consider the confidence of the predicted label (b) should result in a high value even if the model is uncertain between two labels in a multi-class problem.

The **margin uncertainty** (Scheffer et al., 2001) metric satisfies these criteria. This is computed as:

$$u_M(x) \leftarrow 1 - (p_{C_1} - p_{C_2}) \tag{4}$$

Here, $p_{C_1}$ and $p_{C_2}$ denote the probabilities of the most confident and next most confident classes, provided by model $M$ for instance $x$. Lower differences between the top two probabilities lead to higher scores for this metric. We calibrate (Platt, 1999) our oracles for reliable probability estimates.

See Section A.5 for a discussion on alternative uncertainty metrics.

### 3.3 Sampling based on Uncertainty

Since we want to learn a distribution over uncertainties, $p(u_M(x))$ needs to have a flexible representation. A desiderata for such a distribution is:

1. Since we want to avoid any assumptions, we want the distribution to be able to assume an arbitrary "shape", unlike, say using a normal distribution that is unimodal, and the mode is centered.

2. It should be defined over the bounded interval $[0, 1]$ since $u_M(x) \in [0, 1]$.

3. A *fixed set of parameters* is preferred over a conditional parameter space. An example of a distribution with a conditional parameter space is the popular *Gaussian Mixture Model (GMM)*, where the number of parameters is determined by the number of components.

   We list this requirement since the parameters of this distribution are to be learned via optimization, and there are many more optimizers that can handle fixed than conditional parameter spaces. This affords us the flexibility of exploring a much wider variety of optimizers. Further discussed in Section 3.5.

The *Infinite Beta Mixture Model (IBMM)* satisfies the above requirements. The IBMM is a *Dirichlet Process (DP) mixture model* with *Beta* components. It may be seen as a variation of the *Infinite Gaussian Mixture Model* (Rasmussen, 1999) and an example of prior use may be found in Ghose & Ravindran (2020).

A mixture model allows us to model an arbitrary distribution, satisfying our first requirement. Using *Beta* components enables support for a bounded interval - this satisfies our second requirement. As a point of note, a *Beta* mixture model can approximate any distribution in $[0, 1]$ arbitrarily well given a sufficient number of components (Diaconis & Ylvisaker, 1983). The DP is described by the *concentration parameter* $\alpha \in \mathbb{R}_{>0}$, which identifies the components that have at least one point assigned to them[7]. The shape parameters of all the *Beta* components are drawn from *shared* prior distributions; the latter are also assumed to be *Beta* distributions. Use of a DP, with shared priors, gives us a fixed parameter space; this satisfies our third requirement.

In our representation, this is how we sample $N_s$ points, from a dataset $D$, using an oracle $M_O$:

1. Determine partitioning over the $N_s$ points induced by the $DP$. We use *Blackwell-MacQueen* sampling (Blackwell & MacQueen, 1973) for this. Let's assume this step produces $k$ partitions $\{c_1, c_2, ..., c_k\}$ and quantities $n_i \in \mathbb{N}$ where $\sum_{i=1}^{k} n_i = N$. Here, $n_i$ denotes the number of points that belong to partition $c_i$.

2. We determine the $Beta(A_i, B_i)$ component for each $c_i$. We assume the priors for the *Beta* parameters are also represented by *Beta* distributions, i.e., $A_i \sim scale \times Beta(a, b)$ and $B_i \sim scale \times Beta(a', b')$. Since samples from the standard *Beta* are within $[0, 1]$, we use a parameter *scale* as a common multiplier to obtain a wide range of $A_i, B_i$.

   Thus we have exactly two prior *Beta* distributions associated with our IBMM. Here, $a, b, a', b'$ are positive reals.

3. Repeat for each $c_i$: for each instance-label pair $(x_j, y_j)$ in our training dataset, we calculate the oracle uncertainty score, $u_{M_O}(x_j)$. We then calculate $p_j = Beta(u_{M_O}(x_j)|A_i, B_i)$. We scale the probabilities across instances to sum to 1. These quantities are used as sampling probabilities for various $(x_j, y_j)$, and $n_i$ points are sampled with replacement based on them.

The parameters for the IBMM are collectively denoted by $\Psi = \{\alpha, a, b, a', b'\}$. The best values for $\Psi$ are learned via an optimization process detailed in Section 3.4.

The above procedure is summarized in Algorithm 2. Note that *temp* and $D'$ are multisets in the algorithm, since we sample with replacement. Accordingly, line 13 uses the **multiset sum**, $\uplus$: if $(x_i, y_i)$ occurs $m$ times in $D'$ and $n$ times within *temp*, then $D' \leftarrow D' \uplus temp$ has $m + n$ occurrences of $(x_i, y_i)$.

### 3.4 Learning Interpretable Models using an Oracle

We are now prepared to revisit the overall algorithm. We have already discussed the parameters $\Psi$ for the IBMM. We now introduce two additional parameters:

---

[7]In theory, the DP has an infinite number of components, with only a finite number of them actually representing instances in the data.

---

**Algorithm 2:** Sample based on uncertainties and $\Psi$

---

**Data:** Sample size $N_s$, oracle $M_O$, dataset $D = \{(x_1, y_1), (x_2, y_2), ..., (x_N, y_N)\}$, IBMM parameters
 $\quad\quad \Psi = \{\alpha, a, b, a', b'\}$
**Result:** Sample $D'$, where $|D'| = N_s$

**1** $D' = \{\}$ `// assumed to be a multiset`
**2** $\{(c_1, n_1), (c_2, n_2), ..., (c_k, n_k)\} \leftarrow$ partition $N_s$ using the $DP$ `// Here` $\sum_{i=1}^{k} n_i = N_s$`.`
**3** **for** $i \leftarrow 1$ **to** $k$ **do**
**4** $\quad$ $A_i \sim scale \times Beta(a, b)$
**5** $\quad$ $B_i \sim scale \times Beta(a', b')$
**6** $\quad$ **for** $j \leftarrow 1$ **to** $N$ **do**
**7** $\quad\quad$ $p_j \leftarrow Beta(u_{M_O}(x_j); A_i, B_i)$
**8** $\quad$ **end**
**9** $\quad$ **for** $j \leftarrow 1$ **to** $N$ **do**
**10** $\quad\quad$ $p'_j \leftarrow c \cdot p_j$, where $c = 1 / \sum_{j=1}^{N} p_j$ `// normalize the probabilities`
**11** $\quad$ **end**
**12** $\quad$ $temp \leftarrow$ sample with replacement $n_i$ instance-label pairs based on $p'_j$ `// assumed to be a multiset`
**13** $\quad$ $D' \leftarrow D' \uplus temp$ `//` $\uplus$ `is the multiset sum`
**14** **end**
**15** **return** $D'$

---

1. $p_o \in [0, 1]$, proportion of instance-label pairs from the original training data. This parameter serves two purposes: (1) it acts as a "shortcut" for the optimizer to sample from the original distribution, as opposed to determining the right $\Psi$ to do so (2) the relationship of $p_o$ and model size enables us to conveniently study the role of the original distribution at different sizes.

2. $N_s \in \mathbb{N}$, sample size. Since the sample size can have a significant effect on model performance, we allow the optimizer to determine its best value. $N_s$ is constrained to be at least as large as what is needed for statistically significant results.

The complete set of parameters is denoted by $\Phi = \{\Psi, N_s, p_o\}$, where the IBMM parameters are denoted by $\Psi = \{\alpha, a, b, a', b'\}$.

For a given $\Phi$, our technique creates a sample based on Algorithm 2 and the original training data (based on $p_o$), learns an interpretable model of size $\eta$ on this sample, and evaluates it on a validation set. Based on the validation score, an optimizer modifies the parameters $\Phi$, and repeats the process. Our stopping criteria is an iteration budget $T$. Algorithm 3 lists these steps.

Some details to note in Algorithm 3:

1. We will consider the initialization to happen at $t = 0$, while the iterations range from 1 to $T$. $\Phi_0$ is set to: $\alpha = 0.1, a = 1, b = 1, a' = 1, b' = 1, N_s = |D_{train}|, p_o = 1$. A model is constructed based on $\Phi_0$ and a score $s_0$ is recorded. $(\Phi_0, s_0)$ serve as the history for the iteration at $t = 1$. The values for $\alpha, a, b, a', b'$ carry no significance and are arbitrary, since setting $p_o \rightarrow 1$ forces sampling only from the original distribution[8]. Combined with $N_s = |D_{train}|$, this setting mimics the baseline, i.e., training the interpretable model without our algorithm, thus providing the optimizer with a good initial reference point in its search space[9].

2. The optimizer is represented by the function call $suggest()$ which takes as input all past parameter values and validation scores. $suggest()$ denotes a generic optimizer; not all optimizers require this extent of historical information.

---

[8]Note that this one setting is insufficient for the optimizer to discover the irrelevance of the values for $a, b, a', b', \alpha$ given $p_o = 1$.

[9]This is still not equivalent to the baseline because since models are trained in a resource constrained environment within the optimizer - as described in Section 4.1.1.

---

**Algorithm 3:** Learning interpretable model using oracle

---

**Data:** Dataset $D$, model size $\eta$, $train_{\mathcal{O},h}()$, $train_{\mathcal{I},g}()$, iterations $T$
**Result:** Optimal parameters $\Phi^*$, test set accuracy $s_{test}$ at $\Phi^*$, and interpretable model $M^*$ at $\Phi^*$

**1** Create splits $D_{train}, D_{val}, D_{test}$ from $D$, stratified wrt labels
**2** $M_O \leftarrow train_{\mathcal{O},h}(D_{train}, *)$
**3** **for** $t \leftarrow 1$ **to** $T$ **do**
**4**     $\Phi_t \leftarrow suggest(s_0, s_1, ...s_{t-1}, \Phi_0, \Phi_1, ..., \Phi_{t-1})$ `// `$s_0, \Phi_0$` initialized at `$t=0$`, see text`
     `// Note: `$\Phi_t = \{\Psi_t, N_{s,t}, p_{o,t}\}$` where `$\Psi_t = \{\alpha_t, a_t, b_t, a'_t, b'_t\}$`.`
**5**     $N_o \leftarrow p_{o,t} \times N_{s,t}$
**6**     $N_u \leftarrow N_{s\_t} - N_o$
**7**     $D_o \leftarrow$ uniformly sample, with replacement, $N_o$ points from $D_{train}$
**8**     $D_u \leftarrow$ sample $N_u$ points from $D_{train}$ using **Algorithm 2** with input $(N_u, M_O, D_{train}, \Psi_t)$.
**9**     $D_s \leftarrow D_o \uplus D_u$ `// `$D_o$`, `$D_u$` are assumed to be multisets`
**10**    $M_t \leftarrow train_{\mathcal{I},g}(D_s, \eta)$
**11**    $s_t \leftarrow accuracy(M_t, D_{val})$
**12** **end**
**13** $t^* \leftarrow \arg\max_t \{s_1, s_2, ..., s_{T-1}, s_T\}$
**14** $\Phi^* \leftarrow \Phi_{t^*}$
**15** $M^* \leftarrow M_{t^*}$
**16** $s_{test} \leftarrow accuracy(M^*, D_{test})$
**17** **return** $\Phi^*$, $s_{test}$, $M^*$

---

3. While the training algorithm for the oracle, $train_{\mathcal{O},h}()$ is taken as input, a pre-constructed oracle $M_O$ may also be used. This would eliminate the oracle training step in line 2.

4. $accuracy()$ on the validation data, $D_{val}$, serves as both the objective and fitness function.

5. Evaluation on the test set, $D_{test}$ is done only once, in line 16, with the model that produces the best validation score.

6. Since we sample with replacement, both temporary datasets $D_o$ and $D_u$, procured from uniformly sampling the original training data and sampling based on uncertainties respectively, are multisets. Accordingly, line 9 uses the multiset sum operator $\uplus$ to combine them.

7. $M_t$ is created (line 10) with limited or no hyperparameter search using simple random validation, i.e., a stratified (by labels) random sample of size $0.2N_{s,t}$ is used as the validation set. A restricted search is performed because often hyperparameters are correlated with model size, and setting them to particular values would fail to produce a model of the required size $\eta$. As an example, consider DTs: setting a high threshold for the number of instances in a node for it be split (hyperparameter *min_samples_split* in *scikit-learn's* (Pedregosa et al., 2011) implementation) would produce only short trees.

   We don't use cross-validation since at small values of $N_{s,t}$, the amount of training data, i.e., $(\frac{k-1}{k})N_{s,t}$ for $k$-folds, may become too small to obtain a good model. For example, for 3-folds, the training data size is $0.67N_{s,t}$. The data shortage problem can be addressed by increasing the number of folds, but that also increases the running time per iteration owing to the larger number of models that now need to be trained. As a practical compromise, we perform simple validation *thrice* and average the outcomes. This number is configurable, and may be decreased for models that are expensive to train.

8. Since the validation score $s_t$ (line 11) needs to be reliable, in our implementation we repeat lines 7-10 *thrice* and use the averaged validation score as $s_t$.

9. Class imbalance is accounted for in our implementation when training model $M_t$ in line 10. We either balance the data by sampling (this is the case with a *Linear Probability Model*), or an appropriate cost function is used to simulate balanced classes (this is the case with DTs and GBMs).

It is important to note here that $D_{val}$ *and* $D_{test}$ *are not modified by our algorithm in any way*, and therefore $s_t$ and $s_{test}$ measure the accuracy on the original distribution. The possibility of eliminating various repeated trials - in fitting $M_t$ and in calculating $s_t$ - is discussed in Section 6.

Algorithm 3 presents the core contribution of the paper. Importantly, we note that the following properties make it practical to use: (a) the optimization loop has a fixed set of seven variables, irrespective of the dimensionality of the data or the number of mixture components, (b) $train_{\mathcal{I},g}()$ is treated as a black-box function; not only does this make the algorithm model-agnostic, but also it allows it to work with cases of non-differentiable losses, e.g., DTs built using CART, and (c) the Dirichlet Process is purely used for representation, and we do not need to perform any form of inferencing (which can be expensive).

Clearly, the choice of the optimizer $suggest()$ is crucial - we discuss this next.

### 3.5 Choice of Optimizer

We list below the challenges faced by our optimizer:

1. **Black-box objective function**: Our objective function is $accuracy()$, which depends on the interpretable model produced by $train_{\mathcal{I},g}()$ in Algorithm 3. Since we want our technique to be model agnostic, nothing is assumed about the form of $train_{\mathcal{I},g}()$. This effectively makes our objective a black-box function.

2. **Noisy objective function**: The interpretable model is trained on a *sample* based on the current parameters $\Phi_t$. This implies two models constructed for the same $\Phi_t$ may not be identical. There might be other sources of noise intrinsic to the learning algorithm too, e.g., local search with random initialization used for training.

3. **Expensive objective function**: Every evaluation of the objective function requires an interpretable model to be trained, which is expensive. We want our optimizer to be conservative in its calls to the objective function.

We use *Bayesian Optimization (BO[10])* to implement $suggest()$. Such optimizers build their own model of the response surface as a function of the optimization variables, that is refined over multiple iterations. They optimize this *surrogate* objective. This strategy enables them to work with black-box objective functions, satisfying our first requirement. BOs explicitly model variability (due to noise or lack of information) in the response surface model, by using appropriate representations such as *Gaussian Processes (GP)* or *Kernel Density Estimators (KDE)*; this satisfies our second requirement. The evolving response surface (over iterations) allows BOs to balance *exploitation and exploration* to make well-informed proposals for the best next point to evaluate, making it *sample-efficient*. This satisfies our third requirement.

BO has enjoyed continued success for *hyperparameter optimization* (Feurer & Hutter, 2019; Turner et al., 2021), a domain with similar optimization challenges as ours. This makes it an attractive choice over other promising candidates such as Particle Swarm Optimization (Kennedy & Eberhart, 1995) and evolutionary algorithms like *CMA-ES* (Hansen & Ostermeier, 2001; Hansen & Kern, 2004). A known limitation of BO is its inability to handle high-dimensional spaces: Frazier (2018) and Moriconi et al. (2020) warn that BO works well up to 20 dimensions. However, this does not affect us since we have only seven optimization variables. See the reference Brochu et al. (2010) for further details on BO. We use BO with box constraints - these are discussed in Section 4.1.5.

Among BO techniques, of which there are many today, e.g., Hutter et al. (2011); Bergstra et al. (2011); Malkomes & Garnett (2018); Dai et al. (2019); Tiao et al. (2021), we use the *Tree Structured Parzen Estimator (TPE)* algorithm (Bergstra et al., 2011) since it scales linearly with the number of evaluations[11] and has a popular and mature library: *Hyperopt* (Bergstra et al., 2013).

---

[10]We use the abbreviation BO for both *Bayesian Optimization* and *Bayesian Optimizers*; it should be clear from the context which is intended.

[11]The runtime complexity of a naive BO algorithm is *cubic* in the number of evaluations (Shahriari et al., 2016).

However, let us emphasize that any optimizer that satisfies the criteria listed at the beginning of this section may be used to implement *suggest()*, and in fact, we show examples of this in Section A.8, with the optimizers *pySOT* (Eriksson et al., 2019) and *LIPO* (Malherbe & Vayatis, 2017). We note again that the design choice of a fixed parameter space is what enables this extensibility[12] - the universal support for such spaces allows us to explore the use of various optimizers. This also allows us to create faster and better implementations of Algorithm 3 over time as newer optimizers become available.

### 3.6 Smoothing the Optimization Landscape

We conclude our discussion on algorithm design with a practical consideration: can we facilitate finding the maxima $\Phi^*$ in Algorithm 3?

Since BOs model the response surface of the actual objective function using a finite number of evaluations ($s_t$ in Algorithm 3), a certain degree of *smoothness* is assumed (Shahriari et al., 2016; Brochu et al., 2010). Here, the optimization variables $\Phi$ influence the objective value $s$ via this indirect chain: $\Phi_t \rightarrow D_s \rightarrow M_t \rightarrow s_t$ (symbols as in Algorithm 3), and for BO to work well, it is required that small changes in $\Phi_t$ result in small changes in $s_t$.

However, we have noticed that an oracle might produce uncertainty score distributions that are "spiky" or "jagged" - as an example, see the curve labelled "original" in Figure 2(a); which leads us to hypothesize that this principle is violated in general. A spiky distribution implies that small shifts $\Phi_t + \Delta\Phi_t$ may lead to sampling of instances with very different uncertainties; and since such instances may occur in regions far from those indicated by $\Phi_t$, they produce models with different class prediction behavior. This indirectly causes a disproportionate shift in $s_t$. While, in theory, a good BO algorithm should adapt to such problem characteristics, in practice they make the optimization problem harder, especially when the optimization budget is small.

To address this, we "flatten" the distribution[13] within $[0, 1]$. Our transformation is simple: we divide the interval $[0, 1]$ into $B$ bins, and map approximately $|D_{train}|/B$ uncertainty scores to each bin, while maintaining order between the original and mapped scores. Within a bin, the mapped scores are linearly spread across its range. This distributes the mapped scores approximately uniformly in the range $[0, 1]$. The algorithm is detailed in Section A.6.

Figure 2 visualizes the process of flattening. The original and modified uncertainty distributions for the datasets `Sensorless` and `covtype.binary` are shown in Figure 2(a) and 2(b) respectively.

While `Sensorless` appears to have a non-smooth distribution, and flattening here might help, this seems redundant for `covtype.binary`. *However, since this step is computationally cheap, we perform this for all our experiments, saving us the effort of assessing its need.* The benefits from flattening are evaluated in Section 5.

Our transformation is invertible, which is useful in analyzing the observations from our experiments. Note however, it is not differentiable because of the discontinuities at the bin-boundaries; we also don't require this property.

The transformation affects line 7 in Algorithm 2. Instead of sampling based on the actual oracle uncertainty scores:

$$p_j \leftarrow Beta(u_{M_O}(x_j); A_i, B_i) \tag{5}$$

we sample based on the transformed uncertainty scores, $u'_{M_O}(x_j)$:

---

[12]TPE supports conditional parameter spaces, which would have allowed us to use a finite mixture model such as GMMs by setting the number of mixture components as the top level optimization variable - but we deliberately ignore this feature in the interest of extensibility.

[13]Distribution transformations have a long history in statistics, e.g., *power transforms* like the *Box-Cox* (Box & Cox, 1964) and *Yeo-Johnson* (Yeo & Johnson, 2000) transforms. Within ML, *Batch Normalization* (Ioffe & Szegedy, 2015) is a popular example of a distribution transformation applied to a loss landscape (Santurkar et al., 2018).

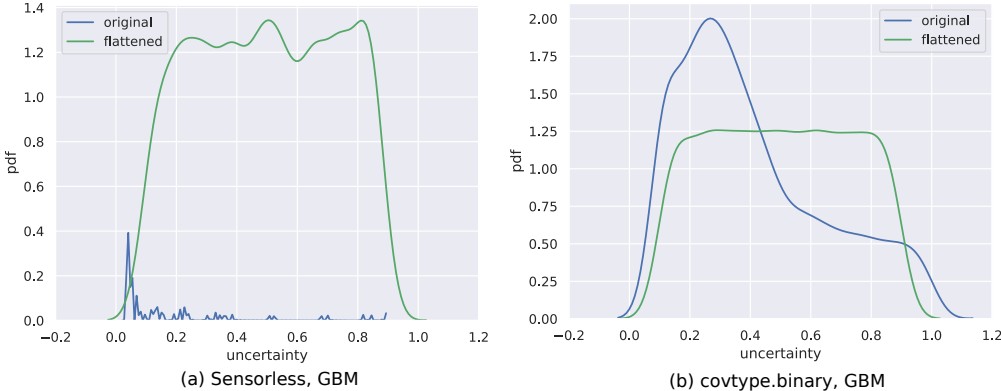

Figure 2: Example of curve-flattening, for datasets (a) `Sensorless` and (b) `covtype.binary`. The uncertainty scores shown are obtained using the *GBM* oracle.

$$p_j \leftarrow Beta(u'_{M_O}(x_j); A_i, B_i) \tag{6}$$

$\square$

This concludes our discussion of algorithmic details. In summary, we require seven parameters $\Phi = \{\Psi, N_s, p_o\}$, where $\Psi = \{\alpha, a, b, a', b'\}$. Hyperparameters are discussed in Section 4.1.5. Our experimental validation of the technique is discussed next.

## 4 Experiments

We now look at extensive evaluation of our technique. Our experiments maybe categorized into the following types:

1. **Validation**, Section 4.1: this set of experiments exhibit statistically significant improvements across multiple datasets, using different models and oracles (Section 4.1.6). Various properties of the learned distributions are analyzed (Section 4.1.7). The relationship between model capacity and the efficacy of our technique is also discussed (Section 4.1.9).

2. **Comparisons**, Section 4.2: here we compare the improvements produced by our technique with (a) a supervised version of uncertainty sampling and (b) using density trees.

3. **Additional applications**, Section 4.3: fundamentally, our technique learns a sampling distribution that leads to effective training. This can be used as a tool for the following interesting applications - (a) different feature representations may be used across the interpretable model and the oracle, e.g., a DT as the interpretable model with n-grams as input, and a Gated Recurrent Unit the oracle, that operates on a sequence of tokens, (b) a minimal sample for effective learning maybe identified using our technique and (c) a multivariate notion of model size may be used.

The section on validation experiments is the most comprehensive, empirically demonstrating various properties of our technique. A discussion of **running times** appears in Section A.8 of the Appendix.

### 4.1 Validation

To empirically validate our technique we consider different real-world datasets, on which we train *Linear Probability Models (LPM)* and DTs, using *Gradient Boosted Models (GBM)* and *Random Forests (RF)* as

oracles. The experimental setup is described in the following sections in terms of the metrics used, data, models, baselines and oracles, and the optimization search space explored.

### 4.1.1 Metrics

We measure two quantities - improvements in model accuracy and their statistical significance:

1. To measure *accuracy*() as in Equation 1 or Algorithm 3, our metric of choice is the $F1$ (macro) score, evaluated on $D_{test}$. We use this since it accounts for class imbalance, e.g., it doesn't allow good results for a majority class to eclipse poor results for a minority class.

   To measure the *improvements* obtained from our technique, we record the percentage *relative improvement* in the **test** $F1$ score compared to a *baseline* model (described later in this section):

   $$\delta F1 = \frac{100 \times (\overline{F1}_{new} - \overline{F1}_{baseline})}{\overline{F1}_{baseline}} \tag{7}$$

   Here $\overline{F1}_{new}$ and $\overline{F1}_{baseline}$ represent averages over **five runs** of Algorithm 3. In other words, if the runs are indexed by $i$, and $F1_{baseline,i}$ and $F1_{new,i}$ are the test F1 scores at run $i$ produced by the baseline and our techniques respectively, then $\overline{F1}_{new} = \sum_{i=1}^{5} F1_{new,i}/5$ and $\overline{F1}_{baseline} = \sum_{i=1}^{5} F1_{baseline,i}/5$.

   We take an average of the scores first since $F1_{baseline}$ can be a small value, especially at smaller model sizes, and being in the denominator, slight changes to it across runs can produce outsize differences in the per-run $\delta F1$ scores.

2. To measure statistical significance of our results we use the *Wilcoxon signed-rank test*, where the paired set of samples are $F1_{baseline}$ and $F1_{new}$ scores (from Equation 7) for a dataset. The *p-value* is reported. This test is separately performed for different model sizes.

There are *two* kinds of baseline models we use, resulting in slightly different values for $\delta F1$; this allows us to highlight different aspects of our technique. These are described below. It is relevant to note that the data split ratios in our experiments are $|D_{train}| : |D_{val}| : |D_{test}| :: 60 : 20 : 20$.

1. **Improvements in accuracy** (Section 4.1.6): Here, the baseline is a model trained with stratified 3-fold cross-validation (which we'll denote by $M_{base}$). The lower bound of $\delta F1$ is *set* to 0, since in a setting with a larger optimization budget (or a score-based stopping criteria, e.g., Nguyen et al. (2017); Makarova et al. (2022)), the learned distribution $\Psi$ would perfectly approximate the original distribution due to the flexibility of the *Beta* mixture model (Diaconis & Ylvisaker, 1983), leading naturally to[14] $\delta F1 = 0$. This setting also simulates the practical outcome that a user cannot do worse than $M_{base}$.

   Here, $\delta F1 \in [0, \infty)$.

   Of course, this leaves open the question of well does the BO perform relative to random chance. The differences in environments available to the baseline model vs the within-iteration model limit the resolution of such an analysis based on a direct comparison. We describe these differences next, and propose an alternative measurement strategy.

2. **Optimizer performance** (Section 4.1.8): Recall from the description of Algorithm 3 in Section 3.4 that the per-iteration model $M_t$ is allowed to train on only 80% of $N_{s,t}$ instances during training. Being an optimization variable, $N_{s,t}$ can be quite small at an iteration (the lower bound is a user setting, discussed in Section 4.1.5) and is drawn entirely from $D_{train}$. Compare this to $M_{base}$, where

---

[14]It is possible to have some irreconcilable difference between $F1_{new}$ and $F1_{baseline}$ due to the difference in statistical properties of the validation sets, based on which the respective models are selected, and the common test set, based on which the $\delta F1$ is reported. However, we expect the resultant score differences, just arising from this factor, to be negligible, and to entirely disappear with large enough samples or enough runs.

the training algorithm is provided $D_{train}$ and $D_{val}$ pooled together for constructing the best model with hyperparameters determined via cross-validation.

In terms of data diversity, $M_{base}$ uses 80% (i.e., $|D_{train}| + |D_{val}|$) of the total dataset, while $M_t$ can use a maximum of 60% (i.e., $|D_{train}|$). Also consider that the hyperparameter search space of $M_t$ is possibly restricted (described in Section 3.4). Taken together these facts indicate that $M_{base}$ has inherent advantages over $M_t$, and therefore, is not a fair reference for a fine-grained analysis.

However, the model constructed at initialization, which we'll denote[15] as $M_{init}$, can serve as a fair reference, since: (a) it has the same resource constraints (wrt hyperparameter space and data) as any $M_t$, and (b) its use of $p_o \to 1$ and $N_{s,0} = |D_{train}|$ (described in Section 3.4) mimics the distributional setup for $M_{base}$.

Section 4.1.8 presents $\delta F1$ computed with $M_{init}$ as the baseline. Additionally, we account for variability across runs in the following manner:

(a) Let runs be denoted by index $i$. We denote the model at $t = 0$ for various runs by $M_{init,i}$. Let their respective validation and test $F1$ scores be denoted by $F1_{init,i}^{val}$ and $F1_{init,i}^{test}$.

(b) Let the best model found in each run by Algorithm 3 be denoted by $M_i^*$. Let their respective validation and test $F1$ scores be denoted by $F1_{*,i}^{val}$ and $F1_{*,i}^{test}$.

(c) Then, we define the quantities:

$$\overline{F1}_{baseline} = \frac{\sum_{i=1}^{5} F1_{init,i}^{test}}{5} \quad \text{and} \quad \overline{F1}_* = \frac{\sum_{i=1}^{5} F1_{*,i}^{test}}{5} \tag{8}$$

(d) An independent *t-test* is conducted between the sets of **validation** scores $\{F1_{init,i}^{val}\}_{1 \le i \le 5}$ and $\{F1_{*,i}^{val}\}_{1 \le i \le 5}$, with the null hypothesis that the validation scores produced by $M_i^*$ are not significantly different from those produced by $M_{init,i}$. Based on the *p-value* of the test we set:

$$\overline{F1}_{new} = \overline{F1}_{baseline} \qquad \qquad \text{if } p \ge 0.1 \tag{9}$$

$$\overline{F1}_{new} = \overline{F1}_* \qquad \qquad \text{if } p < 0.1 \tag{10}$$

These values for $\overline{F1}_{baseline}$ and $\overline{F1}_{new}$ are used in Equation 7 to calculate $\delta F1$.

Here, $\delta F1 \in (-\infty, \infty)$. Negative values can occur when the validation scores of $M_i^*$ are picked in the *t-test*, and therefore $\overline{F1}_{new} = \overline{F1}_*$, but test scores are lower, i.e., $\overline{F1}_{new} < \overline{F1}_{baseline}$.

Thus we present metrics - $\delta F1$ and its statistical significance - corresponding to two baselines, $M_{base}$ and $M_{init}$, from the same set of experiments. For experiments not pertaining to validation, i.e., for comparisons (Section 4.2) and additional applications (Section 4.3), $M_{base}$ is used as the baseline because we believe that to be closer to the expected interpretation of $\delta F1$. For comparison to the density tree based approach, $M_{base}$ is required as a baseline, since this is what they use to report their results (Section 4.1 in Ghose & Ravindran (2020)).

### 4.1.2 Data

We use *13* real-world datasets to validate our technique. Table 1 lists relevant details. These are picked to vary in their dimensions, number of labels and label distribution, enabling a broad validation of our technique. Although we use the version of data available on the *LIBSVM* website (Chang & Lin, 2011), we mention their original source in Table 1. 10000 instances from each dataset are used. We use a $train : val : test$ split ratio of $60 : 20 : 20$ to create $D_{train}, D_{val}$ and $D_{test}$ in all our experiments (line 1, Algorithm 3). The data splits are stratified wrt class labels.

In terms of the label distribution, we are interested in knowing whether a dataset is balanced wrt labels. We quantify this with the "Label Entropy", which is computed for a dataset with $N$ instances and $C$ labels in

---

[15]This is technically $M_0$ since its created at $t = 0$, but we use the term $M_{init}$ to avoid confusion with $M_O$, the symbol for the oracle model.

the the following manner:

$$\text{Label Entropy} = \sum_{j \in \{1,2,...,C\}} -p_j \log_C p_j \tag{11}$$

$$\text{Here, } p_j = \frac{|\{x_i | y_i = j\}|}{N}$$

Label Entropy $\in [0, 1]$, where values close to 1 denote the dataset is nearly balanced, and values close to 0 represent relative imbalance.

### 4.1.3 Models

For interpretable models $\mathcal{I}$, we consider the following model families:

1. *Linear Probability Model (LPM)* (Mood, 2010): This is a linear classifier. We use the commonly accepted notion of model size here: the number of terms in the model, i.e., features from the original data, with non-zero coefficients. We use the *Least Angle Regression* (Efron et al., 2004) algorithm, that grows the model one term at a time, to enforce the size constraint. We use our own implementation based on the *scikit-learn* library (Pedregosa et al., 2011).

   Since LPMs inherently handle only binary class data, for a multiclass problem, we construct a *one-vs-rest* model, comprising of as many binary classifiers as there are distinct labels. The given size is enforced for *each* binary classifier. For instance, consider the dataset *letter* in Table 1, with 26 classes. A model size of 10 implies we construct 26 binary classifiers, each with 10 terms. We have not used the more common *Logistic Regression* classifier because: (1) from the perspective of interpretability, LPMs provide a better sense of variable importance (Mood, 2010) (2) the technique is well validated for the case of linear classification by any standard linear classifier.

   **Sizes**: For a dataset with dimensionality $d$, we construct models of sizes:
   $\{1, 2, ..., min(d, 15)\}$. We end up with sizes less than 15 only for the dataset *cod-rna*, which has $d = 8$. All other datasets have $d > 15$ (see Table 1).

   **Hyperparameters**: The specified size is used for training both $M_{base}$ and the models within iterations. No other hyperparameters are explored.

2. *Decision Trees (DT)*: We use the implementation of CART in the *scikit-learn* library. Our notion of size here is the depth of the tree.

   **Sizes**: For a dataset, we first learn a tree (with no size constraints) with the highest *F1-score* (macro) using standard $5-$fold cross-validation. We refer to this as the optimal tree $T_{opt}$, and its depth is denoted by $depth(T_{opt})$. We then experiment with model sizes $\eta \in \{1, 2, ..., min(depth(T_{opt}), 15)\}$. This is controlled by setting the values of CART's *max_depth* to $\eta$.

   Stopping early makes sense since the model is saturated in its learning from the data; changing the input distribution is not helpful beyond this point.

   Note that while our notion of size is the *actual* depth of the tree produced, the parameter we vary is *max_depth*; this is because decision tree libraries do not allow specification of an exact tree depth[16]. This is important to remember since we might not see actual tree depths take all values in $\{1, 2, ..., min(depth(T_{opt}), 15)\}$, e.g., *max_depth* = 5 might give us a tree with $depth = 5$, *max_depth* = 6 might also result in a tree with $depth = 5$, but *max_depth* = 7 might give us a tree with $depth = 7$. We report improvements *at actual depths*, although the parameter controlled is *max_depth*.

   **Hyperparameters**: In learning $M_{base}$, in addition to the specified *max_depth*, the parameter space *min_impurity_decrease* $\in \{0, 0.25, 0.5, 0.75, 1\}$ is explored. This setting allows a node to be split only if it decreases impurity by an amount that is greater than or equal to set value. Within an iteration, only *max_depth* is used as hyperparameter.

---

[16]The training phase may be declared complete before growing till *max_depth*, based on other settings like leaf purity, minimum number of samples required at a leaf, etc.

Table 1: We use the following datasets available on the LIBSVM website (Chang & Lin, 2011). Their original source is mentioned in the "Description" column. 10000 instances from each dataset are used. A $train : val : test$ split ratio of $60 : 20 : 20$ is used for $D_{train}, D_{val}$ and $D_{test}$ in Algorithm 3. The splits are stratified wrt labels.

| S.No. | Dataset | Dimensions | # Classes | Label Entropy | Description |
|---|---|---|---|---|---|
| 1 | cod-rna | 8 | 2 | 0.92 | Predict presence of non-coding RNA common to a pair of RNA sequences, based on individual sequence properties and their similarity (Uzilov et al., 2006). |
| 2 | ijcnn1 | 22 | 2 | 0.46 | Time series data produced by an internal combustion engine is used to predict normal engine firings vs misfirings (Prokhorov, 2001). Transformations as in Chang & Lin (2001). |
| 3 | higgs | 28 | 2 | 1.00 | Predict if a particle collision produces Higgs bosons or not, based on collision properties (Baldi et al., 2014). |
| 4 | covtype.binary | 54 | 2 | 1.00 | Modification of the *covtype* dataset (see row 12), where classes are divided into two groups (Collobert et al., 2002). |
| 5 | phishing | 68 | 2 | 0.99 | Various website features are used to predict if the website is a *phishing* website (Mohammad et al., 2012). Transformations used as in Juan et al. (2016) |
| 6 | a1a | 123 | 2 | 0.80 | Predict whether a person makes over 50K a year, based on census data variables (Dua & Graff, 2017). Transformations as in Platt (1998). |
| 7 | pendigits | 16 | 10 | 1.00 | Classify handwritten digit samples into the digits 0-9 (Alimoglu & Alpaydin, 1996; Dua & Graff, 2017). |
| 8 | letter | 16 | 26 | 1.00 | Images of the capital letters A-Z were produced by random distortion of these characters from 20 fonts. The task is to classify these character images as one of the original letters (Michie et al., 1995). Transformations as in Hsu & Lin (2002). |
| 9 | Sensorless | 48 | 11 | 1.00 | Based on phase current measurements of an electric motor, predict different error conditions (Paschke et al., 2013). We use the transformations from Wang et al. (2018). |
| 10 | senseit_aco | 50 | 3 | 0.95 | Predict vehicle type using acoustic data gathered by a sensor network (Duarte & Hu, 2004). |
| 11 | senseit_sei | 50 | 3 | 0.94 | Predict vehicle type using seismic data gathered by a sensor network (Duarte & Hu, 2004). |
| 12 | covtype | 54 | 7 | 0.62 | Predict forest cover type from cartographic variables (Dean & Blackard, 1998; Dua & Graff, 2017). |
| 13 | connect-4 | 126 | 3 | 0.77 | Predict if the first player wins, loses or draws, based on board positions of the board game *Connect Four* (Dua & Graff, 2017). |

### 4.1.4  Oracles

We want our oracle models $\mathcal{O}$ to be fairly accurate, so that the derived uncertainty information is reliable. Hence we pick the following model families:

1. *Gradient Boosted Models (GBM)*: We used a gradient boosting model with DTs as our base classifiers. The *LightGBM* library (Ke et al., 2017) is used in our experiments. Effective parameters were determined using a validation set. **NOTE:** This is *not $D_{val}$* from Algorithm 3, since that would constitute *data leakage*. A sample, stratified by labels, from within $D_{train}$ was held out for learning good *GBM* parameters.

2. *Random Forests (RF)*: We used the implementation available in *scikit-learn*. Parameters were learned using 5-fold cross-validation over $D_{train}$.

The above oracles were calibrated (Platt, 1999) for reliable probability estimates.

### 4.1.5  Optimization Search Space

The optimizer we use, TPE, requires *box constraints*. Here we specify our search space for the optimization variables, $\Phi$ in Algorithm 3:

1. $p_o$: We want to allow the algorithm to pick an arbitrary fraction of samples from the original data; we set $p_o \in [0, 1]$.

2. $N_s$: We set $N_s \in [400, 10000]$. The lower bound ensures we have statistically significant results. The upper bound is set to a reasonably large value.

3. $\{a, b, a', b'\}$: Each of these parameters are allowed a range $[0.1, 10]$ to allow for a wide range of shapes for the component *Beta* distributions.

4. *scale*: We fix $scale = 10000$ for our experiments, to allow for $A_i$ and $B_i$ to model skewed distributions where shape parameter large values might be required. For small values, the algorithm adapts by learning the appropriate $\{a, b, a', b'\}$.

5. $\alpha$: For a DP, $\alpha \in \mathbb{R}_{>0}$. We use a lower bound of 0.1.

   To determine the upper bound, we rely on the following empirical relationship (Ohlssen et al., 2007) between the number of components $k$ and $\alpha$:

$$E[k|\alpha] \approx 5\alpha + 2 \tag{12}$$

   We empirically estimated a fairly inclusive upper bound on the number of components to be 500, which provides us the $\alpha$ upper bound of 99.6. Thus, we use $\alpha \in [0.1, 99.6]$.

   We draw a sample from the IBMM using *Blackwell-MacQueen* sampling (Blackwell & MacQueen, 1973).

   We use a flattening transformation (discussed in Section 3.6) on the original uncertainty distributions, with a fixed number of 20 bins. *However, all visualizations of distributions in the following sections were prepared after performing an inverse transformation*; hence, in studying them, it might be convenient to assume that no transformation was applied.

**Hyperparameters**: In theory, the box constraints and the iteration budget required by the optimizer constitute our hyperparameters, which may be tuned for a specific task. However, as we note above, we don't need to estimate a range for $p_o$ and reasonable defaults may be applied to $N_s$, $\{a, b, a', b'\}$, *scale* and $\alpha$. This results in the practical convenience of having to set the value for only a single hyperparameter: $T$, the iteration budget. This was set to $T = 1000$ for LPMs and $T = 3000$ for DTs based on limited search. Since

the LPMs we use construct multiple *one-vs-rest* classifiers, higher iteration budgets are computationally expensive to use.

□

This completes our discussion of the experimental setup; we present our observations next.

### 4.1.6 Improvements in Accuracy

Here, we use $M_{base}$ as the baseline model, and $\delta F1 \in [0, \infty)$.

Figure 3 shows the improvements for different combinations of interpretable and oracle models, $\{LPM, DT\} \times \{GBM, RF\}$. The model size is on the x-axis, and is normalized to be in $[0, 1]$, so that performance across datasets may be conveniently compared in the same plot.

For LPMs, the model sizes for a dataset, i.e., number of non-zero terms, are multiplied by $1/min(d, 15)$, where $d$ is the dimensionality of the data. For DTs, the model sizes are multiplied by $1/min(depth(T_{opt}), 15)$. All $\delta F1$ values are *averaged over five runs*, in the manner described in Section 4.1.1.

Table 2 enumerates the observations corresponding to the plots in Figure 3. The column *model_ora* represents the model and oracle combination used. For example, *dt_gbm* implies $DT$ was used as the model and $GBM$ as an oracle.

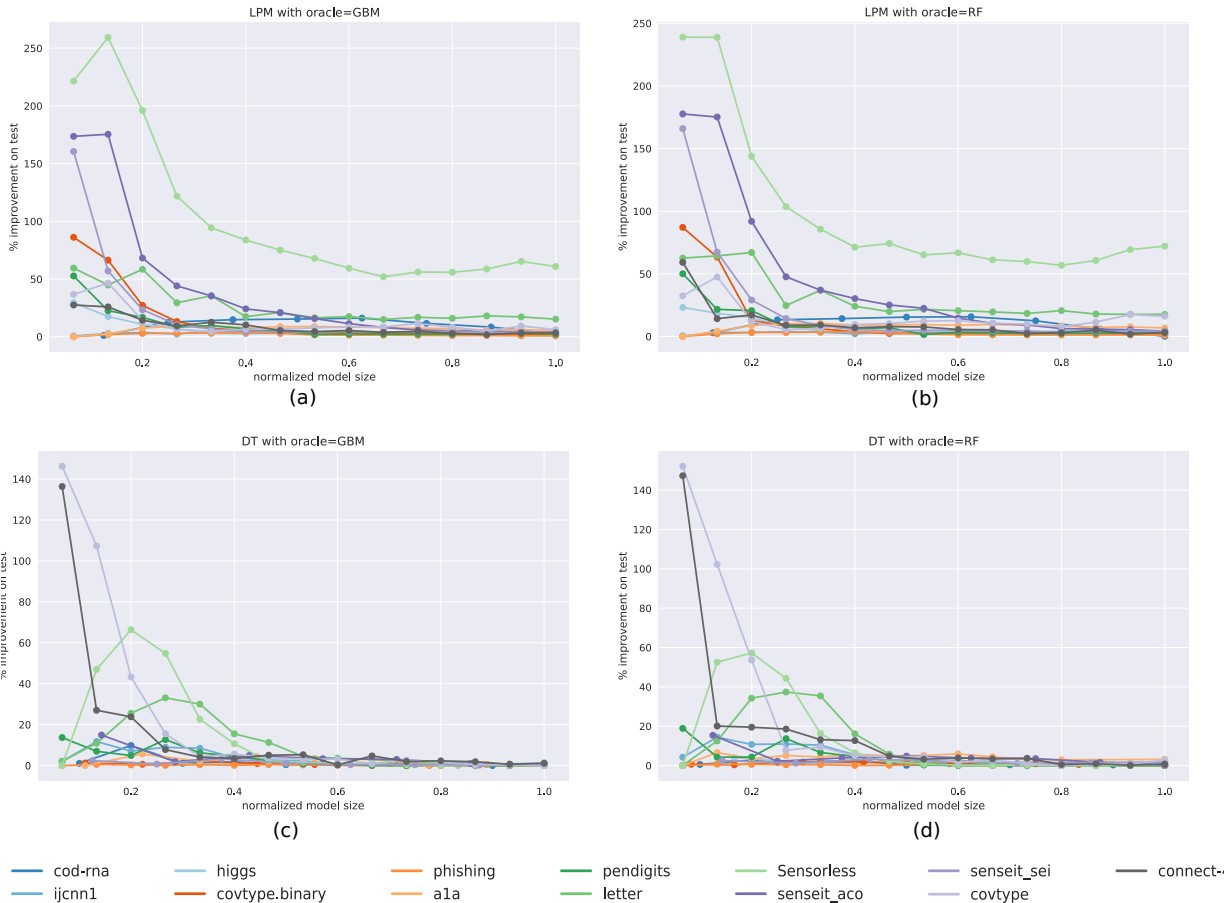

Figure 3: For different combinations of models and oracles: $\{LPM, DT\} \times \{GBM, RF\}$, these plots show improvements in $\delta F1$, averaged over **five** runs, for different model sizes and data. Table 2 shows the corresponding improvement scores.

We observe that the oracle based approach indeed works on a variety of datasets, across different combinations of interpretable and oracle models. In some cases, such as the dataset `Sensorless`, for the $LPM$ and $RF$ combination, improvements are as high as $\delta F1 = 238.94\%$. The general trend seems to be that $\delta F1$ decreases as model sizes increase, with eventually $\delta F1 \approx 0$. This decrease seems to be faster for $DT$s, which makes intuitive sense given that a unit increase in size for a $DT$ adds more representational power (a layer of nodes) than for an $LPM$ (another term), making it harder to beat the baseline performance of $DT$s.

This decrease empirically verifies the property expressed by Equations 2 and 3.

We note that $\delta F1$ does not strictly monotonically decrease for all datasets, possibly due to the optimization terminating at a local maxima, e.g., in Table 2 see the entry for `letter`, $lpm\_rf$, $size = 2$ ($improvement = 64.36\%$) and $size = 3$ ($improvement = 67.06\%$). But it largely appears to follow the general trend of decrease even in these cases.

Table 2: This table shows the average improvements, $\delta F1$, over **five** runs for different combinations of models and oracles: $\{LPM, DT\} \times \{GBM, RF\}$. The best improvement for a model size and oracle is indicated in bold. See Section A.8 for a discussion on running times.

| dataset | model_ora | 1 | 2 | 3 | 4 | 5 | 6 | 7 | 8 | 9 | 10 | 11 | 12 | 13 | 14 | 15 |
|---|---|---|---|---|---|---|---|---|---|---|---|---|---|---|---|---|
| cod-rna | lpm_gbm | 1.11 | 12.65 | **14.80** | 15.36 | **16.18** | 11.60 | **8.41** | **2.61** | - | - | - | - | - | - | - |
| | lpm_rf | **2.95** | **13.37** | 14.31 | **15.49** | 15.78 | **12.53** | 5.89 | 0.18 | - | - | - | - | - | - | - |
| | dt_gbm | **1.09** | **9.80** | 1.09 | 1.84 | **0.44** | **0.75** | **1.08** | 0.42 | 0.00 | **0.28** | - | - | - | - | - |
| | dt_rf | 0.57 | 2.78 | **1.81** | **2.58** | 0.13 | 0.46 | 0.40 | **0.70** | **0.31** | 0.00 | - | - | - | - | - |
| ijcnn1 | lpm_gbm | **0.59** | 2.84 | **3.71** | 2.39 | **4.97** | **4.61** | **3.85** | **3.97** | **3.46** | 2.34 | **3.00** | 2.74 | 2.85 | 3.46 | 2.62 |
| | lpm_rf | 0.50 | **3.26** | 3.22 | **3.88** | 3.43 | 2.17 | 3.50 | 3.21 | 3.43 | **4.05** | 2.84 | **4.37** | **3.99** | **3.48** | **4.23** |
| | dt_gbm | 2.10 | 11.75 | 7.07 | 8.95 | 8.47 | 3.89 | 2.67 | **2.36** | 0.60 | 0.39 | 0.02 | 0.29 | 0.67 | 0.74 | 0.39 |
| | dt_rf | **4.24** | **14.43** | **10.77** | **11.00** | **10.38** | **5.46** | **4.74** | 2.32 | **2.76** | **1.25** | **1.72** | **1.31** | **1.70** | **1.24** | **1.70** |
| higgs | lpm_gbm | **29.54** | 17.59 | 10.79 | 6.81 | 2.88 | 2.69 | **3.12** | **2.91** | **2.86** | **2.39** | **2.59** | 1.59 | 2.17 | **1.96** | 0.76 |
| | lpm_rf | 23.18 | **18.59** | **15.03** | **7.96** | **4.33** | **3.61** | 2.29 | 2.55 | 1.78 | 1.34 | 2.02 | **2.22** | **2.79** | 1.63 | **1.51** |
| | dt_gbm | 1.62 | 0.59 | 1.22 | 0.75 | 0.01 | **1.41** | - | - | - | - | - | - | - | - | - |
| | dt_rf | **3.98** | **0.85** | **1.90** | **1.63** | **1.69** | 0.79 | - | - | - | - | - | - | - | - | - |
| covtype.binary | lpm_gbm | 86.19 | **66.39** | **27.19** | **13.19** | **7.26** | **5.47** | **4.01** | **3.95** | **3.26** | **3.36** | **3.20** | **3.06** | **2.49** | **2.39** | 1.46 |
| | lpm_rf | **87.10** | 63.38 | 12.76 | 8.33 | 6.25 | 3.75 | 2.49 | 2.41 | 2.76 | 2.77 | 2.41 | 2.67 | 2.42 | 2.34 | **2.19** |
| | dt_gbm | **1.24** | **0.62** | **2.09** | 0.99 | 0.52 | 1.02 | 0.15 | 0.50 | 0.06 | - | - | - | - | - | - |
| | dt_rf | 0.68 | 0.40 | 1.61 | **2.01** | **1.70** | **1.45** | **1.04** | **1.30** | **0.97** | 0.50 | - | 0.00 | - | - | - |
| phishing | lpm_gbm | **0.00** | 1.88 | 2.88 | 3.05 | 3.22 | 3.37 | 2.86 | 1.61 | 1.37 | 1.44 | **1.21** | 1.03 | 1.07 | 0.84 | 0.86 |
| | lpm_rf | 0.00 | **2.14** | **3.29** | **3.22** | **3.59** | **3.79** | **3.29** | **1.85** | **1.46** | **1.46** | 1.18 | **1.18** | **1.22** | **1.27** | **1.08** |
| | dt_gbm | **0.00** | 0.57 | 0.33 | 0.13 | **0.44** | 0.11 | **0.48** | 0.33 | **0.13** | 0.00 | 0.01 | 0.00 | 0.00 | 0.00 | **0.05** |
| | dt_rf | 0.00 | **0.72** | **0.61** | **0.44** | 0.44 | 0.08 | 0.12 | **0.42** | 0.13 | **0.07** | **0.10** | **0.06** | **0.04** | **0.01** | 0.00 |
| a1a | lpm_gbm | 0.00 | 2.55 | 7.58 | 9.11 | 9.03 | 7.87 | 8.72 | 8.86 | 8.56 | 7.90 | 7.38 | 7.14 | 5.78 | 6.15 | 5.56 |
| | lpm_rf | **0.02** | **4.17** | **8.81** | **10.24** | **10.46** | **9.11** | **9.18** | **9.52** | **8.97** | **9.70** | **8.98** | **8.50** | **7.34** | **7.67** | **6.77** |
| | dt_gbm | **0.01** | 5.67 | 2.10 | 4.33 | 3.53 | 2.91 | 0.40 | 0.64 | 0.27 | - | - | - | - | - | - |
| | dt_rf | 0.00 | **6.62** | **3.44** | **5.14** | **4.36** | **5.70** | **4.99** | **5.14** | **5.92** | 4.43 | 3.02 | 2.94 | - | - | 3.16 |
| pendigits | lpm_gbm | **52.66** | **22.62** | 16.88 | **8.34** | **9.73** | **6.90** | 5.12 | **2.03** | 2.44 | 2.21 | 2.31 | 2.13 | **3.26** | **3.03** | 2.39 |
| | lpm_rf | 50.10 | 21.68 | **20.70** | 7.77 | 8.16 | 6.15 | **7.04** | 1.61 | **3.38** | **2.97** | **2.45** | **2.48** | 2.75 | 2.65 | **2.97** |
| | dt_gbm | 13.70 | **6.98** | **4.92** | 12.72 | 6.21 | **4.68** | **2.40** | **0.87** | **0.48** | **0.04** | **0.00** | **0.19** | **0.00** | **0.00** | **0.01** |
| | dt_rf | **18.93** | 4.26 | 4.36 | **13.70** | **6.67** | 4.58 | 2.38 | 0.32 | 0.00 | 0.02 | 0.00 | 0.00 | 0.00 | 0.00 | 0.00 |
| letter | lpm_gbm | 59.54 | 44.83 | 58.49 | **29.47** | 35.49 | 17.43 | **21.06** | 16.47 | 17.48 | 15.02 | 16.88 | 15.98 | **18.10** | 17.24 | 15.30 |
| | lpm_rf | **62.61** | **64.36** | **67.06** | 24.71 | **36.95** | **24.14** | 19.88 | **21.70** | **20.63** | **19.64** | **18.42** | **20.65** | 17.93 | **17.71** | **17.67** |
| | dt_gbm | 2.10 | 10.91 | 25.55 | 33.11 | 30.04 | 15.54 | **11.32** | 4.47 | **3.59** | **3.06** | **1.85** | **1.26** | **1.08** | **0.62** | **0.35** |
| | dt_rf | 0.12 | **12.53** | **34.24** | **37.39** | **35.43** | **16.14** | 5.82 | 1.22 | 2.30 | 0.71 | 0.49 | 0.16 | 0.00 | 0.00 | 0.00 |
| Sensorless | lpm_gbm | 221.53 | **259.30** | **195.99** | **121.89** | **94.41** | **83.82** | **75.07** | **67.90** | 59.42 | 51.98 | 56.11 | 55.86 | 58.70 | 65.27 | 60.88 |
| | lpm_rf | **238.94** | 238.83 | 143.82 | 103.65 | 85.69 | 71.32 | 74.25 | 65.22 | **66.78** | **61.20** | **59.88** | 56.84 | **60.67** | **69.39** | **72.12** |
| | dt_gbm | **0.04** | 46.99 | **66.38** | **54.79** | **22.65** | **10.67** | **2.28** | **1.69** | **0.84** | **0.85** | **0.41** | **0.14** | **0.16** | 0.16 | 0.07 |
| | dt_rf | 0.01 | **52.54** | 57.27 | 44.33 | 16.26 | 6.49 | 2.23 | 0.69 | 0.27 | 0.06 | 0.29 | 0.08 | 0.00 | **0.23** | **0.16** |
| senseit_aco | lpm_gbm | 173.63 | **175.44** | 68.21 | 44.09 | 35.41 | 24.18 | 20.83 | 15.80 | 11.39 | 8.09 | 5.83 | 5.21 | 4.77 | 4.21 | 3.95 |
| | lpm_rf | **177.67** | 175.20 | **91.96** | **47.67** | **37.03** | **30.28** | **25.19** | **22.54** | **14.74** | **10.46** | **9.28** | **6.31** | **5.91** | **5.46** | **4.23** |
| | dt_gbm | 14.92 | 1.83 | **4.89** | 3.35 | 3.03 | 0.97 | 0.57 | - | - | - | - | - | - | - | - |
| | dt_rf | **15.41** | **2.22** | 3.88 | **4.82** | **3.81** | **3.54** | **1.66** | 0.25 | - | - | - | - | - | - | - |
| senseit_sei | lpm_gbm | 160.59 | 57.00 | 23.42 | 10.47 | 6.70 | 4.49 | 4.49 | 4.12 | 4.55 | 4.14 | **4.40** | **4.91** | 3.83 | 3.97 | **4.29** |
| | lpm_rf | **165.98** | **67.35** | **29.13** | **14.35** | **8.80** | **5.53** | **4.72** | **4.90** | **5.11** | **4.47** | 4.39 | 3.58 | **4.16** | **4.26** | 4.15 |
| | dt_gbm | 2.42 | 0.75 | **3.26** | 1.23 | 0.39 | 0.42 | 0.27 | 0.27 | - | - | - | - | - | - | - |
| | dt_rf | **2.54** | **1.28** | 3.23 | **2.26** | **1.18** | **1.49** | **1.91** | - | - | - | - | - | - | - | - |
| covtype | lpm_gbm | 36.69 | 46.55 | **14.35** | **6.51** | 4.64 | 6.49 | 6.73 | 7.72 | 8.42 | 8.39 | **11.47** | **8.39** | 4.73 | 9.37 | 6.17 |
| | lpm_rf | 32.52 | **47.56** | 12.23 | 5.46 | **6.46** | **9.65** | **10.31** | **12.33** | **12.62** | **10.56** | 9.86 | 7.85 | **11.52** | **17.49** | **16.24** |
| | dt_gbm | 146.22 | **107.32** | 43.31 | **15.53** | 4.30 | **5.64** | 3.83 | 3.22 | **3.15** | 0.93 | **2.68** | **1.56** | 0.63 | 0.34 | 0.00 |
| | dt_rf | **152.12** | 102.28 | **53.72** | 7.86 | **9.41** | 4.97 | **4.67** | **4.42** | 1.68 | **2.86** | 1.08 | 0.00 | **0.85** | **0.90** | **2.79** |
| connect-4 | lpm_gbm | 27.54 | **25.89** | 14.36 | **9.34** | **12.56** | **10.29** | 5.47 | 4.48 | 5.45 | 3.79 | **4.72** | 2.99 | 1.92 | **3.31** | **3.46** |
| | lpm_rf | **59.19** | 14.24 | **17.20** | 9.00 | 9.22 | 6.98 | **7.87** | **7.65** | **5.58** | **5.32** | 2.40 | **3.04** | **4.94** | 1.94 | 3.23 |
| | dt_gbm | 136.35 | **27.02** | **23.83** | 7.80 | 4.13 | 3.46 | **5.09** | **5.31** | 0.39 | **4.73** | 2.02 | **2.31** | **1.90** | **0.62** | **1.14** |
| | dt_rf | **147.32** | 20.08 | 19.51 | **18.57** | **13.17** | **12.70** | 4.82 | 3.23 | **3.90** | 3.41 | **3.72** | 0.53 | 0.92 | 0.19 | 0.80 |

**Statistical Significance**: We perform the *Wilcoxon signed-rank test* (Wilcoxon, 1945) to measure the statistical significance of the $\delta F1$ scores. These are shown in Figure 4. We use this test as it has been shown to be useful in comparing classifiers (Demšar, 2006; Benavoli et al., 2016; Japkowicz & Shah, 2011). The test setup is as follows:

1. We compare the classifiers learned by our technique with the baseline, for a given range of model sizes. Separate tests are performed for different model size ranges since size strongly influences $\delta F1$.

2. Normalized model sizes are used for ease of comparison with Figure 3. Binning of model sizes is done using *Sturges rule* (Sturges, 1926).

3. The *one-sided* version of the *paired* test is performed for each bin, where pairs of scores $F1_{baseline}$ and $F1_{new}$ for a dataset, for models with sizes assigned to the bin, are compared. In cases were where multiple model sizes for a dataset fall within the same bin, $F1_{baseline}$ and $F1_{new}$ are first averaged and then compared.

4. The following hypotheses are tested:

   - **$H_0$**, null hypothesis: accuracies of models trained using the oracle are not better.
   - **$H_1$**, alternate hypothesis: accuracies of models trained using the oracle are better.

   *p-values* are shown for each bin. Small *p-values* favor **$H_1$**, i.e., our algorithm.

5. Scores of $\delta F1 = 0$ are split equally between positive and negative ranks[17].

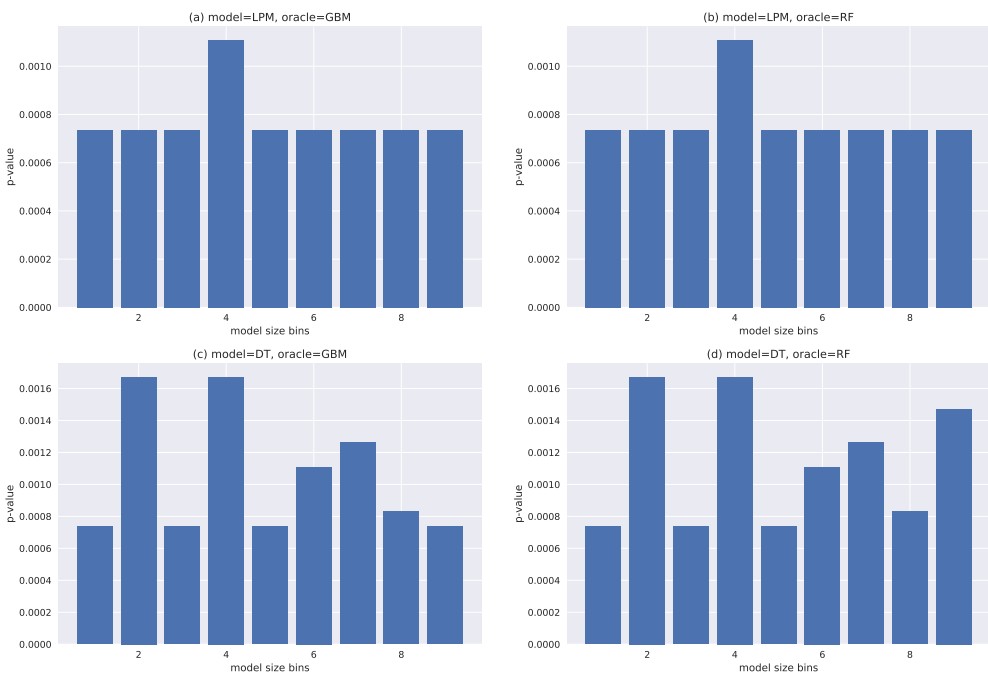

Figure 4: These plots show the *p-values* for the Wilcoxon signed-rank test, with the null hypothesis $H_0$: using the oracle does *not* produce better F1 test scores. The bin boundaries are selected using the *Sturges* rule (Sturges, 1926). Low *p-values* favor our algorithm.

We observe that the improvements from using an oracle are indeed significant for various model size and model-oracle combinations, when measured across multiple datasets.

$\square$

Before we conclude this section, we present an additional way to visualize improvements: create a correspondence of model sizes, without and with our technique, for the same accuracy. See Figure 5 as an example. The point $(12, 2)$ for `senseit_aco` implies that the accuracy of a LPM with 2 non-zero terms produced by our technique equals, or is greater than, the accuracy of a baseline LPM with 12 non-zero terms. The model size on the y-axis is the median of five runs. We refer to such a plot as the *compaction profile* for a model-oracle combination. See Section A.10 for more compaction profiles.

---

[17]The `zplit` option in `https://numpy.org/doc/stable/reference/generated/numpy.histogram_bin_edges.html` is used.

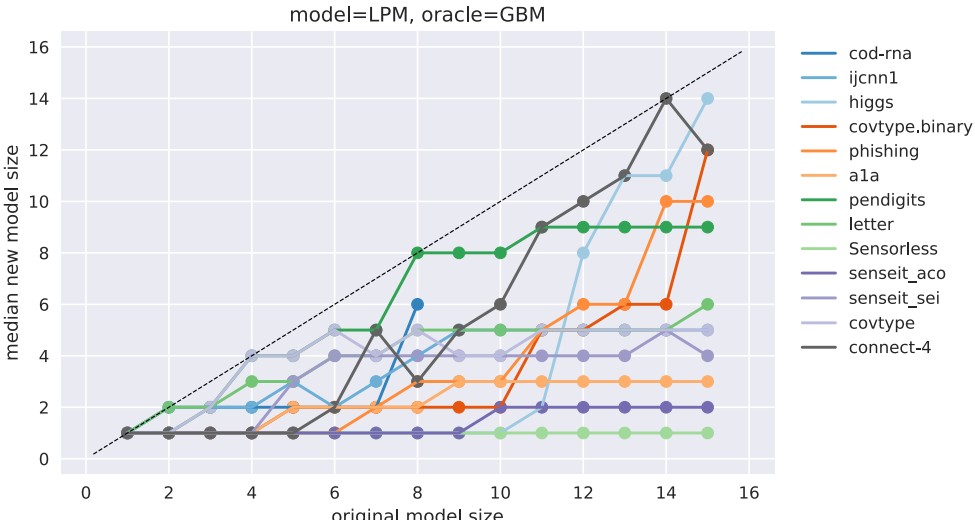

Figure 5: The compaction profile of $LPM$ models using $GBM$ as an oracle. A point $(x, y)$ denotes the minimum size $y$ of a model obtained using our technique that is *at least* as accurate as the baseline model of size $x$.

### 4.1.7 Learned Distributions

It is also instructive to analyse the distributions we have learned: this includes both the parameter $p_o$ and the parameters for the IBMM $\Psi = \{\alpha, a, b, a', b'\}$.

Figure 6 shows how $p_o$ varies with normalized model size when the interpretable model is DT and the oracle is (a) GBM or (b) RF. This plot ignores the datasets where the largest tree depth explored was less than $depth(T_{opt})$ - so we can compare distributions in size regimes where our technique is effective against when it is not (recall, at sizes close to $depth(T_{opt})$ we expect $\delta F1 \approx 0$). The datasets ignored are[18]: `a1a, ijcnn1, covtype, connect-4`. Here, we clearly see $p_o \rightarrow 1$ as model size increases, thus implying the training algorithm tends to use more of the original distribution[19]. This observation is a **key contribution** of this work, since it challenges the conventional wisdom that the training data must be drawn from the same distribution as the test data, for effective learning. This reinforces a similar observation from Ghose & Ravindran (2020).

Section A.7 in the Appendix presents visualizations and analysis of the learned IBMMs.

---

[18]These datasets are easy to identify in Table 2: the ones where the last column(s) is neither $\approx 0$ nor "-".

[19]In theory, the parameters $\Psi$ could have been learned such that they mimic the original distribution, but we hypothesize that it is easier for the optimizer to learn the appropriate value of one parameter $p_o$ as opposed to equivalent values of the multiple parameters $\Psi$. This is why we see the clear pattern in Figure 6.

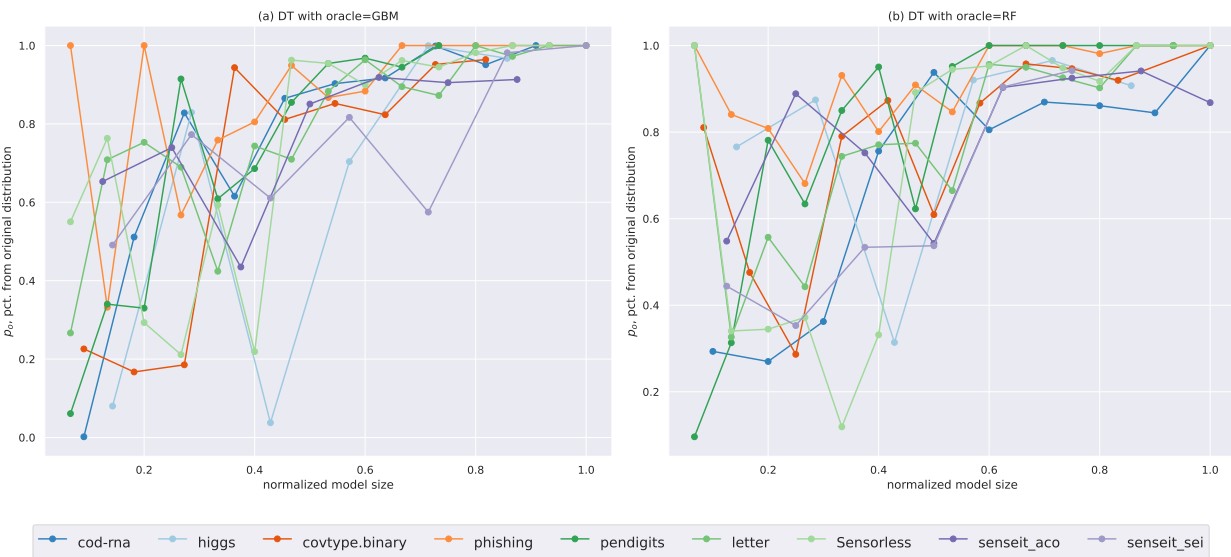

Figure 6: These plot shows the effect of increasing model size on $p_o$, when the interpretable model is DT. These plots strongly indicate that larger model sizes learn better with the original distribution. Some datasets are ignored - see text for explanation.

### 4.1.8 Optimizer Performance

Here we use $M_{init}$ as the baseline for computing $\delta F1$. As discussed in Section 4.1.1, $\delta F1 \in (-\infty, \infty)$.

These improvements are shown in Table 3, where negative improvements are indicated in red. Also note that some of the small positive $\delta F1$ scores from Table 2, e.g., dataset a1a for model size of one, now become exactly 0. In general, scores in Table 3 are conservative compared to Table 2. We also note that there aren't many instances of negative scores.

Table 3: This table shows the average improvements, $\delta F1$, over five runs for different combinations of models and oracles: $\{LPM, DT\} \times \{GBM, RF\}$. The improvements are measured relative to the model at the first iteration.The best improvement for a model size and oracle is indicated in bold. Here, $\delta F1 \in (-\infty, \infty)$. Negative improvements are shown in red.

| dataset | model_ora | 1 | 2 | 3 | 4 | 5 | 6 | 7 | 8 | 9 | 10 | 11 | 12 | 13 | 14 | 15 |
|---|---|---|---|---|---|---|---|---|---|---|---|---|---|---|---|---|
| cod-rna | lpm_gbm | 1.39 | 12.53 | **14.76** | **15.73** | 14.97 | 12.00 | 0.00 | **0.08** | - | - | - | - | - | - | - |
|  | lpm_rf | **2.66** | **13.91** | 14.69 | 15.34 | **16.06** | **12.49** | **8.30** | 0.00 | - | - | - | - | - | - | - |
|  | dt_gbm | **0.00** | **0.00** | 0.00 | 1.26 | 0.00 | **0.00** | 0.00 | 0.00 | -0.28 | 0.08 | - | - | - | - | - |
|  | dt_rf | 0.00 | 0.00 | **1.78** | **2.28** | **0.39** | -0.02 | **0.17** | **0.47** | **0.00** | **0.72** | - | - | - | - | - |
| ijcnn1 | lpm_gbm | -0.16 | **3.36** | **3.93** | 0.00 | **5.19** | **4.18** | **3.85** | **3.79** | **3.69** | 2.99 | **2.97** | 3.21 | 3.11 | 3.26 | 3.02 |
|  | lpm_rf | **0.19** | 2.80 | 3.36 | **3.65** | 1.94 | 3.58 | 3.30 | 3.46 | **3.81** | 2.66 | **4.65** | **3.99** | **3.82** | **4.85** |  |
|  | dt_gbm | 1.96 | 12.00 | **10.15** | **11.37** | **10.63** | 7.18 | 3.63 | **4.52** | **2.91** | **1.78** | **1.93** | **2.29** | 1.47 | **2.26** | 0.00 |
|  | dt_rf | **4.06** | **12.10** | 8.95 | 10.75 | 10.13 | **8.25** | **5.38** | 2.46 | 2.63 | 1.25 | 1.46 | 1.37 | **1.91** | 0.00 | **1.38** |
| higgs | lpm_gbm | **29.29** | **17.80** | 11.40 | 6.56 | 3.06 | 2.68 | **3.16** | **2.90** | **2.67** | **2.82** | 2.65 | 1.79 | 2.62 | 2.19 | 1.63 |
|  | lpm_rf | 26.71 | 17.29 | **15.06** | **10.60** | **5.35** | **4.04** | 2.35 | 2.03 | 1.66 | 1.89 | **2.91** | **2.94** | **3.31** | **2.58** | **2.22** |
|  | dt_gbm | 0.00 | 0.00 | **1.86** | 0.26 | 0.93 | 0.45 | - | - | - | - | - | - | - | - | - |
|  | dt_rf | **4.04** | **1.26** | 1.74 | **1.32** | **1.54** | **0.91** | - | - | - | - | - | - | - | - | - |
| covtype.binary | lpm_gbm | 76.52 | **66.39** | **29.17** | **12.51** | **9.18** | **5.28** | **4.94** | **4.56** | **3.92** | **3.56** | **3.62** | **3.31** | **2.59** | **2.83** | **2.39** |
|  | lpm_rf | **96.77** | 63.38 | 14.36 | 9.61 | 6.79 | 3.94 | 2.93 | 2.81 | 2.96 | 2.84 | 2.31 | 2.26 | 2.00 | 2.43 | 2.22 |
|  | dt_gbm | **0.00** | **0.00** | **2.35** | 1.27 | 1.18 | 1.11 | 0.00 | 0.00 | 0.00 | - | - | - | - | - | - |
|  | dt_rf | 0.00 | 0.00 | 2.10 | **2.33** | **2.44** | **2.39** | **1.84** | **2.19** | **1.65** | 0.70 | - | 0.89 | - | - | - |
| phishing | lpm_gbm | **0.00** | 1.88 | 2.88 | 3.05 | 3.22 | 3.25 | 2.99 | 1.69 | 1.42 | **1.45** | **1.29** | 0.00 | 0.00 | 0.00 | 0.00 |
|  | lpm_rf | 0.00 | **2.14** | **3.29** | **3.22** | **3.59** | **3.79** | **3.29** | **2.05** | **1.42** | 1.44 | 1.24 | **1.23** | **1.16** | **1.26** | **1.02** |
|  | dt_gbm | **0.00** | 0.00 | **0.00** | 0.07 | **0.39** | 0.00 | 0.28 | 0.22 | 0.44 | 0.23 | 0.00 | **0.00** | 0.00 | 0.00 | **0.00** |
|  | dt_rf | 0.00 | **0.72** | 0.00 | **0.57** | 0.00 | -0.17 | 0.13 | **0.48** | 0.13 | 0.05 | **0.03** | -0.03 | -0.28 | 0.00 | -0.16 |
| a1a | lpm_gbm | 0.00 | 2.55 | 7.58 | 8.98 | 8.40 | 8.03 | 8.90 | 8.23 | 8.17 | 7.90 | 5.96 | 7.10 | 6.97 | 6.18 | 5.73 |
|  | lpm_rf | 0.00 | **4.17** | **8.81** | **9.92** | **9.88** | **9.47** | **8.99** | **9.31** | **9.19** | **9.26** | **9.33** | **8.25** | **7.15** | **7.55** | **7.98** |
|  | dt_gbm | **0.00** | 5.54 | 2.39 | 3.84 | **3.55** | 2.55 | 1.51 | 2.25 | 4.87 | - | - | - | - | - | - |
|  | dt_rf | 0.00 | **6.44** | **4.36** | **5.60** | 3.40 | **5.94** | **6.06** | **4.97** | **4.89** | 4.01 | 4.73 | 5.21 | - | - | 4.53 |
| pendigits | lpm_gbm | **51.39** | **23.44** | 16.18 | **8.95** | **8.84** | **6.63** | 4.86 | **1.83** | 2.27 | 2.16 | **2.44** | 2.16 | **3.33** | 2.97 | **2.73** |
|  | lpm_rf | 46.28 | 22.74 | **21.72** | 8.80 | 8.47 | 6.29 | **6.48** | 1.69 | **3.03** | **2.79** | 2.34 | **2.68** | 2.70 | **3.02** | 0.00 |
|  | dt_gbm | 14.02 | **6.72** | 5.11 | 13.14 | 6.42 | 4.20 | 2.46 | **1.09** | **0.98** | **0.16** | -0.26 | **0.00** | **0.00** | **0.00** | **0.00** |
|  | dt_rf | **21.46** | 4.18 | **5.22** | **14.51** | **7.36** | **4.55** | **2.86** | 0.00 | 0.00 | 0.00 | **0.00** | 0.00 | 0.00 | 0.00 | 0.00 |
| letter | lpm_gbm | 57.06 | 48.48 | 59.85 | **29.76** | **36.09** | 19.27 | 20.37 | 16.08 | 17.55 | 15.16 | 17.26 | 16.51 | 18.46 | 17.19 | 15.55 |
|  | lpm_rf | **61.06** | **65.34** | **64.26** | 23.69 | 35.20 | **26.15** | **22.10** | **20.74** | **20.91** | **20.31** | **19.28** | **21.40** | **20.77** | **19.39** | **18.18** |
|  | dt_gbm | 0.00 | **13.98** | 25.05 | **33.96** | 32.05 | 15.49 | **11.17** | 0.00 | **4.26** | **3.50** | **1.99** | 0.00 | 0.00 | 0.00 | 0.00 |
|  | dt_rf | 0.00 | 12.21 | **28.67** | 33.47 | **33.51** | **18.41** | 8.10 | 0.00 | 1.84 | 1.21 | 1.31 | **0.67** | **0.61** | **0.11** | -0.08 |
| Sensorless | lpm_gbm | 216.47 | **257.56** | **178.31** | **117.01** | **90.70** | **83.90** | **73.50** | **65.95** | 61.57 | 57.97 | 56.54 | 57.15 | 55.45 | 66.24 | 68.24 |
|  | lpm_rf | **224.18** | 210.28 | 134.44 | 115.00 | 85.85 | 74.96 | 66.77 | 61.10 | **66.88** | **64.65** | **69.00** | **70.09** | **72.91** | **80.14** | **82.15** |
|  | dt_gbm | -0.01 | 42.42 | **68.13** | 44.38 | **17.39** | **10.32** | 1.82 | **1.44** | **0.79** | **0.64** | 0.41 | 0.12 | **0.00** | -0.02 | **0.34** |
|  | dt_rf | 0.00 | **52.54** | 57.10 | **44.61** | 16.63 | 6.19 | **2.19** | 0.96 | 0.51 | 0.00 | **0.48** | **0.33** | 0.00 | 0.00 | 0.10 |
| senseit_aco | lpm_gbm | 173.71 | 170.68 | 63.95 | **44.20** | 33.49 | 22.99 | 19.14 | 13.50 | 10.29 | 7.59 | 6.26 | 5.92 | **5.30** | 4.89 | 4.32 |
|  | lpm_rf | **177.67** | **181.26** | **79.86** | 42.86 | **37.60** | **28.80** | **23.75** | **19.06** | **13.91** | **10.74** | **8.48** | **6.09** | 5.20 | **5.32** | **4.62** |
|  | dt_gbm | 14.89 | 0.00 | **3.71** | 2.32 | **4.85** | 0.81 | **0.00** | - | - | - | - | - | - | - | - |
|  | dt_rf | **20.03** | **2.54** | 3.64 | **5.91** | 3.34 | **2.63** | 0.00 | 0.00 | - | - | - | - | - | - | - |
| senseit_sei | lpm_gbm | 160.59 | **65.27** | 23.44 | 10.48 | 6.76 | 4.86 | 4.82 | 4.46 | 4.79 | 4.12 | 4.54 | **5.17** | 3.91 | 4.21 | **4.46** |
|  | lpm_rf | **165.98** | 63.72 | **31.58** | **14.94** | **9.07** | **5.79** | **4.95** | **5.07** | **5.24** | **4.70** | **4.60** | 3.74 | **4.30** | **4.35** | 4.35 |
|  | dt_gbm | **2.66** | **1.01** | **3.49** | 2.29 | 0.95 | **1.30** | 1.37 | 0.00 | - | - | - | - | - | - | - |
|  | dt_rf | 2.33 | 0.00 | 3.36 | 1.65 | 0.87 | 0.00 | -1.23 | - | - | - | - | - | - | - | - |
| covtype | lpm_gbm | **36.87** | **49.24** | **12.78** | **11.21** | 7.84 | 7.15 | 7.15 | 8.07 | 7.70 | 8.25 | **10.94** | **8.35** | 4.37 | 8.77 | 5.84 |
|  | lpm_rf | 32.15 | 39.49 | 10.49 | 8.53 | **8.11** | **8.59** | **9.61** | **11.99** | **11.22** | **9.91** | 8.47 | 8.16 | **10.34** | **13.76** | **12.92** |
|  | dt_gbm | 342.27 | 92.85 | 43.23 | **20.04** | 8.14 | **8.05** | **5.67** | 3.26 | **4.92** | **3.52** | **2.72** | 0.00 | **0.00** | **0.00** | 1.74 |
|  | dt_rf | **354.45** | **98.94** | **50.87** | 14.10 | **9.46** | 7.38 | 4.76 | **4.20** | 0.94 | 1.81 | 2.30 | **0.71** | -0.37 | 0.00 | 0.00 |
| connect-4 | lpm_gbm | **37.62** | 11.66 | 12.01 | 6.84 | 5.68 | 6.82 | 4.58 | 2.10 | 3.82 | 3.21 | **3.02** | **3.64** | 2.32 | **2.97** | **3.40** |
|  | lpm_rf | 33.77 | **12.99** | **17.60** | **14.66** | **15.91** | **10.73** | **6.38** | **5.35** | **7.07** | **6.98** | 2.84 | 3.14 | 2.09 | 2.52 | 2.46 |
|  | dt_gbm | 89.33 | **29.23** | 20.20 | **12.10** | 9.73 | 9.88 | 7.82 | 7.43 | 0.57 | 4.61 | 1.08 | **3.35** | 2.23 | **1.15** | **1.55** |
|  | dt_rf | **113.71** | 21.91 | **20.52** | 11.23 | **16.86** | **10.96** | **10.64** | **9.11** | **6.51** | **5.88** | **6.76** | 2.16 | **2.97** | 0.61 | 0.00 |

**Statistical Significance**: We also perform a Wilcoxon signed-rank test to measure statistical significance of the $\delta F1$ scores here. The setup is identical to that in Section 4.1.6 : one-sided test, paired over datasets, performed separately for ranges of normalized bin sizes. Scores of $\delta F1 = 0$ are split equally between positive and negative ranks.

The results are shown in Figure 7.

We note that *p-values* are small here as well. In comparison to Figure 4, the *p-values* are typically larger. This is to be expected since we now have cases where $\delta F1 < 0$. Interestingly, for DTs, these plots show that

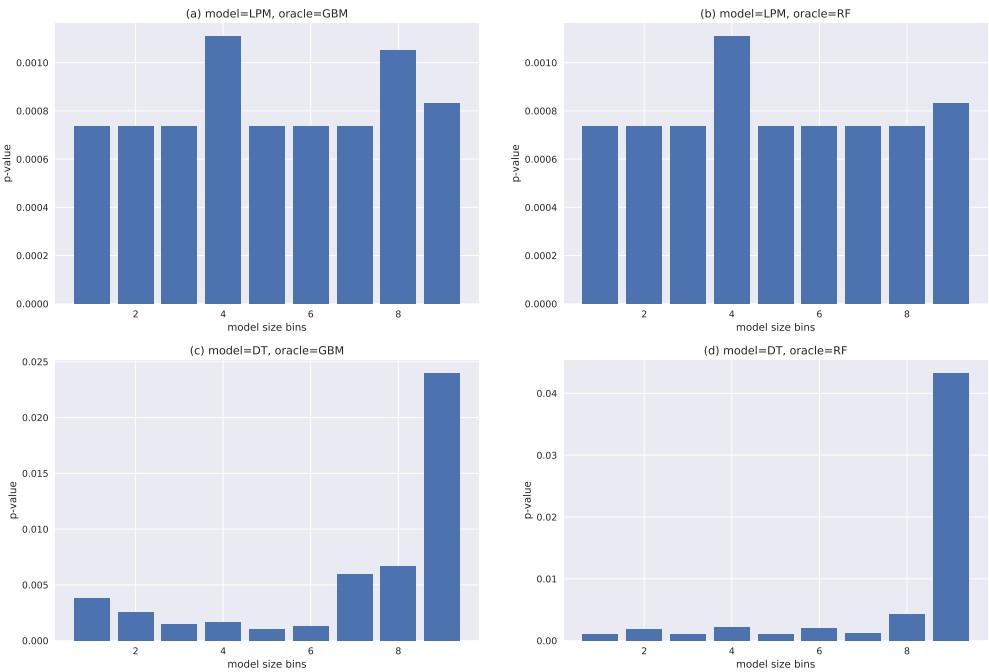

Figure 7: These plots show the *p-values* for the Wilcoxon signed-rank test, with the null hypothesis $H_0$: using the oracle does *not* produce better F1 test scores. The bin boundaries are selected using the *Sturges* rule (Sturges, 1926). Low *p-values* favor our algorithm.

at larger sizes our technique becomes less useful: we had noted this in our main results in Table 2, but this effect does not show up in Figure 4.

### 4.1.9   Effect of Model Capacity

If we closely look at the improvements in Table 2 or Figure 3, we would note that the improvements for DTs diminish faster than LPMs, as model size increases. This naturally leads to the question: how does model capacity influence improvements? This is difficult to answer in general since (a) there isn't a standard way to easily quantify capacity across model families, and (b) the notion of model size is subjective. And while the LPM vs DT data indicates a trend, we want to isolate this effect in a manner that is not affected by differences in the model families.

To that end, we adopt the following approach: we use two different instances of GBMs, where the notion of model size is the number of DTs in a GBM (or equivalently, the number of boosting rounds), and their model capacities are decided by the maximum depth of the constituent DTs; these are set to 2 and 5 for these GBM instances. We refer to these as the *GBM-2* and *GBM-5* "pseudo model families" respectively, where we understand *GBM-5* to possess higher capacity than *GBM-2*. Since the training algorithms and model representations are identical for *GBM-2* and *GBM-5*, this setup allows us to sidestep challenges with quantifying capacity for different model families.

The oracle used is another GBM, with no size/capacity restrictions, learned on the training dataset. The model sizes explored are $\{1, 2, ..., 10\}$. Figure 8 shows how $\delta F1$ varies with model size (denoted as "num_trees") for the datasets `senseit-aco, senseit-sei, cod-rna` and `higgs`, for each of the models *GBM-2* and *GBM-5*.

As we might expect, we observe that improvements for *GBM-5*, the model with the higher capacity, diminish faster with increasing size, compared to *GBM-2*.

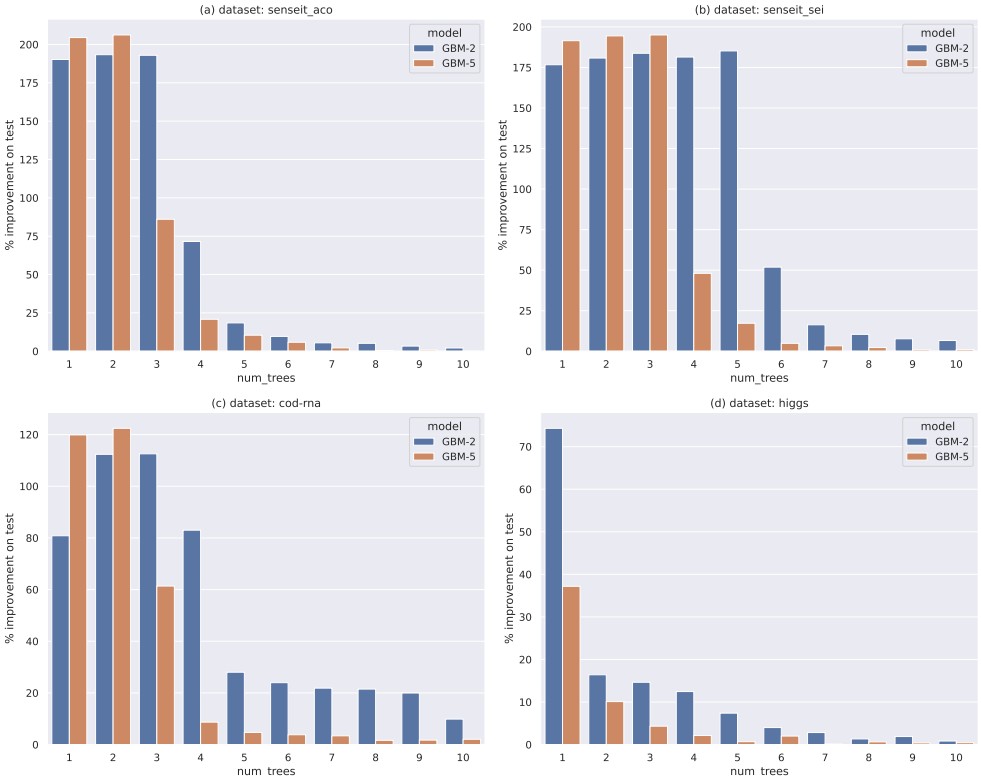

Figure 8: The above plots show how the capacity of a model family influences improvements, for different datasets. With a higher *max_depth* setting for GBMs, the improvements decline faster with an increase in number of trees.

## 4.2 Comparisons

In this section, we present a comparative evaluation of our technique. We compare against the following techniques:

1. *Supervised* uncertainty sampling: the interpretable model, of a given size, is iteratively trained on a growing subset of training data; this subset starts with the $b$ most uncertain points in the training data, with $b$-sized batches of the most uncertain points from the remaining training data being progressively added to it. At every iteration, the model is evaluated on a validation set, and the one with the highest $F1$-macro score is picked for comparison. We compare against this technique because:

   (a) This explores an obvious possibility: can a heuristic-driven, simple algorithm outperform our algorithm?

   (b) Although we borrow this technique from Active Learning (Lewis & Gale, 1994), this version is significantly more powerful, primarily because of the oracle's supervision: we have reliable uncertainty scores from a powerful model. Because of this, we are able to avoid sampling bias arising due to a partial view of the uncertainty distribution (detailed in Section A.3).

   (c) Even within the Active Learning community, uncertainty sampling is a strong baseline for Logistic Regression (Yang & Loog, 2018), and by extension, we expect it to be a strong baseline for learning LPMs.

   We use a batch size of $b = 10$. The algorithm is described in detail in Section A.2.

2. Density Trees: We also compare against our previous work on density trees since it uses a similar philosophy of determining an optimal distribution to build accurate small models. We use the parameter search space described in Ghose & Ravindran (2020).

### 4.2.1 Setup

The experimental setup is identical to the one used for the validation experiments in terms of the datasets (see Section 4.1.2), models (Section 4.1.3), oracles (Section 4.1.4) and optimization search space (Section 4.1.5). The metrics differ, and these are described next.

### 4.2.2 Metrics

To compare techniques, we wish to measure the following outcomes over multiple trials:

1. The extent to which a technique is better.

2. The proportion of times a technique is better.

The following properties are desirable for a metric that measures the first kind of outcomes:

1. It should be bounded, so that scores across different data, model sizes, etc., are on the same scale.

2. It should be easy to infer which approach is better.

We introduce a score called the *Scaled Difference in Improvement (SDI)*, that possesses these properties. The *SDI* is defined in terms of the improvement produced by our method, $\delta F1_{ora}$, and the alternative method, $\delta F1_{alt}$:

$$
SDI = \begin{cases} \frac{\delta F1_{ora}}{H} - \frac{\delta F1_{alt}}{H}, & \text{if } H > 0 \\ 0, & \text{if } H = 0 \end{cases} \tag{13}
$$
$$
\text{where } H = \max\{\delta F1_{alt}, \delta F1_{ora}\}
$$

The central idea here is that the improvements possible across the competing techniques are in $[0, H]$, and the *SDI* score measures the difference between the fractions of this range realized by either technique. Note that $H \geq 0$ since $\delta F1_{ora} \geq 0$ and $\delta F1_{alt} \geq 0$. This score has the following intuitive properties:

1. $SDI \in [-1, 1]$

2. $SDI > 0$ when $\delta F1_{ora} > \delta F1_{alt}$

3. $SDI = 0$ when $\delta F1_{ora} = \delta F1_{alt}$

4. $SDI < 0$ when $\delta F1_{ora} < \delta F1_{alt}$

The *SDI* score may be seen as the *Mean Signed Deviation*[20] *(MSD)*: $\delta F1_{ora} - \delta F1_{alt}$, normalized with the maximum possible improvement $H$. We don't directly use MSD as $\delta F1 \in [0, \infty)$ makes it unbounded.

For ease of interpretation, we average the *SDI* scores at the level of a dataset, across model sizes, for a given model and oracle. This averaged score is denoted by $\overline{SDI}$.

To measure the second kind of outcomes, we report the percentage of times $\delta F1_{ora} > \delta F1_{alt}$ across these model sizes. This is denoted as *pct_better*.

We consider the oracle-based approach to be a meaningful contribution if $\overline{SDI} > \mathbf{0}$ *and pct_better* > **50%** compared to alternatives.

---

[20]https://en.wikipedia.org/wiki/Mean_signed_deviation

Table 4: LPM, DT compared to Supervised Uncertainty Sampling

| dataset | LPM | | | DT | | |
|---|---|---|---|---|---|---|
| | GBM | RF | **ANY** | GBM | RF | **ANY** |
| cod-rna | 0.60, 100.00% | 0.25, 87.50% | 0.61, 100.00% | 0.29, 60.00% | 0.39, 60.00% | 0.69, 80.00% |
| ijcnn1 | 0.28, 66.67% | 0.17, 66.67% | 0.37, 73.33% | -0.42, 26.67% | 0.32, 80.00% | 0.32, 80.00% |
| higgs | 0.88, 100.00% | 0.23, 60.00% | 0.91, 100.00% | 0.75, 83.33% | 0.28, 66.67% | 0.83, 100.00% |
| covtype.binary | 0.49, 93.33% | 0.33, 93.33% | 0.59, 100.00% | 0.06, 44.44% | 0.18, 45.45% | 0.32, 54.55% |
| phishing | 0.62, 93.33% | 0.44, 93.33% | 0.65, 93.33% | 0.25, 40.00% | -0.27, 13.33% | 0.25, 40.00% |
| a1a | 0.23, 93.33% | 0.32, 93.33% | 0.35, 100.00% | -0.13, 44.44% | 0.49, 91.67% | 0.58, 100.00% |
| pendigits | 0.73, 100.00% | 0.81, 100.00% | 0.83, 100.00% | 0.22, 60.00% | 0.13, 40.00% | 0.32, 60.00% |
| letter | 0.92, 100.00% | 0.95, 100.00% | 0.97, 100.00% | 0.46, 73.33% | 0.04, 46.67% | 0.55, 73.33% |
| Sensorless | 0.63, 100.00% | 0.72, 100.00% | 0.73, 100.00% | 0.63, 73.33% | 0.41, 60.00% | 0.65, 73.33% |
| senseit_aco | 0.15, 53.33% | 0.30, 86.67% | 0.31, 86.67% | 0.39, 71.43% | 0.44, 87.50% | 0.60, 87.50% |
| senseit_sei | 0.07, 53.33% | 0.27, 60.00% | 0.28, 60.00% | -0.18, 37.50% | 0.13, 57.14% | 0.20, 50.00% |
| covtype | 0.83, 100.00% | 0.65, 93.33% | 0.85, 100.00% | 0.60, 73.33% | 0.64, 73.33% | 0.82, 86.67% |
| connect-4 | 0.13, 60.00% | 0.43, 86.67% | 0.50, 100.00% | 0.11, 60.00% | 0.13, 73.33% | 0.53, 86.67% |
| **OVERALL** | 0.50, 85.11% | 0.46, 86.17% | 0.61, 93.09% | 0.23, 57.14% | 0.25, 59.75% | 0.50, 73.75% |

### 4.2.3 Observations

Table 4 and Table 5 compare our approach to Supervised Uncertainty Sampling and the Density Tree based approach, respectively. All $\delta F1_{ora}$ and $\delta F1_{alt}$ scores used are the *average over five runs*. This is the presentation format followed:

1. For each dataset, model and oracle combination we present two scores: (1) $\overline{SDI}$ and (2) *pct_better*.

2. Favorable outcome values - $\overline{SDI} > 0$ or *pct_better* $> 50$ - are colored green, unfavorable outcomes are colored red, and tied values are unformatted.

3. In the case of Supervised Uncertainty Sampling, Table 4, scores are compared across the same oracles, i.e., a score using oracle $GBM$ in our method, is compared to a score from Supervised Uncertainty Sampling using a $GBM$.

4. Unlike supervised uncertainty sampling, there is no notion of an oracle in the Density Tree based approach. In Table 5, for a combination of dataset, model and model size, improved scores from using either the $GBM$ or $RF$ as the oracle are compared to the same reference score from the density tree based approach.

5. We also introduce two special groupings:

   - **ANY**: For each model size, the $SDI$ score considered is the higher of the ones obtained from using the $GBM$ or $RF$ as oracles. The $\overline{SDI}$ and *pct_better* scores are computed based on these scores. This grouping represents the ideal way to use our technique in practice: try multiple oracles and pick the best.
   - **OVERALL**: This averages results across datasets, to provide an aggregate view of the comparison.

   The entries identified by **OVERALL** and **ANY** provide comparison numbers aggregated over datasets, model sizes and oracles.

The predominant amount of values colored green, indicate that our technique performs better in most settings. In both cases, the **OVERALL** +**ANY** entries indicate that our technique works better on average. The *pct_better* scores in these entries also indicate that we seem to do better much more frequently in the case of $LPMs$ than $DTs$.

We note here that the space of sampling distributions modeled by our technique subsume the ones modeled by either competing technique:

Table 5: LPM, DT compared to the Density Tree approach.

| | LPM | | | DT | | |
|---|---|---|---|---|---|---|
| dataset | GBM | RF | **ANY** | GBM | RF | **ANY** |
| cod-rna | -0.38, 0.00% | -0.45, 0.00% | -0.33, 0.00% | 0.51, 60.00% | 0.50, 70.00% | 0.65, 80.00% |
| ijcnn1 | 0.06, 66.67% | 0.11, 80.00% | 0.20, 93.33% | 0.23, 53.33% | 0.68, 100.00% | 0.68, 100.00% |
| higgs | -0.07, 40.00% | -0.07, 40.00% | 0.04, 46.67% | 0.23, 50.00% | 0.61, 83.33% | 0.61, 83.33% |
| covtype.binary | -0.16, 40.00% | -0.33, 13.33% | -0.15, 40.00% | 0.23, 66.67% | 0.26, 72.73% | 0.38, 81.82% |
| phishing | 0.30, 80.00% | 0.37, 86.67% | 0.38, 86.67% | 0.11, 26.67% | -0.00, 26.67% | 0.23, 46.67% |
| a1a | -0.03, 60.00% | 0.13, 66.67% | 0.13, 66.67% | -0.06, 44.44% | 0.43, 75.00% | 0.52, 83.33% |
| pendigits | 0.59, 100.00% | 0.59, 93.33% | 0.62, 100.00% | 0.23, 60.00% | 0.16, 46.67% | 0.25, 60.00% |
| letter | 0.79, 100.00% | 0.81, 100.00% | 0.81, 100.00% | 0.02, 33.33% | -0.34, 13.33% | 0.06, 40.00% |
| Sensorless | 0.64, 100.00% | 0.65, 100.00% | 0.66, 100.00% | -0.23, 20.00% | -0.39, 20.00% | -0.23, 20.00% |
| senseit_aco | 0.55, 100.00% | 0.63, 100.00% | 0.63, 100.00% | 0.50, 85.71% | 0.37, 75.00% | 0.39, 75.00% |
| senseit_sei | 0.61, 100.00% | 0.66, 100.00% | 0.67, 100.00% | -0.25, 42.86% | 0.51, 100.00% | 0.51, 100.00% |
| covtype | 0.20, 80.00% | 0.39, 93.33% | 0.43, 100.00% | 0.26, 66.67% | 0.16, 66.67% | 0.40, 80.00% |
| connect-4 | 0.23, 73.33% | 0.24, 66.67% | 0.38, 86.67% | -0.23, 33.33% | -0.13, 53.33% | 0.08, 66.67% |
| **OVERALL** | 0.28, 75.00% | 0.32, 75.00% | 0.37, 81.38% | 0.10, 47.06% | 0.16, 57.23% | 0.31, 67.30% |

1. Supervised Uncertainty Sampling assumes high uncertainty points are favorable; this may be modeled with an IBMM with appropriate parameters.

2. Density Trees learn distributions that are based on the proximity of instances to class boundaries; since uncertainty values also correlate with distance from class boundaries - a high uncertainty value for an instance indicates it's near a class boundary and vice versa - this too is well within the scope of what an IBMM may represent.

Our hypothesis as to when the competing techniques outperform our technique is that the optimal sampling distribution is easier to discover given their distributional assumptions. For example, if the optimal distribution indeed turns out to be one where instances with high uncertainty are preferred, the Supervised Uncertainty Sampling technique would quickly discover this, while our technique would need to navigate a larger search space to converge to this solution. Our technique would likely do better on such problems with a larger iteration budget or an appropriately defined prior; we leave this analysis for future work.

Both Supervised Uncertainty Sampling and our technique use distributions over uncertainty values. This makes it interesting to contrast them, and is reviewed in Section A.4.

### 4.3 Additional Applications

Viewing our technique purely as a tool to find the optimal distribution for effective learning, we explore some additional interesting applications in this section.

#### 4.3.1 Different Feature Spaces

In our previous experiments, the feature vector representation was identical for the oracle and the interpretable model. This is also what Algorithm 3 implicitly assumes. Here, we consider the possibility of going a step further and using different feature vectors. If $f_{\mathcal{O}}$ and $f_{\mathcal{I}}$ are the feature vector creation functions for the oracle and the interpretable model respectively, and $x_i$ is a "raw data" instance, then:

1. The oracle is trained on instances $f_{\mathcal{O}}(x_i)$, and provides uncertainties $u_{\mathcal{O}}(f_{\mathcal{O}}(x_i))$.

2. The interpretable model is provided with data $f_{\mathcal{I}}(x_i)$, but the uncertainty scores available to it are $u_{\mathcal{O}}(f_{\mathcal{O}}(x_i))$.

The motivation for using different feature spaces is that the combination $(\mathcal{O}, f_{\mathcal{O}})$ may be known to work well together and/or a pre-trained oracle might be available only for this combination.

We illustrate this application with the example of predicting nationalities from surnames of individuals. Our dataset (Rao & McMahan, 2019) contains examples from 18 nationalities: *Arabic, Chinese, Czech, Dutch, English, French, German, Greek, Irish, Italian, Japanese, Korean, Polish, Portuguese, Russian, Scottish, Spanish, Vietnamese*. The representations and models are as follows:

1. The oracle model is a *Gated Recurrent Unit (GRU)* (Cho et al., 2014), that is learned on the sequence of characters in a surname. The GRU is calibrated with *temperature scaling* (Guo et al., 2017).

2. The interpretable model is a DT, where the features are character n-grams, $n \in 1, 2, 3$. The entire training set is initially scanned to construct an n-gram vocabulary, which is then used to create a sparse binary vector per surname - 1s and 0s indicating the presence and absence of an n-gram respectively.

Figure 9 shows a schematic of the setup.

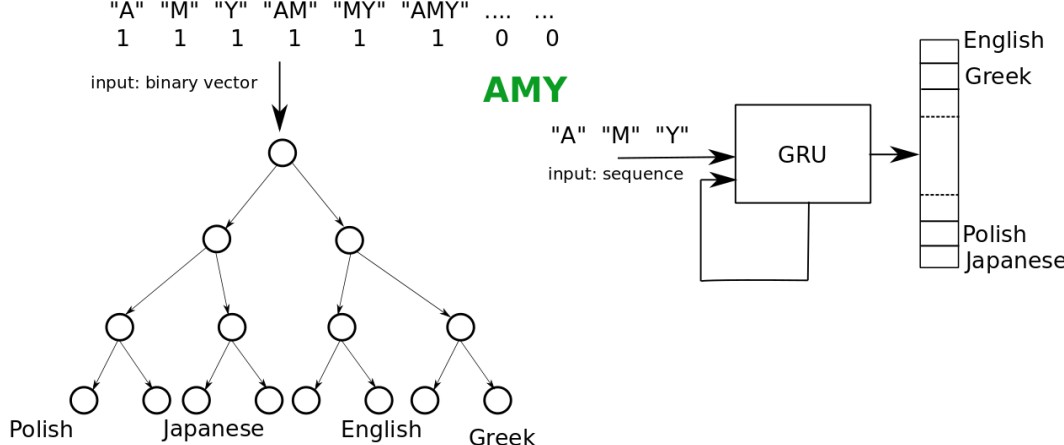

Figure 9: The feature representations for the oracle and the interpretable model may be different. Consider the name "Amy": the GRU is provided its letters, one at a time, in sequence, while the DT is given an n-gram representation of the name.

The n-gram representation leads to a vocabulary of $\sim 5000$ terms, that is reduced to 600 terms based on a $\chi^2$-test in the interest of lower running time (see Section A.13 for details). DTs of different $depth \leq 15$ were trained. A budget of $T = 3000$ iterations was used (the search space for $\Phi$ is the same as in Section 4.1.5), and the relative improvement in the $F1$ macro score (as in Equation 7) is reported, averaged over three runs. Figure 10 shows the results.

We see large improvements at small depths, that peak with $\delta F1 = 83.04\%$ at $depth = 3$, and then again at slightly larger depths, which peak at $depth = 9$ with $\delta F1 = 12.34\%$.

To obtain a qualitative idea of the changes in the DT using a oracle produces, we look at the prediction rules for *Polish* surnames, when DT $depth = 3$. For each rule, we also present examples of true and false positives.

**Baseline rules** - $precision = 2.99\%, recall = 85.71\%, F1 = 5.77\%$:

Rule 1. $k \wedge ski \wedge \neg v$

- True Positives: *jaskolski, rudawski*
- False Positives: *skipper (English), babutski (Russian)*

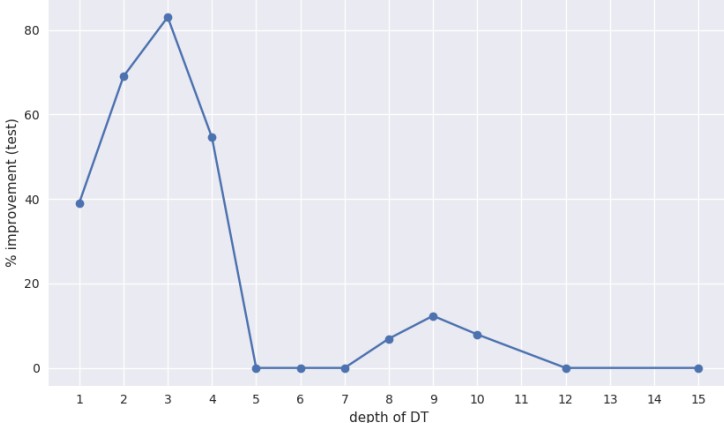

Figure 10: Improvements $\delta F1$ are shown for different depths of the DT.

Rule 2. $k \wedge \neg ski \wedge \neg v$

- True Positives: *wawrzaszek, koziol*
- False Positives: *konda (Japanese), jagujinsky (Russian)*

**Oracle-based DT rules** - $precision = 25.00\%, recall = 21.43\%, F1 = 23.08\%$:

Rule 1. $ski \wedge \neg(b \vee kin)$

- True Positives: *jaskolski, rudawski*
- False Positives: *skipper (English), aivazovski (Russian)*

We note that the baseline rules are in conflict w.r.t. the literal "ski", and taken together, they simplify to $k \wedge \neg v$. This makes them extremely permissive, especially *Rule 2*, which requires the literal "k" while needing "ski" and "v" to be absent. Not surprisingly, these rules have high recall (= 85.71%) but poor precision (= 2.99%), leading to $F1 = 5.77\%$.

In the case of the oracle-based DT, now we have only one rule, that requires the atypical trigram "ski". This improves precision (= 25%), trading off recall (= 21.43%), for a significantly improved $F1 = 23.08\%$.

The difference in rules may also be visualized by comparing the distribution of nationalities represented in their false positives, as in Figure 11. We see that the baseline DT rules, especially *Rule 2*, predict many nationalities, but in the case of the DT learned using the oracle, the model confusion is concentrated around *Russian* names, which is reasonable given the shared *Slavic* origin of many *Polish* and *Russian* names.

We believe this is a particularly powerful and exciting application of our technique, and opens up a wide range of possibilities for translating information between models of varied capabilities.

### 4.3.2 Size-Constrained Training Sample

Recall from Section 3.4, we make use of a parameter $N_s$, denoting sample size, that we had constrained to $\in [400, 10000]$ (Section 4.1.5) in our experiments. But it is possible to set this to much smaller values to study the sampling distribution for patterns, significance of regions in the input space, etc. Figure 12 shows an example of this: we set $N_s \in [50, 50]$ (so it can take exactly one value, 50), and for the dataset shown in Figure 12(a), we visualize the sampling distribution when the model is a DT of $depth = 2$ in Figure 12(b) vs when $depth = 4$ in Figure 12(c). The dataset is balanced, and the oracle used is a GBM.

We see the following interesting patterns: (a) at $depth = 2$, the DT picks points from both regions where $label = 1$, but the larger region shows higher density. This is possibly because owing to its limited capacity,

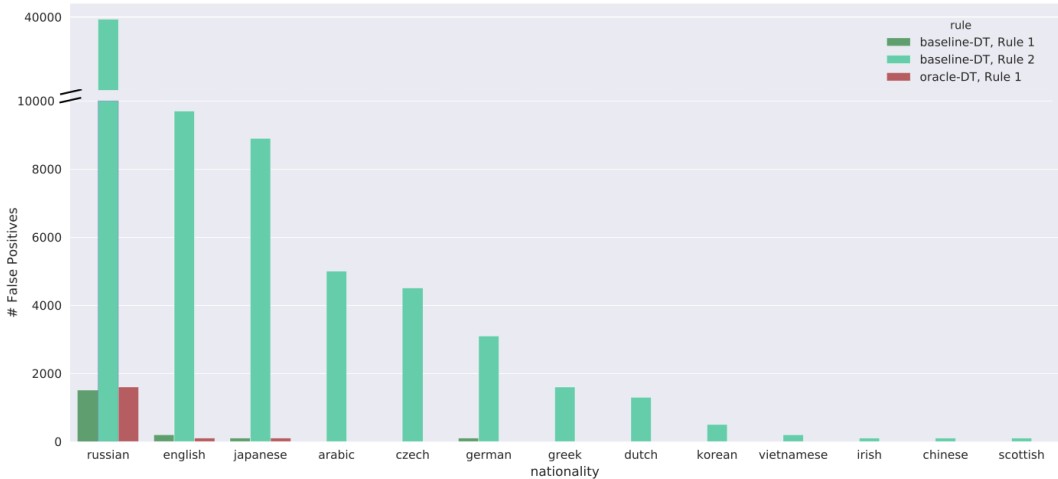

Figure 11: The distribution of nationalities in false positive predictions for the baseline and oracle based models, shown for predicting *Polish* names. Only nationalities with non-zero counts are shown.

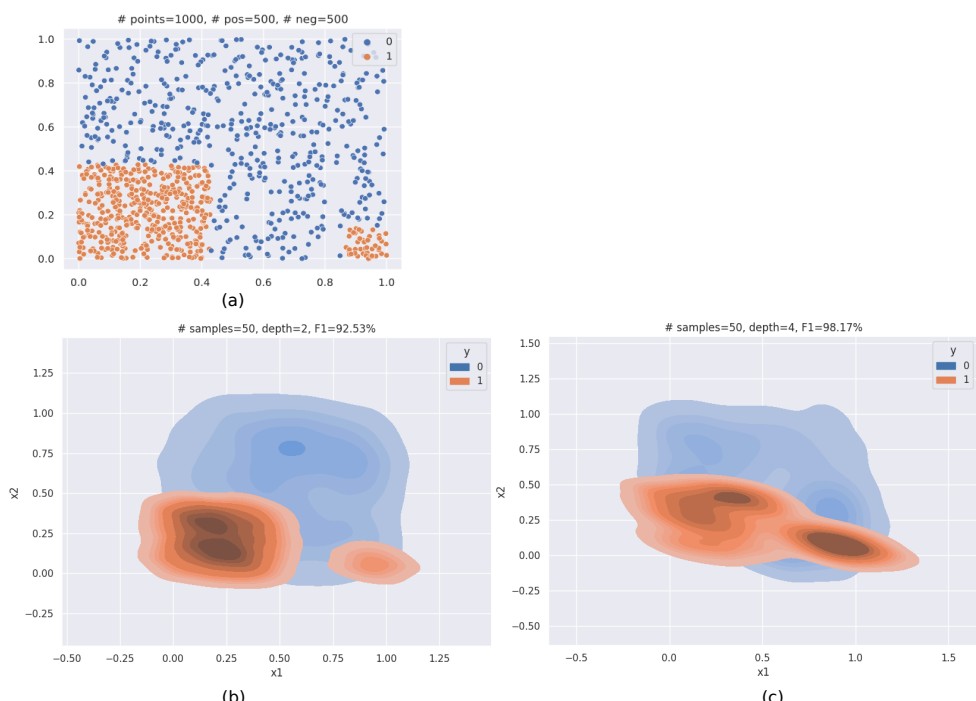

Figure 12: Our technique might be used to identify the optimal sample of a given size. (a) shows the original dataset. (b) and (c) visualize the learned distribution of points, using a KDE, for DTs with $depth = 2$ and $depth = 4$ respectively, for a sample size of 50. NOTE: the connection shown in (c), between the two originally disjoint regions with $label = 1$, is an artifact of the KDE.

the model is able to effectively parameterize only one region, and therefore it prioritizes correct classification of points around the larger region, (b) at $depth = 4$, we see increased sampling density in the smaller region with $label = 1$ as well.

### 4.3.3 Vector Model Size

Although we have been using a scalar notion of model size - depth for DT, number of terms for LPM, number of trees for a GBM - Algorithm 3 doesn't restrict us from using a vector-valued model size $\eta$. For example, in the case of GBMs, we may consider the notion of model size $\eta = [max\_depth, num\_trees]$, where the quantities respectively denote the maximum depth allowed for each constituent DT in a GBM, and the number of DTs in the GBM. In Figure 13 we show how improvements for GBMs vary when $1 \leq max\_depth \leq 5$ ($x$-axis) and $1 \leq num\_trees \leq 5$ ($y$-axis); the oracle used is a GBM as well (unconstrained in size), and results for these datasets are shown: (a) `higgs` (b) `cod-rna` (c) `senseit-sei` and (d) `senseit-aco`. The improvements are averaged over three runs. We observe the familiar pattern that as model sizes increase, in terms of both $max\_depth$ and $num\_trees$, improvements decrease.

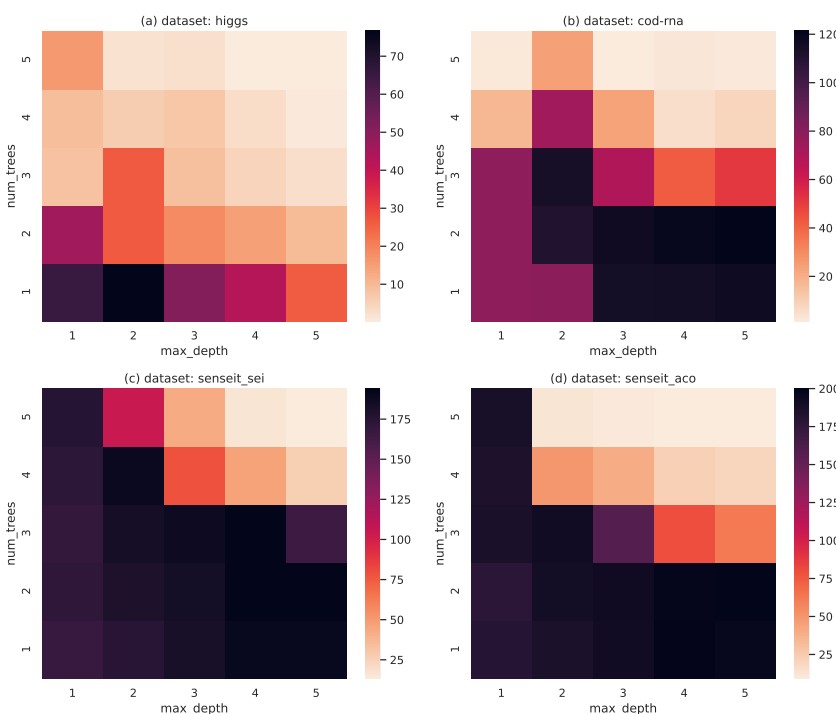

Figure 13: Improvements in test $F1$-macro for multiple datasets for different sizes of $GBM$ models are shown. Here, model size is the combination of $max\_depth$ and $number\ of\ trees$ in the $GBM$ model. Greater improvements are seen at lower sizes.

### 4.4 Summary

We summarize our observations from our experiments here:

1. For all combinations of interpretable and oracle models - $\{LPM, DT\} \times \{GBM, RF\}$ - we see good improvements, $\delta F1$, especially at small sizes (Section 4.1.6). Sometimes these may be $> 100\%$. For model sizes beyond a point, we observe $\delta F1 \approx 0$.

2. The results in Section 4.2 strongly indicate that the precise relationship of the sampling distribution and the uncertainty needs to be learned, and a heuristic strategy of exclusively sampling high

uncertainty points is not optimal. We believe this is an important result, especially given that this is true for the supervised version of uncertainty sampling, which is significantly more powerful than standard uncertainty sampling.

3. Our approach produces better accuracy, in general, compared to both supervised uncertainty sampling and the density tree based approach.

   The results in Table 4 and Table 5 from Section 4.2.3 are summarized in Table 6. Recall that the combination **OVERALL + ANY** averages over datasets and oracles; Table 6 shows these summary statistics.

Table 6: Summary comparison results, **OVERALL + ANY**

| compared to | model | $\overline{SDI}$ | pct_better |
|---|---|---|---|
| supervised uncertainty sampling | LPM | 0.61 | 93.09% |
| | DT | 0.50 | 73.75% |
| density trees | LPM | 0.37 | 81.38% |
| | DT | 0.31 | 67.30% |

We observe that density trees are more competitive to our technique than supervised uncertainty sampling: smaller $\overline{SDI}$ and *pct_better* scores. This is to be expected since the density tree based approach is capable of learning flexible distributions over the input space.

4. A remarkable fact of practical value is that we *don't tune the parameters* $\Phi$ for a specific problem. The value ranges for these are fixed across tasks, with only the iteration budget $T$ being changed - as described in Section 4.1.5. This highlights another strength of the technique: $\Phi$ need not be tuned for obtaining meaningful improvements, as long as it admits a broad enough set of uncertainty distributions.

5. Section 4.3 showcases the generality of the proposed technique: we successfully used it with differing feature spaces across the oracle and the interpretable model, to identify the optimal training sample for a given size, and with vector valued model sizes. These applications considerably broaden the impact of our work.

**Importantly**, the various positive results from this section should be seen as representative of the proposed *framework*, and not just our *implementation*. In other words, these results establish a lower bound for the outcomes, because they may be potentially improved by using different components within the larger framework, e.g., by using a different optimizer from among the ones discussed in Feurer & Hutter (2019) or Turner et al. (2021). We show examples in Section A.8.

## 5 Discussion

Having looked at both the theory and empirical outcomes, we revisit a few points of interest in this section.

1. **Effect of flattening**: We first consider the question: does flattening (Section 3.6) help? Table 7 contrasts *improved* $F1$ scores obtained without (rows denoted as "original") and with (denoted "flattened") flattening the uncertainty distribution. This is shown for the datasets `Sensorless` and `covtype.binary`, for model $size \in \{1, 2, 3\}$, with $model = LPM$ and $oracle = GBM$. Two different parameter settings are used: (a) Setting 1 is what we have used in the experiments in Section 4: maximum allowed $Beta$ components are 500 and $scale = 10000$ (b) Setting 2 looks at much lower values of these parameters where maximum allowed components is 50 and $scale = 10$. The scores presented are the average over three trials.

   We observe that while flattening influences results, other parameters determine the magnitude of its effect. At Setting 1, `Sensorless` is affected at $size = 1$ (flattening is better), but at higher sizes the

Table 7: Improved scores averaged over three trials, shown for different parameter settings, with and without flattening. Here, Setting 1 is $\{max\_components = 500, scale = 10000\}$ and Setting 2 is $\{max\_components = 50, scale = 10\}$. "curr." signifies this is the current setting for our experiments in Section 4, while "low" signifies lower values of parameters. Highlighted cells indicate positve effect of flattening.

| dataset | dist. | Setting 1 (curr.) | | | Setting 2 (low) | | |
|---|---|---|---|---|---|---|---|
| | | 1 | 2 | 3 | 1 | 2 | 3 |
| Sensorless | original | 0.39 | 0.54 | 0.57 | 0.38 | 0.42 | 0.41 |
| | flattened | 0.44 | 0.53 | 0.55 | 0.43 | 0.54 | 0.59 |
| covtype.binary | original | 0.66 | 0.69 | 0.71 | 0.64 | 0.66 | 0.71 |
| | flattened | 0.68 | 0.73 | 0.73 | 0.65 | 0.71 | 0.71 |

differences seem to be from random variations across trials. At Setting 2 however, the differences are seen for $size \in \{1, 2, 3\}$ (flattening is better). For `covtype.binary` only $size = 2$ seems to be affected in either setting.

Recall we had noted in Figure 2 that the datasets `Sensorless` and `covtype.binary` have non-smooth and smooth uncertainty distributions respectively. The observations in Table 7 align well with the expectation that `Sensorless` is positively affected by the transformation, while results for `covtype.binary` remain mostly unchanged.

Based on these tests, we hypothesize that for non-smooth uncertainty distributions, flattening makes our technique robust across parameter settings. It does not affect smooth distributions in a significant way. Of course, rigorous and extensive tests are required to conclusively establish this effect.

2. **Measuring compaction**: We mentioned in the Introduction, Section 1, that a possible area of application of this work might be model compression. We would like to point out that the *compaction profile* (Figure 5, Figure 24) plots emphasize this use-case: they're a visual tool to determine the minimal model size achievable using our technique, given a baseline model size.

   To formalize this connection, we introduce the score *Compaction Index (CI)* that denotes the extent of model size decrease possible, up to a size where $\delta F1 \approx 0$. Figure 14 shows a sample compaction profile. The $CI$ score, where $CI \in [0, 1]$, is the ratio of the area in red to the area in green.

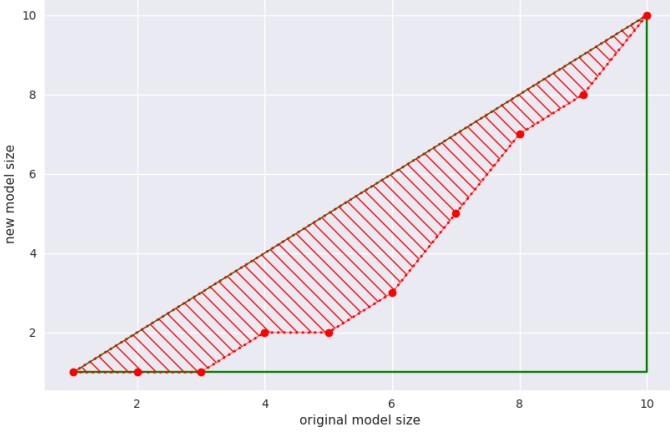

Figure 14: Compaction Index

The more reduction in model size our technique can obtain, the closer the red curve is to the green boundary, and $CI \approx 1$. If no reduction is possible at any model size, the red line coincides with the diagonal and $CI = 0$. Clearly, this score is specific to a model family $\mathcal{F}$, a training algorithm $f$ and

a specific notion of model size. And ideally, this should be averaged over all possible datasets and oracles.

Here are the $CI$ scores for our experiments :

- $LPM : CI = 0.57$
- $DT : CI = 0.17$

These scores indicate that $LPMs$ may be compacted better than $DTs$, for the respective notions of size we use here - this may also be seen from the plots in Figure 3, where the improvements for $DTs$ decrease faster, with growing model size, than those for $LPMs$.

3. **Upper bound of improvements**: In Equation 1, and then in Equations 2 and 3, the improved accuracy of the interpretable model is shown bounded by the oracle accuracy. For example, see the rightmost term in Equation 1, reproduced below:

$$accuracy(train_{\mathcal{I},g}(p,\eta),p) \lesssim accuracy(train_{\mathcal{I},g}(q,\eta),p) \lesssim accuracy(train_{\mathcal{O},h}(p,*),p) \qquad (14)$$

We empirically show this to be true now. In Figure 15, we show the distribution of relative difference between the improved accuracy of a $LPM$ model and the accuracy of a $GBM$ oracle.

Using the notation in the equation above, we calculate the relative difference $\Delta F1$ as:

$$\Delta F1 = \frac{accuracy(train_{\mathcal{I},g}(q,\eta),p) - accuracy(train_{\mathcal{O},h}(p,*),p)}{accuracy(train_{\mathcal{O},h}(p,*),p)} \qquad (15)$$

Here, of course, we measure accuracy using the $F1$ macro score.

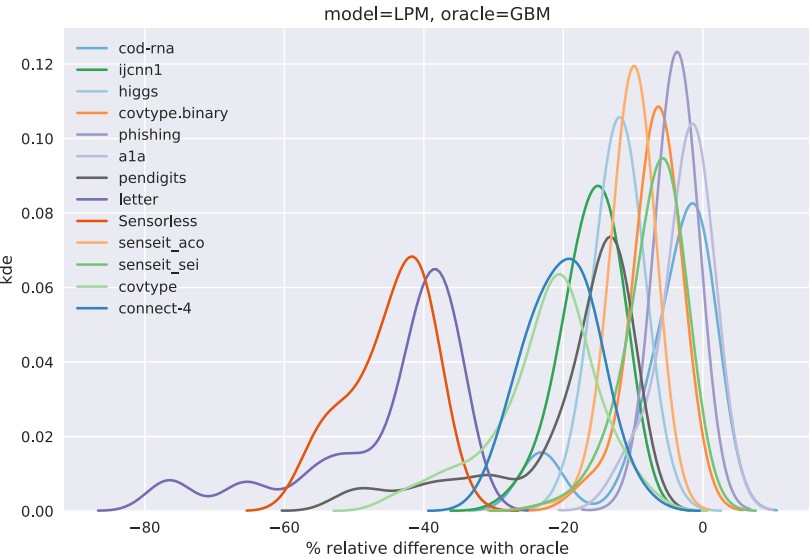

Figure 15: Distribution of %age accuracy difference from the oracle accuracy.

There is one distribution plotted per dataset, where the distribution uses information from multiple runs, for multiple model sizes. It may be seen in Figure 15 that all relative differences are at most 0 (there is some spillover to the right of 0 owing to the use of KDEs for visualization).

For precise numbers, we look at Table 8, which lists the %age of cases where the interpretable model's accuracy exceeded that of the oracle, and the average value of the relative difference for the cases where it is positive. We also consider the statistical significance of these values: using a one-sided *Wilcoxon signed-rank test*, we determine the *p-value* for the null hypothesis that accuracy scores produced our technique exceed those of the oracle. This is a *paired* test where oracle and

model scores for a dataset are paired, for a given combination of an interpretable model and an oracle. Only the largest model built for the dataset is used.

For instance, for dataset `a1a`, we note from Table 2 that using the GBM oracle, LPMs of sizes $1 \ldots 15$ are constructed; however, this test uses *only* the LPM of size 15. We don't use all sizes since that biases results in our favor, as smaller models are not expected to match the oracle's accuracy.

See Figure 26 in Section A.12 for plots for other model-oracle combinations.

Table 8: The percentage of cases where we see positive relative difference w.r.t. oracle, and the mean of these positive difference are shown. The *p-value* for the hypothesis that model improvements exceed oracle are also provided.

| model | oracle | %age +ve cases | mean +ve value | *p-value* (largest models) |
|-------|--------|----------------|----------------|----------------------------|
| LPM | GBM | 1.60% | 0.81 | 0.000737 |
| LPM | RF | 4.47% | 1.36 | 0.001183 |
| DT | GBM | 3.51% | 1.84 | 0.001488 |
| DT | RF | 2.85% | 0.65 | 0.000936 |

## 6 Future Work

The results from our experiments suggest multiple promising avenues for future research. We list some of them here:

1. We might think of our technique as learning a sampling distribution $p'(x_i)$ indirectly on the input space: via $p(u_{M_O}(x_i))$, the distribution over uncertainty values produced by the oracle. An alternative view might be to directly learn instance weights $w_i$ instead, where $w_i = p'(x_i)$. This approach clearly suffers from challenges in scaling - there are as many weights as training instances. From this perspective, the combination of the oracle and mixture model may be seen as providing a computationally cheap means to indirectly learn such weights.

   However, recent work suggests that for *differentiable* model losses, this problem might be efficiently solved by formulating it as a *bi-level optimization* problem (Pedregosa, 2016; Lorraine et al., 2020); which makes this a feasible direction to explore. The expected benefit is this might be faster[21], at least for moderately sized datasets.

   This approach brings its own challenges that future work would need to consider: (a) since gradient information is required, the loss function must be known; therefore this approach is not model-agnostic (b) even if an automatic differentiation framework is used, such as *JAX* (Bradbury et al., 2018), to generalize to unseen loss functions, model families like DTs remain out of scope since their loss isn't differentiable.

   **NOTE** that this aspect distinguishes our method from KD: beyond the uncertainty scores from the oracle, we do not require any additional information, e.g., "dark knowledge" (Hinton et al., 2015b; Korattikara et al., 2015).

2. On the lines of learning a training distribution, alternative representations like the *Pitman-Yor process* (Pitman & Yor, 1997) or *Determinantal Point Processes* (Kulesza & Taskar, 2012) may be explored.

3. Algorithm 3 uses multiple trials at various steps for statistical robustness: for fitting $M_t$ and for computing $s_t$. This obviously impacts the running time of Algorithm 3, and leads to an obvious

---

[21]Pedregosa (2016) compares this approach against BO for the task of hyperparameter tuning; these numbers do not directly translate here since the BO algorithm used is not TPE.

question: are these necessary? An interesting path for exploration here is to eliminate them and instead shift the burden to the BO, in the form of accounting for *noise*. This may be done at the level of the surrogate model, e.g., Gaussian Processes may be parameterized to handle noise (Rasmussen & Williams, 2005) or at the level of the acquisition function, e.g., *Noisy Expected Improvement (NEI)* (Frazier, 2018; Letham et al., 2019). If such a strategy doesn't end up compromising accuracies much, it would make the execution **nine times faster** (3 trials for $s_t \times 3$ validation trials for $M_t$ per iteration).

4. The standard way to evaluate active learning algorithms is to evaluate model accuracy against the number of labeled training data instances. It is interesting to consider an alternative approach: for a given budget of labeled instances, measure the divergence between the sampling distribution our method learns (as in Section 4.3.2) and the one that an active learner proposes for labeling. Such an analysis is insightful since it can indicate precisely which points an active learner is *supposed* to label.

5. We noted that improvements from our technique diminish as model size grows (Section 4.1.6). For larger model sizes, a possible direction to explore might be to "chain" together multiple small models. This is similar to gradient boosting, and it would be especially informative to compare the two approaches.

6. An obvious question to ask is if our observations around the impact of training distribution on accuracy may be theoretically explained. There is some recent work in the area of KD that might serve as fruitful starting points: (a) Dao et al. (2021) provide theoretical tools to analyze distillation by treating it as a semi-parameteric inference problem (b) Menon et al. (2021) propose a connection between the effectiveness of a teacher and its ability to approximate Bayes class-probabilities. (c) study of sample re-weighting on the effectiveness of distillation (Zhang et al., 2021; Lu et al., 2021).

7. It would also be interesting to explore the connection between the sample of a given size our method finds (Section 4.3.2) and the *data Shapley value* (Ghorbani & Zou, 2019): a per-instance value quantifying the contribution of an instance to predictor accuracy. Some questions of interest are: (a) does an instance that has a high sampling probability across a range of sample sizes, per our method, also receive a high data Shapley value? (b) if there is indeed a correspondence between the two techniques, what algorithmic ideas may be borrowed from one technique to another?

## 7 Conclusion

In this paper we introduced a novel technique to learn an interpretable model, that reduces the trade-off between model size and accuracy. The practical implication of this work is that instead of picking an interpretable model family based on accuracy, one may use our method to construct accurate models for their preferred model family.

Producing an accurate model is formulated as an optimization problem of identifying training data that maximizes learning. This process is aided by an oracle model. Our technique is shown to possess multiple favorable properties: (a) the optimization uses a fixed set of seven variables, irrespective of the dimensionality of the data (b) a reasonable choice of the search space produces good results across datasets (c) the technique is model-agnostic wrt both the interpretable and oracle models (d) it may be used even when the feature spaces of the interpretable model and oracle are different (e) its a framework, which leaves open the possibility of improving upon it. We have also shown some additional interesting applications for our technique.

We have provided extensive empirical validation to establish the utility of the technique.

This work also presents some intriguing deeper findings : (a) train and test distributions need not be identical for optimal learning (b) our observations point to a "small model effect": this difference in distributions exists for small model sizes, and it is in this model size regime that we observe most improvements.

We believe that the general theme of the proposed technique, that of shaping data density to influence accuracy, as well as the deeper results, offer promising directions for future research.

# Appendices

## A  Appendix A

### A.1  Related Work

This is an extended version in the discussion presented in Section 2.4. We compare and contrast with various techniques that have some similarity with ours:

1. **Density Tree based sampling**: Ghose & Ravindran (2020) addresses a similar problem of enhancing the accuracy of a size-constrained model. The approach taken was to develop a specific form of decision tree, known as a *density tree*, to capture neighborhood information, and sample from its various nodes to obtain an optimal training distribution. Our current method may be considered a significant evolution of the approach, as it adds the following flexibilities:

   (a) Their choice of the oracle was restricted to a forest of density trees[22]. Here, we might use an oracle from an arbitrary model family - which makes it possible to have an oracle that is more accurate than density trees.
   This also has the practical benefit that the oracle need not be learned from scratch: if there is already a pre-trained probabilistic model like a *deep neural network* available for a dataset, it may be conveniently plugged into our algorithm as-is, to improve the accuracy of an interpretable model.

   (b) The density trees and the interpretable model had to be constructed on the same (or very similar) feature space. Here, this is not required, and the oracle might be a sequence model that classifies text, while the interpretable model may be a classifier that uses character n-grams. This considerably broadens the scope of our technique. We look at an example in Section 4.3.1.

2. **Knowledge Distillation (KD)**: KD looks at using powerful "teacher" models (similar to our oracle) to learn a smaller "student" model (Gou et al., 2021). The key differences with KD are:

   (a) Unlike KD our goal is not to approximate the oracle's performance. In fact, we ignore the oracle's label assignments entirely. This is in contrast to KD methods that may use teacher-assigned labels (Bucilă et al., 2006) or distribution of label confidences (Hinton et al., 2015a)[23], or in general, focus on extracting "dark knowledge" from the oracle in some form. Instead, our goal is to evolve the smaller model towards a more accurate version.

   (b) A lot of KD techniques are specialized for use with Neural Networks, e.g., *FitNets* (Romero et al., 2015), *DistilBERT* (Sanh et al., 2019). In contrast, our technique is model-agnostic.

   (c) Methodological differences aside, our observations suggest that the oracle might only be required to make our algorithm computationally efficient (discussed in Section 6, "Future Work").

3. **Explainable AI (XAI)**: Although we use an oracle model, the goal is not to explain its predictions. Consequently, we place much weaker demands of the information shared by the oracle than in XAI algorithms such as *LIME* (Ribeiro et al., 2016), *SHAP* (Lundberg & Lee, 2017), *TREPAN* (Craven & Shavlik, 1995) or *GlocalX* (Setzu et al., 2021):

   (a) A fundamental difference is we don't request for the predicted labels of the oracle.

   (b) As an extension of the previous point, we ignore *fidelity* to the oracle's predictions, which is explicitly enforced and measured in XAI techniques. This precludes any deliberate alignment between the logical process the oracle follows to classify an instance vs that followed by the interpretable model.

---

[22]Ghose & Ravindran (2020) doesn't refer to density trees as oracles, but they play a role similar to the oracle here.
[23]While we use the uncertainty in the oracle's prediction, note that we don't know which labels is the oracle more or less uncertain about, i.e., we ignore label *identity*.

(c) The information from the oracle is a one-time input to the algorithm, and is restricted to the training instances available to the interpretable model[24]. This is different from queries on synthetic instances performed by various XAI techniques to determine the neighborhood of a data instance, e.g., perturbed instances in case of *GLocalX*[25], *LIME*, *SHAP*, or generated random instances within constrained regions of the input space in *TREPAN*.

In short, we do not infer explanations, local or global, from the oracle. It is used to provide a one-dimensional view of the training data, as a one-time input to our algorithm; this is not equivalent to explaining the oracle. As mentioned earlier, this work suggests that the oracle might be only needed to provide reasonable computational complexity (see Section 6, "Future Work").

4. **Active Learning**: In the case of active learning too, a predictive model maybe learned on a distribution $q(X, Y)$ that is different from the test distribution $p(X, Y)$. However, some significant differences are:

   (a) Active learning works in the setting where only some or none of the labels of the training data are initially known, and there is an explicit label acquisition cost. We work within the traditional supervised setting where labels of all training instances are known.

   (b) The goal of an active learner is to minimize the total label acquisition cost, while being as accurate as a supervised learner that has access to complete label information. This is very different from our goal of performing *better* than a supervised learner, especially when the model size is small, assuming complete label information.

   It must be noted that the term "oracle" in the active learning literature might refer to either a model or a human labeler; in our work, it exclusively refers to a model.

5. **Transfer Learning**: Transfer learning studies informing the training process of a "target" learner, given a "source" learner (Torrey & Shavlik, 2009; Pan & Yang, 2010; Weiss et al., 2016). Our technique is ostensibly similar as we have an oracle (our source learner) informing the interpretable model (our target learner). However, here are some key differences:

   (a) The typical application of transfer learning is in settings where the source learner has access to more data than the model it must transfer knowledge to; here transfer learning is seen as a way to overcome the data shortage by directly having the source learner convey knowledge, in some form, to the target model. This is different from our setting where the same data is available to both the oracle and the interpretable model.

   (b) Transfer learning techniques usually make some assumptions about the model family. Some examples are Boolean concepts (Thrun & Mitchell, 1994), Markov Logic Networks (Mihalkova & Mooney, 2006) or task-specific neural networks like *BERT* (Devlin et al., 2019) or *ULMFiT* (Howard & Ruder, 2018) for Natural Language Processing, and *VGG networks* (Simonyan & Zisserman, 2015) for image recognition. In comparison, our technique is model agnostic, both w.r.t. the oracle and the interpretable model.

   (c) Although instance re-weighting techniques have been investigated as a means of transfer learning[26], their objective is to perform effective learning in situations where the data distribution available in the source task/domain is different from that in the target task/domain (Liao et al., 2005; Dai et al., 2007; Kamishima et al., 2009). In our case, these two distributions, as provided, are identical; we *choose* to use a different training distribution in the interest of improving accuracy.

In short, the limited use of the oracle sets us apart from KD and XAI, and the identical supervised setting available to both the oracle and the interpretable models differentiates us from Active and Transfer learning.

---

**Algorithm 4:** Supervised Uncertainty Sampling

---

**Data:** Dataset $D$, model size $\eta$, $train_{\mathcal{O},h}()$, $train_{\mathcal{I},g}()$, batch size $b$

**Result:** Test set accuracy $s_{test}$, and interpretable model $M^*$

**1** Create stratified splits $D_{train}, D_{val}, D_{test}$ from $D$

**2** $M_O \leftarrow train_{\mathcal{O},h}(D_{train}, *)$

**3** $I_{remaining} \leftarrow \{1, 2, ..., |D_{train}|\}$ be an index set of $D_{train}$

**4** $I_{current} \leftarrow \{\}$

**5 for** $t \leftarrow 1$ **to** $\lceil |D_{train}|/b \rceil$ **do**

**6**     $I_U \leftarrow$ set of top $b$ entries from $I_{remaining}$, based on $u_{M_{\mathcal{O}}}(x_i), i \in I_{remaining}$

**7**     $I_{remaining} \leftarrow I_{remaining} - I_U$

**8**     $I_{current} \leftarrow I_{current} \cup I_U$

**9**     $D_t \leftarrow \{D_{train,i} | i \in I_{current}\}$

**10**     $M_t \leftarrow train_{\mathcal{I},g}(D_t, \eta)$

**11**     $s_t \leftarrow accuracy(M_t, D_{val})$

**12 end**

**13** $t^* \leftarrow \arg\max_t \{s_1, s_2, ..., s_{T-1}, s_T\}$

**14** $M^* \leftarrow M_{t^*}$

**15** $s_{test} \leftarrow accuracy(M^*, D_{test})$

**16 return** $s_{test}, M^*$

---

### A.2  Supervised Uncertainty Sampling

In Algorithm 4:

1. The loop in lines 5-11 runs $\lceil |D_{train}|/b \rceil$ times, where every iteration adds the $b$ most uncertain points to the current training dataset $D_t$. If $b$ doesn't evenly divide $|D_{train}|$, the last iteration picks all remaining points.

2. In our implementation, $u_{M_{\mathcal{O}}}(x_i)$ in line 6 is precomputed and stored as a lookup table to reduce execution time.

3. In our experiments, we use a batch size $b = 10$. Note that this gives us optimal models as per Algorithm 4, for all batch sizes of the form $10k$, where $k \in \{1, 2, ..., \lfloor |D_{train}|/10 \rfloor\}$

The modified algorithm is a **significantly** more powerful version compared to the ones typically used in Active Learning setups, due to the following reasons:

1. We do not assume a cost for procuring or applying the oracle, which contrasts with the typical active learning setup. Thus, our oracle utilizes complete label information and our model has access to reliable uncertainty scores; this avoids the sample bias discussed in Section A.3 (visualized in Figure 16).

2. Since we have complete label information, we have a validation set $D_{val}$ available to us. In active learning, a validation set would be created from within the current labelled subset of data, which often makes it statistically insignificant or non-representative of the true distribution, especially at early iterations.

3. We do not have to estimate how many times the loop in lines 5-11 must run - this is executed till all data from $D_{train}$ has been used up to train the model. Estimating the number of iterations is required when performing active learning since every iteration incurs a cost - that of calling the

---

[24]The oracle only sees the training data available to the interpretable model, and is required to provide uncertainty values for these instances.

[25]GlocalX indirectly does this via the local learner it uses, *LORE* (Guidotti et al., 2019).

[26]We specifically mention this since instance re-weighting maybe seen as a form of sampling.

oracle to compute $I_U$. Consequently, here, we have the liberty of being able to *pick* the best model based on a validation set $D_{val}$.

### A.3 Simple Uncertainty Sampling in Active Learning

In active learning, the goal is to learn a model when we are given none or few of the labels of our training data, but we are allowed to query for labels for a cost (Settles, 2009). This is helpful in scenarios where acquiring labels is expensive, and instead of asking for labels for a random 1000 points to train on, we could ask for the labels of a specific 200 points, chosen in some manner, that leads to comparable model accuracy. *Uncertainty Sampling* was introduced in Lewis & Gale (1994) to solve this problem. We begin by requesting the labels of small batch of randomly sampled points - this is the labelled subset of the data. The following steps are then repeated:

1. Construct a classifier on the current labelled subset.

2. Use it to provide uncertainty scores for unlabelled points in the data, and then request labels for the top $b$ (the precise value of $b$ may be task specific) uncertain points. These now become part of the labelled subset.

Although intuitive, this approach was shown to suffer from sample bias (Dasgupta & Hsu, 2008; Dasgupta, 2011). We illustrate this in Figure 16.

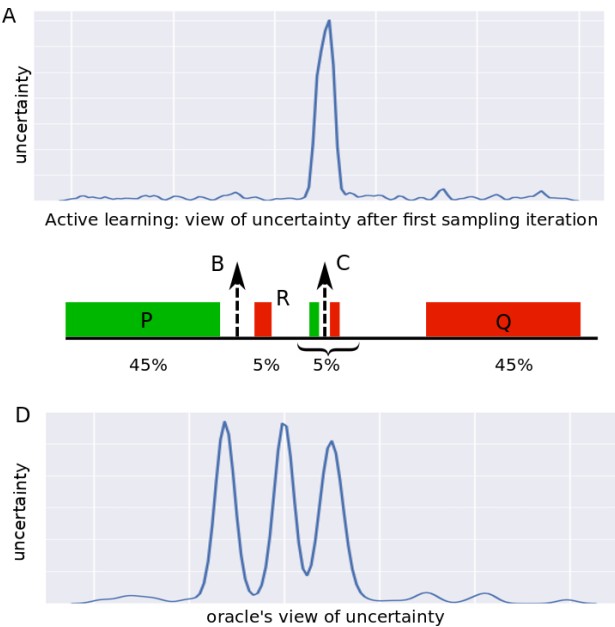

Figure 16: Uncertainty estimates from classifier after first iteration. Smaller boundaries are missed since the sample predominantly comes from $P$ and $Q$.

We consider the simple case where our data is located on a line, has two labels (denoted by red and green in the figure) and most of the data is located at the extremes of the line segment, as shown by blocks $P$ and $Q$, each of which represent 45% of the overall data. Here, learning a classifier is equivalent to identifying a single point on the line, and the classification rule is we assign labels green and red, to left and right of this point, respectively. $B$ and $C$ show two possible classifiers.

In the active learning setup, we observe only the points but not their labels. To use uncertainty sampling, we pick our first small batch of points randomly and query their labels. Because of the distribution of the data, its highly likely that we would only see points from $P$ and $Q$. The best classifier on this sample is $C$, which is midway between $P$ and $Q$. Plot $A$ shows what the uncertainty across the input space looks like according to

$C$. In the next iteration, we will sample close to $C$, since that's where the highest uncertainties are, and the new classifier constructed would again be at location $C$. Subsequent iterations would further reinforce the belief that $C$ is the only class boundary. Here, the classification error of $C$ is 5%, but the optimal classifier is $B$, with an error of 2.5%, which uncertainty sampling fails to discover. The key problem here is we may never see some boundaries, like those defined by $R$, because of the combination of initial sample bias and subsequent aggressive sampling.

This problem does not affect us since the oracle has access to the complete training data. Plot D shows the uncertainty distribution as per the oracle. However, as our results show, even with its complete view of uncertainty landscape, simple uncertainty sampling is not optimal.

### A.4   Comparison of Uncertainty Distributions

It is instructive to look at some specific adjusted IBMMs in the context of the relative performance of techniques. Figure 17 shows the plots from Figure 21 annotated with $SDI$ scores. These are for $LPMs$ using $GBM$ as the oracle.

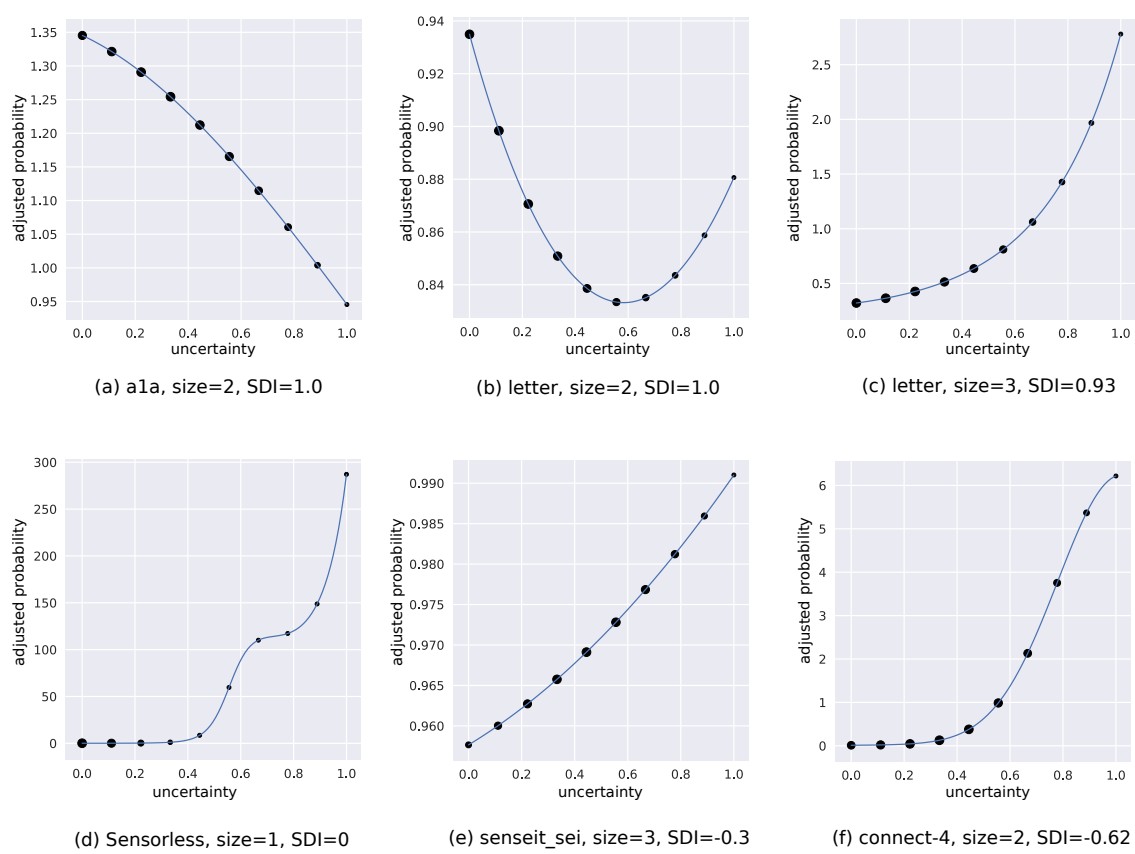

Figure 17: Examples of adjusted distributions are shown, and the $SDI$ scores, measured against supervised uncertainty sampling, are mentioned. The plots in the top row are the same as in Figure 21. The top row - (a), (b), (c) - shows instances where our technique performed relatively better, and the bottom row shows cases where uncertainty sampling was competitive - (d) - or better - (e), (f).

The top row - (a), (b), (c) in Figure 17 - shows instances where our technique did much better ($SDI >$ 0); it would seem that these are cases where sampling exclusively at high uncertainties is not an optimal distribution. Figure 17(d) shows a case where the optimal distribution is composed exclusively of high uncertainty points - so its not surprising that uncertainty sampling is at par with our technique ($SDI = 0$). (e) and (f) show similar trends.

While these plots are helpful in developing intuition for the underlying process, we would like to add the caveat that they are not conclusive in isolation. An example of this is (c) - it is not clear why uncertainty sampling does so poorly here. Possibly, instances with low uncertainties need to be sampled in a very specific manner that cannot be approximated by selecting the top $n$ uncertain points, for any $n$.

## A.5 Uncertainty Metrics

Some other popular uncertainty metrics are:

1. **Least confident**: we calculate the extent of uncertainty w.r.t. the class we are most confident about:
$$u_M(x) = 1 - \max_{y_i \in \{1,2,...,C\}} M(y_i|x) \tag{16}$$

   Here, we have $C$ classes, and $M(y_i|x)$ is the probability score produced by the model[27].

2. **Entropy**: this is the standard Shannon entropy measure calculated over class prediction confidences:
$$u_M(x) = \sum_{y_i \in \{1,2,...,C\}} -M(y_i|x) \log M(y_i|x) \tag{17}$$

We do not use the *least confident* metric since it completely ignores confidence distribution across labels. While *entropy* is quite popular, and does take into account the confidence distribution, we do not use it since it reaches its maximum for only points for which the classifier must be equally ambiguous about *all* labels; for datasets with many labels (one of our experiments uses a dataset with 26 labels - see Table 1) we may never reach this maximum.

Fig 18 visually shows what uncertainty values look like for the different metrics. Panel (a) displays a dataset with 4 labels. A probabilistic *linear Support Vector Machine (SVM)* is learned on this, and uncertainty scores corresponding to the metrics "margin", "least confident" and "entropy" are visualized in panels (b), (c) and (d) respectively. Darker shades of gray correspond to high uncertainty. Observe that only the "margin" metric in panel (b) achieves scores close to 1 at the two-label boundaries.

There is no best uncertainty metric in general, and the choice is usually application specific (Settles, 2009).

---

[27]The possibly confusing name "least confident" for this idea originated within the context of uncertainty sampling, where we are interested in sampling the most uncertain point, $x^* = \arg\min_x [\max_{y_i \in \{1,2,...,C\}} M(y_i|x)]$, which may be considered to be the instance with the "least most confident label".

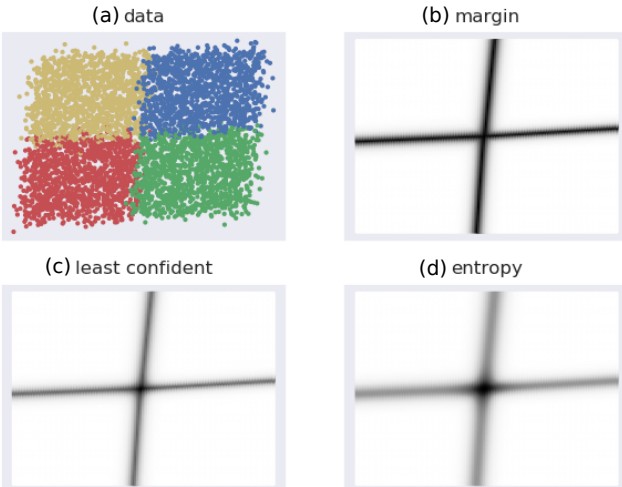

Figure 18: Visualizations of different uncertainty metrics. (a) shows a 4-label dataset on which linear SVM is learned. (b), (c), (d) visualize uncertainty scores based on different metrics, as per the linear SVM, where darker shades imply higher scores.

## A.6 Flattening of the Uncertainty Distribution

Algorithm 5 details the flattening process mentioned in Section 3.6.

---

**Algorithm 5:** Flatten distribution of uncertainty scores $\{u(x_1), u(x_2), ..., u(x_N)\}$

---

**Data:** $\{u(x_1), u(x_2), ..., u(x_N)\}$, number of bins $B$
**Result:** $\{u'(x_1), u'(x_2), ..., u'(x_N)\}$
1  $bin\_size \leftarrow \lceil N/B \rceil, bin\_range \leftarrow 1/B$
2  $bin\_min \leftarrow [\,], bin\_max \leftarrow [\,]$
3  Let $sortedIndex(i) \in \{1, 2, ..., N\}$ be the index of $u(x_i)$ in the sequence of scores ordered by
   non-decreasing values.
4  **for** $j \leftarrow 1$ **to** $B$ **do**
5  $\quad bin\_min[j] \leftarrow \min\{u(x_i)|i \in \{1, 2, ..., N\} \wedge sortedIndex(i) = j\}$
6  $\quad bin\_max[j] \leftarrow \max\{u(x_i)|i \in \{1, 2, ..., N\} \wedge sortedIndex(i) = j\}$
7  **end**
8  **for** $i \leftarrow 1$ **to** $N$ **do**
9  $\quad j \leftarrow sortedIndex(i)$
10 $\quad bin\_num \leftarrow \lceil j/bin\_size \rceil$
11 $\quad boundary\_low \leftarrow (bin\_num - 1) \times bin\_range + \delta$
12 $\quad boundary\_high \leftarrow bin\_num \times bin\_range - \delta$
13 $\quad u'(x_i) \leftarrow low + \frac{u(x_i) - bin\_min[j]}{bin\_max[j] - bin\_min[j]} \times (boundary\_high - boundary\_low)$
14 **end**
15 **return** $\{u'(x_1), u'(x_2), ..., u'(x_N)\}$

---

In lines 11 and 12 of Algorithm 5, we offset bin boundary limits by a small positive value $\delta$ to avoid assignment conflicts across adjacent bins at their boundaries.

This algorithm produces a transformation that looks like the uniform distribution. We prefer the likeness to the uniform distribution since it makes all regions within the interval $[0, 1]$ equally easy to discover.

## A.7 Learned Distributions

This section is an extension of Section 4.1.7.

We consider the IBMM distributions over the uncertainty values. These are difficult to concisely visualize since one IBMM is learned for *each* model size. Hence, we propose the following plot that aggregates distributions across model sizes for a dataset:

1. We set a value for $N$; the number of points to sample.

2. For a model size $\eta_i$, we sample $n_i$ points from its corresponding IBMM, where $n_i \propto \delta F1_i$, the improvement seen at this size. For example, let's say we have explored two model sizes $\eta_1$, $\eta_2$, and these have led to improvements of $\delta F1_1 = 10\%$ and $\delta F1_2 = 20\%$, respectively. Then, $n_1 = 0.33N$ and $n_2 = 0.67N$.

3. The various samples of sizes $n_i$ are pooled together and a *Kernel Density Estimator (KDE)* fit on this data is visualized.

The KDE thus obtained is predominantly shaped by the distributions that resulted in high $\delta F1$. For the case of the LPM these are visualized in Figure 19 for both oracles.

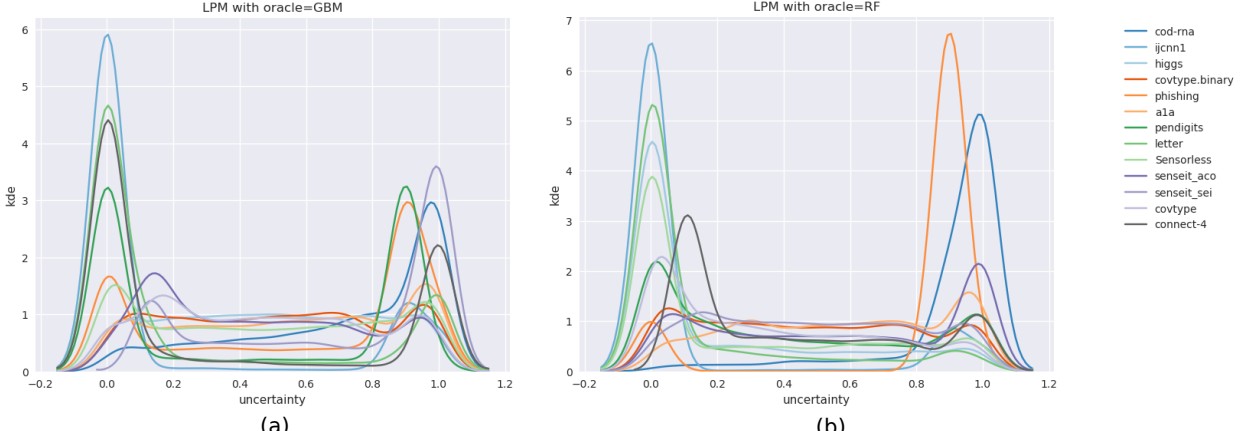

Figure 19: The aggregated IBMMs are visualized for LPMs, when the oracle is a (a) GBM or (b) RF. The corresponding plots for DTs are presented in Section A.9.

It is interesting to see that the optimal strategy, in general, turns out to be to sample from *both* regions of low and high uncertainties.

Going a step further we might wonder what the aggregated distribution would look like if adjusted for the number of instances with a given uncertainty value. For example, we might see a peak on the extreme right for a dataset in Figure 19 simply because most points receive a high uncertainty score.

We use the following technique to visualize such an *adjusted* aggregate distribution:

1. We again pick $N$, the number of points to sample. Exactly like in the previous case: we pool together samples from IBMMs for different model sizes, where the relative sample sizes are decided by the respective $\delta F1$ scores. We fit a KDE to this data, which we refer to as $A$.

2. We fit another KDE to the uncertainty values produced by the oracle for the training data. Let's call this $B$.

3. For $K$ uniformly spaced values of uncertainty $u_k \in [0,1], 1 \leq k \leq K$, we calculate the ratio $p_A(u_k)/p_B(u_k)$, and plot a scaled version of it $c \cdot p_A(u_k)/p_B(u_k)$. The scaling factor $c$ is picked to transform the ratios into probability masses, i.e., $\sum_{k=1}^{K} c \cdot p_A(u_k)/p_B(u_k) = 1$.

Essentially, we *normalize* the sampling probability $p_A(u_k)$ at $u_k$, with $p_B(u_k)$, a quantity representing the number of instances with uncertainty $u_k$.

These plots are shown in Figure 20. The corresponding plots for the DT are shown in Figure 23, Section A.9.

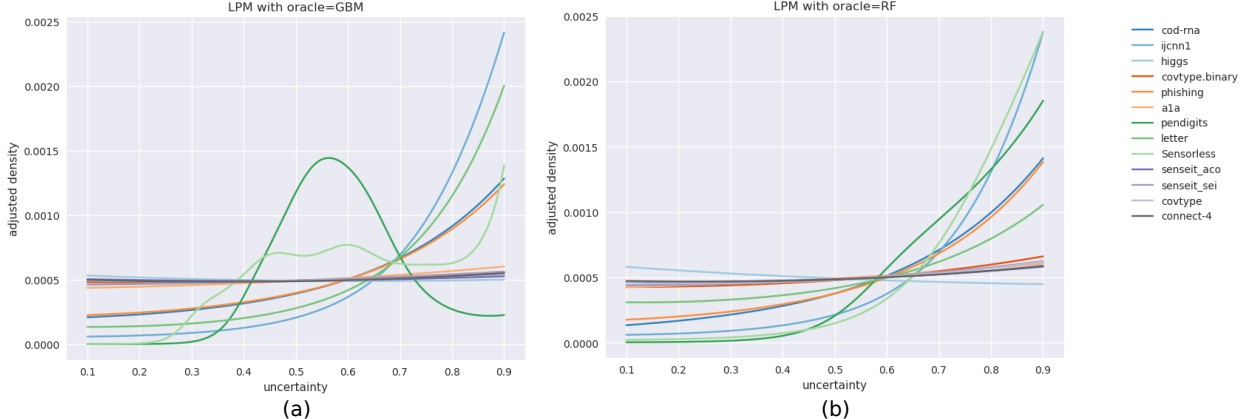

(a)  (b)

Figure 20: Aggregated IBMMs, adjusted for the uncertainty distribution. These plots are for the LPM, using a (a) GBM or (b) RF as an oracle. The corresponding plots for DTs may be found in Section A.9.

While the plots in Figure 19 are indicative of the individual distributions they aggregate (most of the individual distributions have similar shapes; see Figure 25 in Section A.11), this is not true for the adjusted plots in Figure 20 - there are diverse variations that are averaged out. We show some of them in Figure 21, for different datasets and model sizes, for $model = LPM, oracle = GBM$. The size of the dots on the curve represent $p_B(u_k)$ at the corresponding value of $u_k$ on the $x$-axis. These are intended to signify robustness of the adjustment, since they occur in the denominator of the scaled ratios.

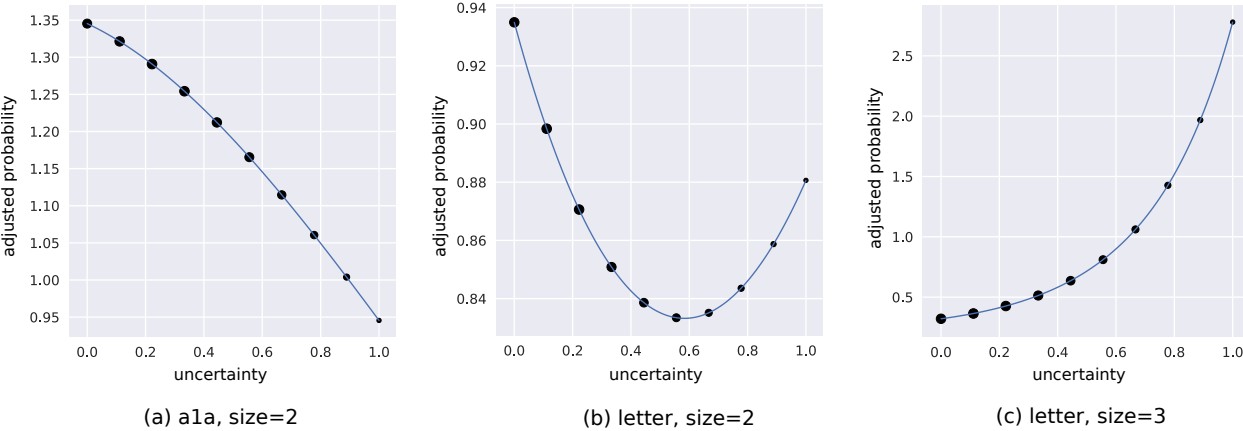

(a) a1a, size=2  (b) letter, size=2  (c) letter, size=3

Figure 21: Adjusted IBMMs for some model sizes and datasets, for $model = LPM, oracle = GBM$. We observe that fairly different distributions may be learned across our experiments.

It is probably important to point out here that typical discussions of uncertainty sampling, such as the classic version (Lewis & Gale, 1994), imply the non-adjusted distributions shown in Figure 19.

## A.8   Running Times

We present the running times (in hour) of our experiments in Table 9. These times should be seen as *indicative only* for the following reasons:

1. We used shared Virtual Machines[28] for our experiments, where we have seen highly variable execution times for our workload. To illustrate this variability both the mean and standard deviation of running times, computed across five runs, are provided.

2. As mentioned earlier, our technique maybe seen as a framework that can work with different optimizers (Section 3.5). Changing the optimizer can greatly influence running times. We show some examples later in this section.

3. Finally, we perform various repeated trials in our experiments, e.g., for computing $s_t$ and fitting $M_t$ in Algorithm 3, which impact running times. However, we haven't explored if they are required (see relevant discussion in Section 6), since our focus in this paper was accurate results and not small running times.

---

[28]Sometimes with differing hardware: 16 GB RAM/6 cores or 32 GB RAM/8 cores, with processor capability equivalent to the Intel Xeon E5 family. All processes were single threaded.

Table 9: This table shows the means and standard deviations of running times in **hours**, as measured across **five** runs. These numbers may be seen as indicative only, since the experiments were run on shared Virtual Machines with varying capacities (which explains the high variability).

| dataset | model_ora | 1 | 2 | 3 | 4 | 5 | 6 | 7 | 8 | 9 | 10 | 11 | 12 | 13 | 14 | 15 |
|---|---|---|---|---|---|---|---|---|---|---|---|---|---|---|---|---|
| cod-rna | lpm_gbm | 3.2±2.1 | 3.1±2.1 | 1.9±0.8 | 1.7±0.6 | 2.0±0.5 | 1.7±0.4 | 3.1±2.0 | 2.3±1.0 | - | - | - | - | - | - | - |
| | lpm_rf | 2.9±1.2 | 3.9±0.7 | 2.8±1.2 | 2.7±0.8 | 2.6±1.1 | 2.4±1.4 | 3.9±1.4 | 3.1±0.4 | - | - | - | - | - | - | - |
| | dt_gbm | 2.1±0.5 | 1.7±0.6 | 2.3±0.3 | 2.0±0.4 | 1.3±0.3 | 1.7±0.3 | 1.6±0.4 | 1.7±0.2 | 1.4±0.3 | 1.4±0.3 | - | - | - | - | - |
| | dt_rf | 1.5±0.8 | 1.9±0.8 | 2.4±0.6 | 2.2±0.5 | 1.3±0.6 | 1.4±0.5 | 1.5±0.2 | 1.5±0.2 | 1.2±0.1 | 1.5±0.0 | - | - | - | - | - |
| ijcnn1 | lpm_gbm | 2.4±1.2 | 3.5±2.5 | 3.8±2.1 | 2.5±1.6 | 3.5±2.1 | 3.6±2.1 | 3.1±2.1 | 3.3±2.0 | 3.1±1.9 | 2.6±1.2 | 2.6±1.0 | 4.4±2.9 | 3.7±2.1 | 3.2±1.9 | 3.8±2.2 |
| | lpm_rf | 3.7±1.5 | 2.4±0.8 | 3.9±0.8 | 4.4±1.4 | 4.1±1.0 | 3.6±1.4 | 4.0±1.1 | 3.1±0.2 | 4.0±0.6 | 4.0±0.8 | 3.7±0.7 | 3.6±1.3 | 3.0±0.6 | 4.5±0.5 | 4.0±1.3 |
| | dt_gbm | 2.6±0.4 | 3.2±0.6 | 3.4±0.9 | 3.4±0.7 | 3.4±0.8 | 3.4±0.7 | 4.0±1.4 | 3.4±1.0 | 3.1±0.3 | 3.4±0.8 | 3.1±1.0 | 3.4±0.9 | 3.1±0.8 | 2.8±1.1 | 2.2±0.2 |
| | dt_rf | 2.6±0.6 | 2.8±0.8 | 3.0±0.3 | 3.1±0.6 | 2.8±0.9 | 3.2±0.4 | 3.5±0.6 | 3.0±1.0 | 2.7±0.2 | 3.0±0.5 | 3.0±0.5 | 2.7±0.6 | 2.7±0.4 | 2.2±0.5 | 2.4±0.2 |
| higgs | lpm_gbm | 2.8±1.4 | 2.9±1.7 | 2.7±1.4 | 2.8±1.5 | 2.3±1.4 | 2.4±1.1 | 2.6±1.6 | 2.5±1.2 | 2.6±1.6 | 1.9±0.8 | 2.4±1.3 | 2.9±1.7 | 3.2±2.0 | 3.3±2.3 | 3.2±1.9 |
| | lpm_rf | 5.6±1.2 | 5.3±0.8 | 3.7±0.9 | 3.6±0.9 | 4.4±1.1 | 4.2±1.6 | 4.5±1.5 | 4.6±1.4 | 3.5±1.2 | 4.3±1.0 | 3.3±1.1 | 2.9±0.7 | 3.4±1.1 | 3.9±0.8 | 3.9±0.7 |
| | dt_gbm | 2.4±0.6 | 2.7±0.2 | 3.8±0.7 | 2.9±0.8 | 4.5±0.5 | 4.7±0.3 | - | - | - | - | - | - | - | - | - |
| | dt_rf | 2.5±0.4 | 2.3±0.6 | 3.2±0.8 | 2.7±0.6 | 3.2±0.5 | 3.3±0.5 | - | - | - | - | - | - | - | - | - |
| covtype.binary | lpm_gbm | 4.0±2.8 | 2.7±1.6 | 2.5±1.5 | 2.8±1.8 | 3.0±1.6 | 3.4±2.3 | 2.3±1.2 | 3.0±1.9 | 3.4±2.4 | 3.1±1.9 | 2.6±1.2 | 3.0±2.1 | 2.6±1.3 | 3.4±2.1 | 3.5±2.1 |
| | lpm_rf | 4.9±1.9 | 3.4±1.2 | 4.1±1.4 | 4.6±1.3 | 4.1±1.6 | 3.1±1.1 | 3.5±0.6 | 3.1±1.1 | 3.9±1.6 | 4.3±1.1 | 4.1±0.9 | 4.4±1.5 | 3.0±1.0 | 3.7±0.5 | 3.8±1.1 |
| | dt_gbm | 2.4±1.0 | 3.1±1.3 | 3.8±1.0 | 2.7±0.8 | 2.5±0.5 | 2.3±0.5 | 2.4±0.6 | 2.5±0.8 | 2.6±0.5 | - | - | 2.2±0.0 | - | - | - |
| | dt_rf | 2.5±0.4 | 2.8±0.6 | 2.9±0.4 | 2.8±0.4 | 2.4±0.7 | 2.0±0.5 | 2.2±0.5 | 2.1±0.5 | 2.1±0.3 | 2.4±0.3 | - | - | - | - | - |
| phishing | lpm_gbm | 2.6±1.3 | 2.8±1.4 | 3.4±2.2 | 3.2±1.7 | 3.3±1.5 | 4.5±2.6 | 3.5±2.0 | 3.1±2.1 | 3.1±1.7 | 2.8±1.6 | 3.7±2.2 | 2.7±1.3 | 2.6±1.7 | 3.7±2.1 | 2.8±1.7 |
| | lpm_rf | 3.4±0.5 | 3.8±0.9 | 3.8±0.9 | 3.8±0.9 | 4.3±2.0 | 4.2±0.5 | 5.3±1.1 | 4.9±0.9 | 4.1±0.8 | 5.4±0.6 | 4.3±1.3 | 5.5±1.5 | 5.1±1.7 | 5.5±1.6 | 5.1±1.7 |
| | dt_gbm | 2.3±0.4 | 2.4±0.4 | 2.8±0.5 | 2.7±1.0 | 2.6±0.6 | 2.7±1.2 | 2.7±0.7 | 2.7±0.7 | 2.4±0.9 | 2.4±0.7 | 2.2±0.7 | 2.1±0.3 | 2.2±0.4 | 2.0±0.6 | 2.1±0.5 |
| | dt_rf | 1.8±0.2 | 2.2±0.2 | 2.2±0.4 | 2.8±0.7 | 1.9±0.4 | 2.1±0.6 | 2.3±0.6 | 2.3±0.6 | 2.0±0.6 | 2.4±0.9 | 1.6±0.3 | 1.8±0.3 | 1.6±0.4 | 1.8±0.3 | 1.7±0.3 |
| a1a | lpm_gbm | 2.9±1.7 | 3.7±2.4 | 3.1±2.1 | 3.4±2.1 | 3.9±2.7 | 4.5±3.1 | 4.3±2.9 | 3.6±2.2 | 2.1±0.9 | 3.9±2.6 | 4.4±2.7 | 4.1±2.7 | 4.4±2.5 | 3.2±1.5 | 2.2±0.8 |
| | lpm_rf | 4.2±1.0 | 4.3±1.2 | 5.6±1.8 | 4.7±0.7 | 5.0±1.3 | 5.6±2.1 | 5.2±1.4 | 3.9±1.1 | 2.8±0.7 | 5.1±1.1 | 4.8±1.2 | 5.5±1.8 | 5.1±1.7 | 4.3±0.9 | 3.2±0.9 |
| | dt_gbm | 2.1±0.2 | 3.4±0.7 | 3.2±0.5 | 3.8±1.0 | 3.4±1.0 | 3.7±0.9 | 2.8±1.0 | 2.3±0.8 | 5.3±0.0 | - | - | - | - | - | - |
| | dt_rf | 1.9±0.2 | 2.5±0.4 | 3.0±0.6 | 3.2±0.6 | 3.1±0.9 | 3.2±0.8 | 2.8±0.5 | 2.6±0.8 | 2.9±0.2 | 2.6±0.8 | 2.7±0.3 | 3.6±0.0 | - | - | 3.9±0.0 |
| pendigits | lpm_gbm | 1.1±0.8 | 1.4±0.9 | 1.2±0.7 | 1.1±0.9 | 0.9±0.6 | 0.9±0.6 | 1.4±0.9 | 1.2±0.8 | 1.2±0.7 | 1.1±0.6 | 1.3±0.9 | 1.3±0.8 | 1.2±0.7 | 1.3±0.7 | 1.0±0.7 |
| | lpm_rf | 1.9±0.3 | 1.8±0.4 | 1.6±0.2 | 1.3±0.3 | 1.5±0.3 | 1.4±0.3 | 1.7±0.2 | 1.7±0.3 | 1.7±0.4 | 1.5±0.3 | 1.7±0.3 | 1.6±0.3 | 1.5±0.3 | 1.9±0.2 | 1.6±0.3 |
| | dt_gbm | 2.1±0.9 | 2.9±0.9 | 2.3±0.6 | 2.7±1.0 | 3.3±0.9 | 2.6±0.9 | 2.3±0.6 | 2.4±0.4 | 2.2±0.7 | 2.2±0.4 | 2.2±0.6 | 1.5±0.1 | 2.0±0.3 | 2.1±0.3 | 1.5±0.1 |
| | dt_rf | 2.9±0.9 | 2.5±0.7 | 2.3±0.3 | 3.0±0.4 | 2.6±0.5 | 3.1±0.7 | 3.5±0.8 | 2.3±0.8 | 2.1±0.5 | 1.9±0.4 | 1.9±0.4 | 2.0±0.3 | 2.0±0.3 | 2.1±0.3 | 2.0±0.1 |
| letter | lpm_gbm | 1.4±0.7 | 1.3±0.8 | 1.6±1.0 | 1.5±0.9 | 1.9±1.2 | 1.6±1.0 | 1.4±0.9 | 1.5±0.9 | 1.7±1.0 | 1.6±1.0 | 1.6±0.9 | 1.7±1.0 | 1.8±1.1 | 2.0±1.3 | 1.8±1.2 |
| | lpm_rf | 1.9±0.4 | 1.9±0.5 | 2.0±0.4 | 1.9±0.3 | 2.2±0.6 | 2.2±0.3 | 2.2±0.4 | 1.9±0.5 | 2.5±0.4 | 2.1±0.3 | 2.2±0.2 | 2.1±0.3 | 2.5±0.4 | 2.4±0.4 | 2.3±0.7 |
| | dt_gbm | 2.6±1.3 | 2.7±0.6 | 2.2±0.6 | 2.9±0.6 | 2.6±0.4 | 2.5±0.7 | 1.8±0.2 | 2.2±0.4 | 2.1±0.7 | 2.3±0.4 | 2.3±0.5 | 2.4±0.3 | 2.3±0.6 | 2.1±0.4 | 2.0±0.4 |
| | dt_rf | 1.8±0.4 | 2.4±0.8 | 2.2±0.7 | 2.3±1.0 | 2.5±0.6 | 2.1±0.6 | 2.5±1.0 | 2.0±0.7 | 2.1±0.4 | 1.9±0.5 | 2.1±0.2 | 2.1±0.4 | 1.9±0.6 | 2.0±0.3 | 1.9±0.3 |
| Sensorless | lpm_gbm | 1.4±0.8 | 1.5±1.2 | 1.2±0.9 | 1.8±1.2 | 1.8±1.2 | 1.7±1.0 | 1.5±0.9 | 1.3±0.7 | 1.0±0.5 | 1.4±0.9 | 1.5±0.9 | 1.4±0.9 | 1.2±0.7 | 1.1±0.6 | 1.0±0.5 |
| | lpm_rf | 2.2±0.4 | 2.2±0.3 | 1.2±0.9 | 2.0±0.6 | 1.9±0.4 | 2.1±0.5 | 1.8±0.5 | 1.6±0.5 | 1.6±0.4 | 2.2±0.5 | 2.0±0.3 | 1.8±0.6 | 1.6±0.4 | 1.3±0.3 | 1.3±0.3 |
| | dt_gbm | 2.3±0.3 | 1.9±0.5 | 1.8±0.6 | 2.5±1.1 | 3.2±1.0 | 3.9±1.7 | 5.0±1.2 | 4.9±0.9 | 5.2±0.9 | 5.3±0.8 | 5.6±1.0 | 6.2±1.5 | 5.3±1.0 | 5.3±1.1 | 3.3±0.0 |
| | dt_rf | 2.1±0.6 | 2.5±1.0 | 3.0±0.9 | 2.3±0.8 | 2.9±1.0 | 4.8±0.3 | 4.8±0.6 | 4.7±0.6 | 5.8±0.8 | 4.9±0.7 | 4.8±0.8 | 5.0±0.5 | 6.0±0.9 | 4.5±0.0 | 5.3±0.4 |
| senseit_aco | lpm_gbm | 1.0±0.6 | 1.0±0.6 | 1.2±0.7 | 0.9±0.4 | 1.0±0.6 | 0.9±0.5 | 0.9±0.4 | 1.0±0.5 | 1.1±0.6 | 1.0±0.6 | 1.0±0.6 | 1.0±0.5 | 1.0±0.5 | 1.0±0.6 | 0.9±0.5 |
| | lpm_rf | 1.4±0.3 | 1.6±0.4 | 1.6±0.2 | 1.6±0.2 | 1.4±0.4 | 1.7±0.2 | 1.5±0.2 | 1.6±0.2 | 1.6±0.2 | 1.4±0.1 | 1.4±0.4 | 1.4±0.3 | 1.1±0.2 | 1.4±0.2 | 1.5±0.2 |
| | dt_gbm | 2.5±0.4 | 3.8±1.2 | 4.1±1.2 | 5.2±1.6 | 5.1±1.8 | 6.0±1.5 | 7.3±0.9 | - | - | - | - | - | - | - | - |
| | dt_rf | 2.9±1.1 | 3.4±0.5 | 3.9±0.8 | 4.6±0.8 | 4.6±0.6 | 5.7±0.4 | 5.7±0.4 | 6.8±0.0 | - | - | - | - | - | - | - |
| senseit_sei | lpm_gbm | 1.2±0.6 | 0.9±0.5 | 1.0±0.5 | 0.9±0.5 | 1.0±0.6 | 1.1±0.7 | 1.0±0.6 | 1.3±0.8 | 1.1±0.7 | 1.2±0.7 | 1.1±0.6 | 1.3±0.7 | 1.0±0.6 | 1.1±0.7 | 1.1±0.6 |
| | lpm_rf | 1.6±0.6 | 1.5±0.1 | 1.4±0.3 | 1.6±0.5 | 1.5±0.2 | 1.3±0.4 | 1.5±0.2 | 1.6±0.3 | 1.7±0.4 | 1.6±0.4 | 1.3±0.2 | 1.5±0.3 | 1.4±0.5 | 1.6±0.5 | 1.7±0.4 |
| | dt_gbm | 1.9±1.2 | 3.2±0.6 | 3.8±0.5 | 5.4±1.7 | 5.1±1.5 | 5.3±1.5 | 5.9±2.0 | 3.4±0.3 | - | - | - | - | - | - | - |
| | dt_rf | 3.1±0.7 | 3.2±0.9 | 3.6±1.0 | 4.0±1.3 | 4.3±0.5 | 5.3±0.5 | 6.8±0.0 | - | - | - | - | - | - | - | - |
| covtype | lpm_gbm | 1.4±0.8 | 1.3±0.8 | 1.1±0.8 | 1.4±0.8 | 1.3±0.8 | 1.1±0.6 | 1.1±0.8 | 1.2±0.7 | 1.2±0.7 | 0.9±0.6 | 1.2±0.7 | 1.4±1.0 | 1.2±0.7 | 1.5±0.9 | 1.5±1.0 |
| | lpm_rf | 1.7±0.6 | 1.6±0.5 | 1.4±0.3 | 1.5±0.4 | 1.6±0.6 | 1.7±0.5 | 1.5±0.4 | 1.6±0.6 | 2.2±0.9 | 1.9±0.7 | 1.8±0.6 | 1.5±0.6 | 1.9±0.9 | 1.7±0.5 | 1.6±0.5 |
| | dt_gbm | 2.1±0.6 | 2.1±0.2 | 3.4±1.2 | 3.0±1.0 | 3.2±0.8 | 3.1±0.4 | 3.0±0.7 | 3.3±0.8 | 3.1±0.8 | 3.4±1.4 | 2.0±0.4 | 3.1±0.9 | 2.6±0.8 | 2.8±0.9 | 3.0±0.8 |
| | dt_rf | 1.6±0.3 | 3.1±0.1 | 3.4±0.8 | 2.6±1.2 | 2.7±0.8 | 2.5±0.6 | 2.3±0.3 | 2.3±0.4 | 2.4±0.4 | 2.7±0.5 | 2.6±0.4 | 2.6±0.4 | 2.7±0.4 | 2.7±0.5 | 2.6±0.6 |
| connect-4 | lpm_gbm | 1.1±0.6 | 1.0±0.7 | 1.0±0.7 | 1.1±0.7 | 1.1±0.7 | 1.0±0.5 | 1.3±0.8 | 1.3±0.8 | 1.1±0.6 | 1.1±0.7 | 1.0±0.7 | 1.1±0.7 | 1.0±0.6 | 0.9±0.5 | 1.1±0.7 |
| | lpm_rf | 1.5±0.3 | 1.3±0.3 | 1.2±0.2 | 1.5±0.3 | 1.6±0.4 | 1.3±0.4 | 1.2±0.7 | 1.8±0.2 | 1.6±0.3 | 1.4±0.4 | 1.3±0.5 | 1.0±0.7 | 1.2±0.7 | 1.4±0.3 | 1.3±0.6 |
| | dt_gbm | 2.5±1.2 | 3.2±0.5 | 3.3±0.7 | 2.9±0.7 | 2.3±1.0 | 2.7±0.9 | 3.2±0.7 | 2.9±1.4 | 2.2±0.5 | 2.8±0.8 | 2.3±1.1 | 3.0±1.0 | 2.7±1.1 | 2.9±1.3 | 2.7±1.4 |
| | dt_rf | 2.5±0.6 | 2.6±0.7 | 3.0±0.4 | 2.9±0.6 | 3.1±0.6 | 3.4±0.6 | 3.3±0.4 | 2.8±0.7 | 3.4±0.5 | 2.8±0.7 | 2.7±0.5 | 3.0±0.3 | 2.6±0.6 | 2.7±0.5 | 2.6±0.2 |

We now illustrate the effect of using different optimizers. We use the *LIPO* (Malherbe & Vayatis, 2017) and *pySOT* (Eriksson et al., 2019) optimizers, in addition to TPE (in the *hyperopt* library) that is used in the main paper. The *pySOT* library supports multiple search strategies; we show results for *Stochastic Radial Basis Function* (denoted as "pysot(srbf)") (Regis & Shoemaker, 2007; 2009) and *Dynamic Coordinate Search* (denoted as "pysot(dycors)") (Regis & Shoemaker, 2013).

In Figure 22, we show runtimes for the two datasets *covtype* (row A) and *letter* (row B), where the model is *LPM* (size 2 and 3 respectively) and the oracle is *GBM*. This setup uses *ten trials*, each with an optimizer budget of 1000 iterations. The datasets contain 2000 instances, and the splits are the same as in our other experiments, i.e., $D_{train} : D_{val} : D_{test} :: 60 : 20 : 20$. The F1 macro score on $D_{train}$ is shown since the optimizer's objective function is defined on this split, making it a good choice for tracking progress. Scores are shown against both number of iterations and wallclock time.

Consider row A in Figure 22. Although hyperopt eventually obtains the highest accuracy, it is clear from (b) that if our budget were less than $\sim 250$ sec, *pysort(srbf)* would be a better choice since it reaches higher scores faster. In the case of row B, the difference in performance is starker: *pysot(dycors)* is not only eventually more accurate on average, but its performance is either at par with *hyperopt* or better, throughout the course of the experiment.

Figure 22: Impact of optimizers on running times is illustrated both in terms of number of iterations and wallclock time. Data is visualized for the LPM (model) and GBM (oracle) combination. Row A corresponds to an LPM of $size = 2$ for the `covtype` dataset, while Row B shows plots for an LPM of $size = 3$ for the `letter` dataset. The curves show average values, while the shaded region represents one standard deviation.

## A.9 Uncertainty Distribution for DT

The uncertainty distributions learned when using a DT with different oracles are shown in Figure 23. The first row shows visualizes the aggregation of the IBMMs that were learned, while the second row shows them adjusted with the uncertainty distribution from the oracle. These are analogues of the LPM plots in Figure 19 and Figure 20.

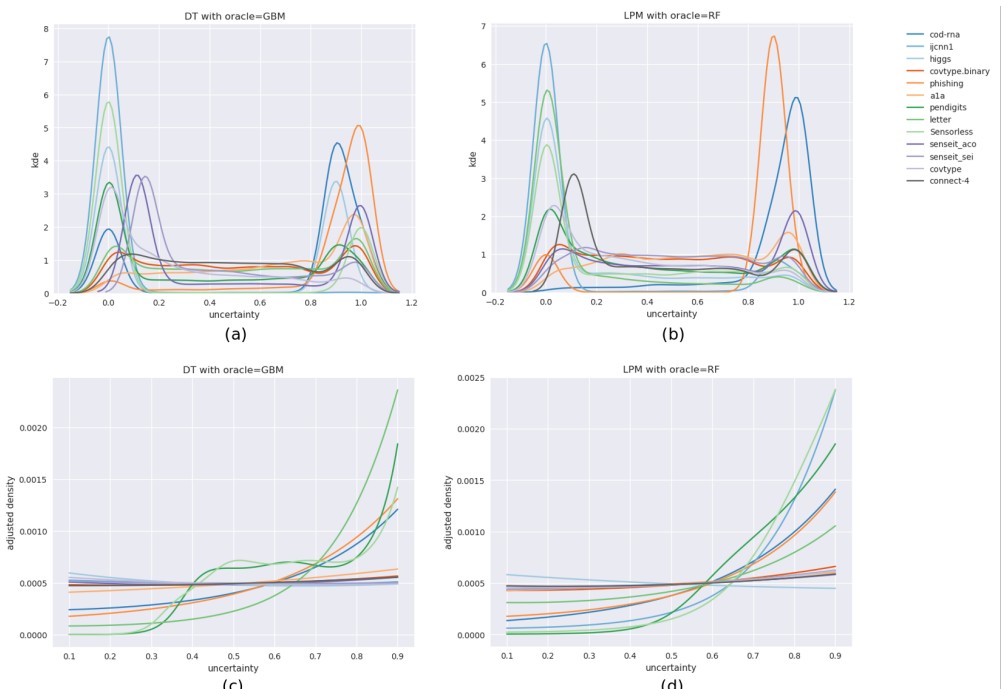

Figure 23: The aggregated IBMMs visualized when using a DT as our interpretable model. The top row shows the aggregated IBMMs for different oracles: GBM (left) and RF (right). The bottom row visualizes the IBMMs adjusted for the uncertainty distribution.

The patterns we observe here are similar to what we saw for LPMs:

1. Top-row: the IBMMs seem to prefer both low and high uncertainty regions.

2. Bottom-row: when adjusted with the oracle's uncertainty distribution, there is sampling across the entire range of uncertainty values, with slight/occasional preference for higher uncertainties.

## A.10 Compaction Profiles

Figure 24 shows the compaction profiles for all model-oracle combinations. These are discussed in Section 4.1.6, in reference to Figure 5.

## A.11 IBMMs for Different Model Sizes

Figure 25 shows the IBMMs learned over uncertainties for individual model sizes of the $LPM$, with $GBM$ as the oracle,. These are *not* adjusted with the density of the uncertainty distribution. The plot shows them for the datasets (a) `covtype.binary` and `Sensorless`. We observe that the unified IBMM weighted by improvements, shown in Figure 19, are indicative of the individual distributions in this Figure 25.

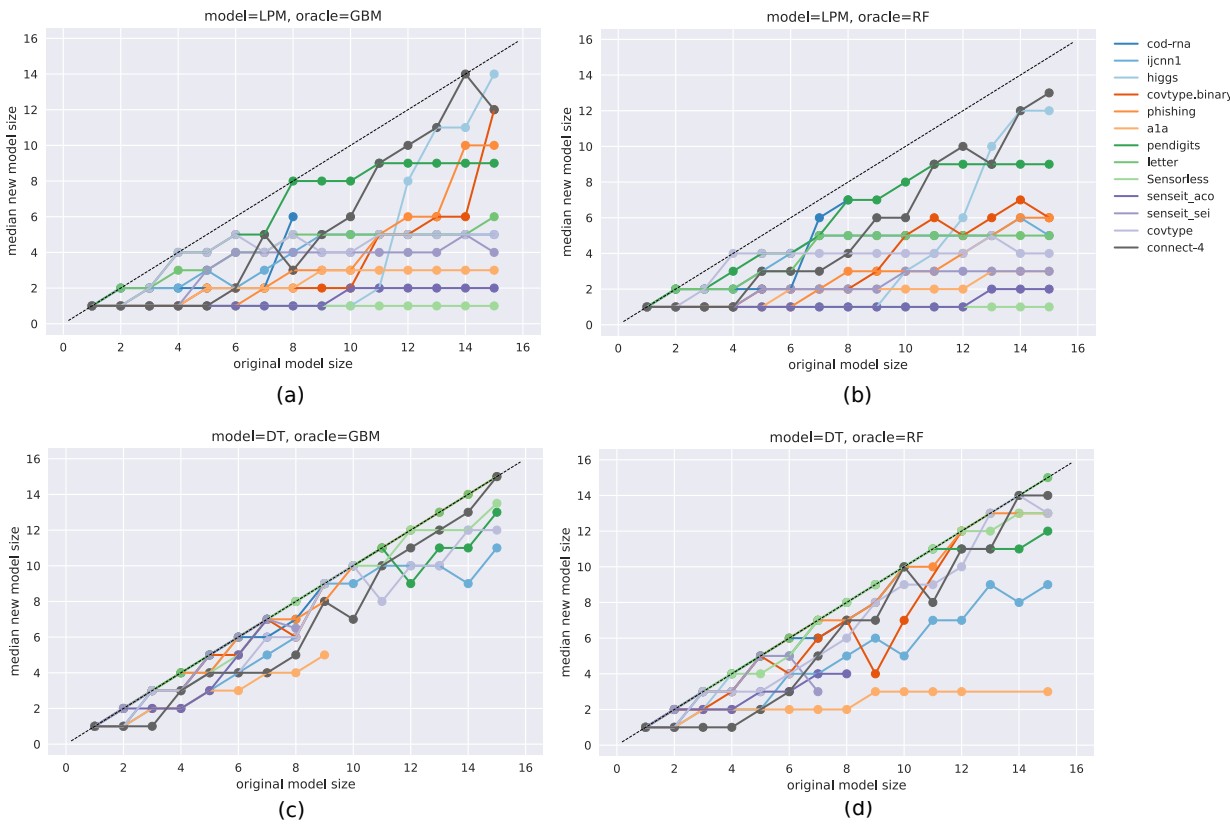

Figure 24: For different combinations of models and oracles: $\{LPM, DT\} \times \{GBM, RF\}$, these plots show the size of an improved model (y-axis), that may replace a traditionally trained model of a given size (x-axis). A model is considered as a replacement for another if its accuracy is at least as high as the latter.

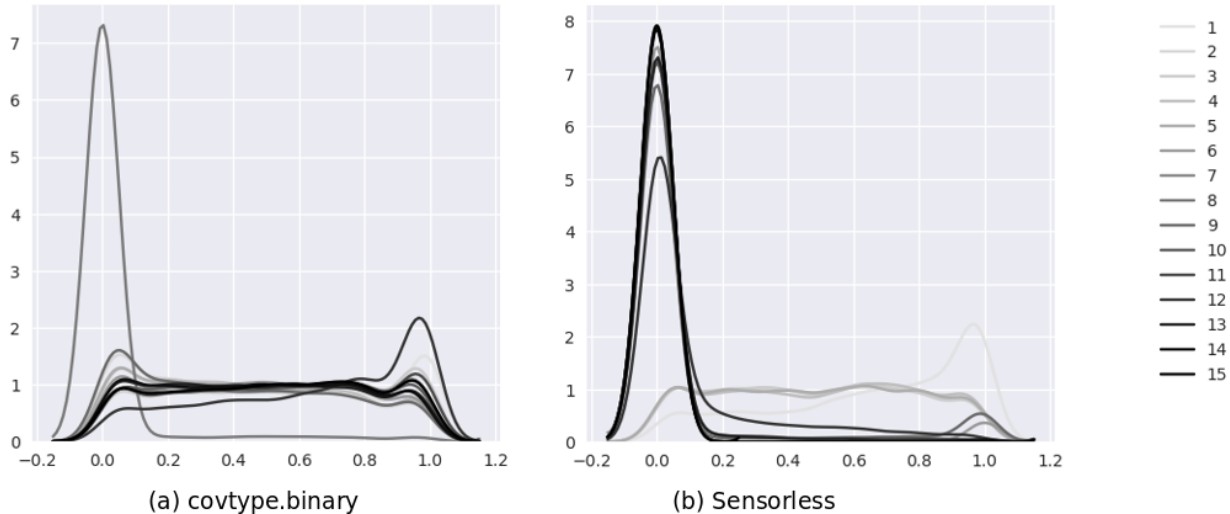

Figure 25: IBMM distributions for model sizes $\{1, 2, ..., 15\}$, for the datasets (a) `covtype.binary` and (b) `Sensorless`. These are for the combination of using $LPM$ as the model with $GBM$ as an oracle. Darker curves indicate higher model sizes.

## A.12 Improvements Relative to Oracle

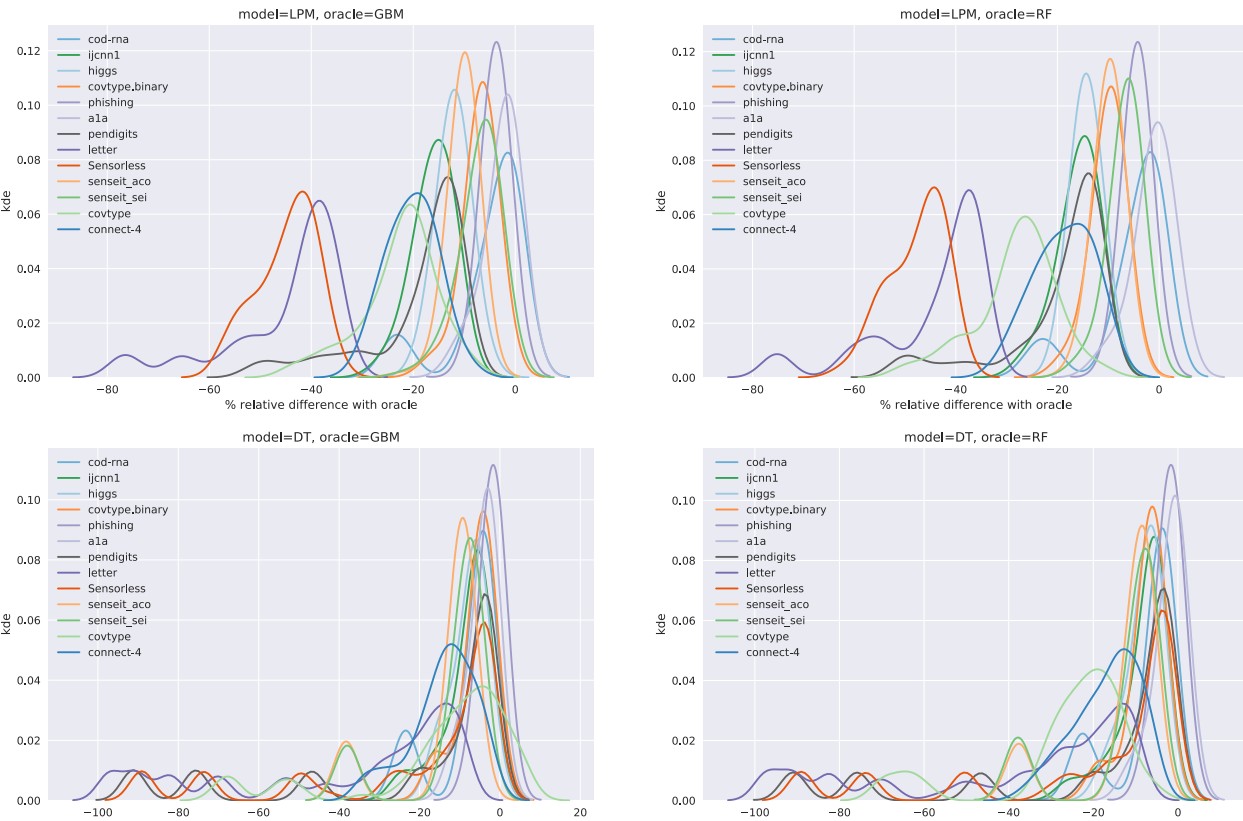

Figure 26: These plots show the distribution of the percentage relative difference of a model's improved score w.r.t. to the accuracy of the oracle it is trained with. We note that this quantity is almost always non-positive as claimed in Equations 1, 2 and 3.

Some of the positive values we see in Figure 26 may be attributed to spillovers due to the *kde* fit. Their magnitudes and occurrences are typically small: these are detailed in Table 8.

## A.13 Feature Selection for n-gram DT

For the experiments in Section 4.3.1, we perform feature selection to reduce their running time. After the n-gram ($n \in \{1, 2, 3\}$) vocabulary is created from the training data, we perform a $\chi^2$-test to select the $k-$best features. The original number of features is 5308. To pick the smallest useful set of features, we test different values of $k \le 1000$. A test constitutes of:

1. Construct a DT, for a given *max_depth*, on the original set of features. Obtain its test accuracy, $F1_{all}$.

2. Construct a DT, with the same *max_depth*, using only the $k$ best features as per the $\chi^2$-test, and obtain its test accuracy $F1_k$.

3. Report:
$$\delta F1 = 100 \times \frac{F1_k - F1_{all}}{F1_{all}}$$

We use the "macro" averaging for the $F1$ score to be consistent with other experiments in the paper. All reported $\delta F1$ are *averaged over ten runs.*

Figure 27 shows how $\delta F1$ varies with $k$.

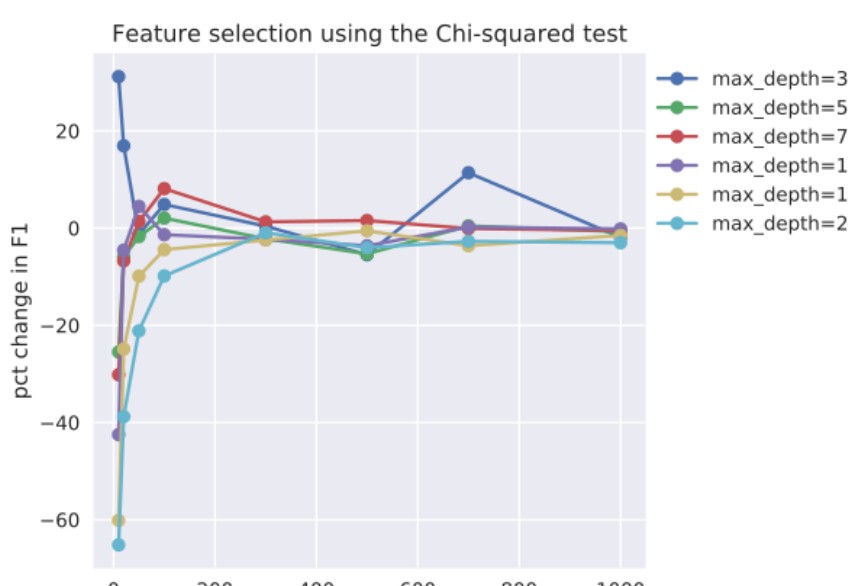

Figure 27: The relationship between $\delta F1$ and $k \leq 1000$. Each data point is an average over ten runs.

We observe that at around 600 features, $\delta F1 \approx 0\%$. The only exception is the case for $max\_depth = 3$, but that is admissible since $\delta F1 > 0$, i.e., we seem to be improving the accuracy.

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
