# OpenReview forum: "Learning Interpretable Models Using an Oracle"
_TMLR — Rejected by TMLR_

### Review · Reviewer_mM6u · 2023-02-14

**Summary Of Contributions:**

This work explores a strategy for mitigating the accuracy vs interpretability (as measured by some notion of model size) trade-off.  To do this, the authors propose training the interpretable model (ie, model with a maximum size) on a re-sampled training set where that re-sampling is guided by a highly accurate model's certainty in its predictions.  One of the key steps in doing this is to parameterize the method for performing that re-sampling in a way such that we can learn what distribution to re-sample from.  Empirically, this approach is found to be quite effective at reducing this trade-off.

**Audience:**

Yes

**Claims And Evidence:**

Yes

**Requested Changes:**

Important:
-  Please explain why the dataset size was capped at 10,000 or provide results without the cap.
-  Please provide (or summarize) the results for the analysis in Section 4.1.1 without lower-bounding $\delta F1$ at 0.

Minor:
-  This may be because I'm not familiar with DPs, but it wasn't clear in Section 3 why the dataset is partitioned (eg, Line 2 of Algorithm 2) before it is re-sampled.
-  Please elaborate on how lower-bounding $N_s$ provides statistically significant results.  This was mentioned in a few places, but I didn't see an explanation.

**Strengths And Weaknesses:**

Strengths:
- Overall, Sections 1 through 3 are well written and clear.
-  The analyses in Section 4 are very thorough (if somewhat complicated) and demonstrate that the proposed method is effective.  Further, several different baselines and ablations are considered.

Weaknesses:
-  It seems rather artificial that all the datasets were capped at 10,000 samples.  It is fine if the proposed method is more effective in lower-data regimes, but, if that is the case, that should be made clear.
-  While the argument that a user can always choose to use $M_{base}$ in practice is reasonable, it still seems artificial to prevent $\delta F1$ from being less than zero when analyzing the effectiveness of the proposed method.

---

> ### Author Response · Authors · 2023-04-03
> **Response to review comments by reviewer mM6u**
>
> We thank the reviewer for their prompt and constructive review. We address their concerns below:
>
> 1. Capping dataset sizes at 10k instances: The dataset sizes were kept constant across experiments to remove size as a confounding factor, and the size was specifically held at 10k since the datasets “phishing” ([https://www.csie.ntu.edu.tw/~cjlin/libsvmtools/datasets/binary.html](https://www.csie.ntu.edu.tw/~cjlin/libsvmtools/datasets/binary.html)) and “pendigits” ([https://www.csie.ntu.edu.tw/~cjlin/libsvmtools/datasets/multiclass.html](https://www.csie.ntu.edu.tw/~cjlin/libsvmtools/datasets/multiclass.html)) have ~10k instances. We will add this explanation to the paper.
> 2. Lower bounding of $\delta F1$: We have presented two sets of results - Table 2 (Section 4.1.6) has the capped lower bound where the baseline is $M_{base}$, and Table 3 (Section 4.1.8) does not have a lower bound, uses the t-test to accept the optimizer best score, and the baseline is resource-equivalent to the optimizer’s environment.
> The reason for presenting two sets of results appears in Section 4.1.1, especially in the list item beginning with “Optimizer performance”, but in short, $M_{base}$ has access to more data and a larger hyperparameter search space, which makes it hard to understand if the difference in performance of the two models (baseline and the one produced by our algorithm) is caused by the proposed algorithm or by these other factors. This is especially true for observed relative improvements close to 0. The results presented in Table 3 (Section 4.1.8) are more rigorous in this respect, and show that indeed in a handful of cases, we perform worse than the baseline.\
> *Minor note*: In our description in 4.1.1., we should have also cross-referenced Section 4.1.3, where the hyperparameter search spaces are described. We missed that, and we’ll be adding it to the next version of the manuscript.
>
> 3. We used the term “partition” in the sense of assigning clusters to instances as per the DP. This terminology is used within the DP literature (for ex. here [https://pluto.huji.ac.il/~galelidan/52558/Material/DirichletProcess.pdf](https://pluto.huji.ac.il/~galelidan/52558/Material/DirichletProcess.pdf)), but we will clarify its use in the paper.
>
> 4. The use of the term “Statistical Significance” in the case of $N_s$ is somewhat loose, and we apologize for that. What we meant is that for the same DP parameters a small sample can lead to highly different training distributions, and hence the learned DP parameters alone cannot be used to reproduce results. This inconsistency reduces as sample size increases: for our setup of a univariate distribution we observed this to happen when the sample size is in a few hundreds.
> We will explain the reason for the lower bounding of $N_s$ in the paper, and remove the use of the  term “Statistical Significance” in this context.

---

> > ### Comment · Reviewer_mM6u · 2023-04-05
> > **Thanks for clarifying, one further question.**
> >
> > Doesn't have $M_{base}$ have access to more data and to a larger hyper parameter search because your algorithm needs to use that data/compute internally?  For example, the training process for $M_{t}$ cannot use $D_{val}$ (while the training process for $M_{base}$ can) because your algorithm uses $D_{val}$ to determine how good $M_{t}$ is.  Similarly, your algorithm trains multiple $M_{t}$'s, so you can't use as thorough of a hyper parameter search as you could while training $M_{base}$.

---

> > > ### Author Response · Authors · 2023-04-10
> > > **Reg. resources available to baselines**
> > >
> > > Our algorithm does not require the resources available to a baseline model and $M_t$  to differ. They should be ideally be the same for rigorous measurement, i.e., the baseline should use the same hyperparam. search space as $M_t$ and the model selection process should be identical, i.e., simple random validation. This is not true for $M_{base}$ (results in Table 2, Section 4.1.6) which shows a workflow that a practitioner might follow (cross-validation, a "typical" large hyperparam space, pick the best model) - but this is true for $M_{init}$, the baseline used for the results in Table 3, Section 4.1.8.
> > >
> > > In other words, the measurements using $M_{base}$ show the outcomes a practitioner is likely to see, while the ones using $M_{init}$  allow for rigorous comparison.

---

### Review · Reviewer_8AEa · 2023-04-16

**Summary Of Contributions:**

The paper proposes a method to learn the training distribution, with the goal of using the learnt training distribution to improve the accuracy of models constrained to be remain interpretable.

**Audience:**

Yes

**Claims And Evidence:**

No

**Requested Changes:**

- Make differences between this paper and Ghose & Ravindran (2020) very clear so that novelty claims are substantiated
- Add distillation baseline
- Rewrite: significantly shorten paper, be more careful with interpretability claims, be more careful with citing related work


**Strengths And Weaknesses:**

I have several concerns about this paper.

First of all, the novelty with respect to Ghose & Ravindran (2020) is unclear to me; the authors claim that "this work
may be seen as a non-trivial extension" but after reading the paper and looking at, for example, Algorithm 2 and comparing it to Algorithm 2 in the older paper, or another example, the equations in Section 2.3 compared to the equations in Section 1 of the older paper, the differences are not clear. As it stands, I do not think that this submission has enough novelty with respect to the older paper.

Secondly, I found the interpretability focus in the paper title, abstract, and introduction confusing and detracting from the method proposed in the paper. For a purported interpretability paper, no user studies were done to lay the foundation to back up the (implicit) claim that size can be compared across different model types, i.e. d=1 for a linear probability model (where d refers to the number of terms) can be compared to d=1 for a decision tree (where d refers to tree depth). This is very confusing when reviewing the results. Take Table 1 as an example, on a few datasets (Higgs, covtype.binary), for the same d, the numbers are very different between lpm_oracle and dt_oracle. Now, if a user study had been done to lay the foundation here, the authors may have found something like -- human subjects take approximately the same amount of time to solve a task using a linear model of d=x and a tree of d=y. Then these 2 different model types have more basis to be compared to each other. As such, I am not convinced by the loose treatment of interpretability. The claim that the method can improve accuracy for a model by learning the training distribution should be a claim of independent interest to anyone who trains ML models.

Third, the results are sensitive to the choice of the oracle (compare model_oracle1 and model_oracle2 results in the tables). How does this help us figure out what oracle should we use for a given model and dataset?

Fourth, I find some claims around distillation in the paper to not be quite accurate, e.g. lines like "A lot of KD techniques are specialized for use with Neural Networks". Distillation has been applied to not just neural nets, whether as teachers or students. TREPAN which the paper cites distills a neural net into a tree; so do Frosst and Hinton (https://arxiv.org/abs/1711.09784). Fakoor et al. (https://arxiv.org/pdf/2006.14284.pdf) distill arbitrary models into boosted trees, random forest, neural nets, etc. I think distillation is a baseline that should be in the comparative evaluation.

Fifth, a significant portion of the related work is relegated to the Appendix. The much shortened version of the related work in the main paper does not feel like it adequately explains how this paper is different from and borrows from prior work besides Ghose & Ravindran (2020).

Finally, the paper is 40 pages long not counting the Appendix, which makes it 65 pages long. The TMLR submission guidelines suggest that while submissions may be any length, a paper’s length should be justified by its content. I really recommend the authors take an editorial eye to the paper and prune it significantly to increase readability.

---

> ### Author Response · Authors · 2023-04-19
> **Response to reviewer's comments**
>
> Thank you for your feedback.
>
> We provide our responses below:
>
> 1. Novelty wrt the density trees paper Ghose and Ravindran (2020) ([https://www.frontiersin.org/articles/10.3389/frai.2020.00003/pdf](https://www.frontiersin.org/articles/10.3389/frai.2020.00003/pdf)).
>     1. Algorithm 2 in Ghose and Ravindran (2020) and this manuscript both show how to sample from a Dirichlet Process defined for a Beta Mixture Model. They are bound to be similar if not identical, because they both list the steps for a standard method of sampling using the Blackwell-MacQueen sampler, which in itself is neither novel for Ghose and Ravindran (2020) or this manuscript. This is a component within the techniques they contribute: in Ghose and Ravindran (2020) this is Algorithm 5 (which uses ideas from Algorithms 2-4), and here, this is Algorithm 3 (plus details in Sec 3.6).
>     2. Reg. eqns in Sec 2.3 here & those in Sec 1 in Ghose and Ravindran (2020): These represent problem statements. They're similar since both works solve the same problem, i.e., determine the optimal training distribution. The contributions are their solutions, Algorithm 5 (in the prior work) and Algorithm 3 (here).
>     3. We have also mentioned what exactly our extension is over Ghose and Ravindran (2020) in Sec 3.1 of this manuscript. Still, to summarize, the density trees technique creates (an ensemble of) specialized tree structures, a complex setup that we claim is not required. Probabilistic oracles may be used with better results. This also makes the technique usable over different feature spaces (Sec 4.3.1).
> 2. Reg. implicit claim that sizes across model families are comparable. We do not make this claim. For ex., we do not claim that DTs of depth one are equivalent LPMs with one non-zero coefficient. Two model families were picked to show the effect is general - we describe this goal at the beginning of Sec 4. Table 2 shows them together for conciseness. In our discussion of results (Sec 4.1.6), we explicitly call attention to the fact that these models have different representational power per unit size. Nowhere in our discussion of results do we mention a *user’s perception* of sizes across model families.
> Additionally, when we *do* want to study the role of capacity, we control for model family variance in a specially designed experiment in Sec 4.1.9.
> 3. Sensitivity due to oracles: the goal of the study was to show that the technique works in general since we do not assume a specific oracle in Eqns (2) and (3).
> That different oracles produce different improvements is not surprising because they possibly have different accuracies and probability calibrations to begin with. In practice, picking the oracle is like picking an optimizer for NNs, e.g., ADAM vs RMSprop; past experience, convenience, etc., may guide a choice.
> 4. Claims around distillation: We discuss the relationship to KD in three places, twice in the main paper and then again in the appendix:
>     1. In the main paper, in Sec 2.4. we mention *"*the primary difference is that we ignore alignment/fidelity between the predictions of the oracle and the interpretable model. In fact, the oracle’s label predictions are discarded."*. This is why the technique is is different from KD - if we ignore the teacher's labels, we cannot be said to perform distillation. There are other differences, such as multiple invocations in case of LIME, SHAP and TREPAN.
>     2. In Sec 6, “Future Work”, point #1 in the list, we mention why our technique is in-theory similar to learning sample weights, and that also sets it apart from KD since this view implies we may not need an oracle at all (if this could be feasibly implemented).
>     3. We again mention the relationship to KD in the Appendix, in Sec A.1, and here we point to the focus of KD research on neural networks (as one of three arguments listed here) to indicate that many KD techniques cannot be used for general models - however, the primary reason for differences with KD still remain. We apologize if the wording here made it seem like this is the *only* difference from KD - we’ll altogether eliminate this argument from a future manuscript.
> 5. Reg. “…much shortened version of the related work…”: We wanted to keep related work (Sec 2.4) concise so we can quickly move on to our contribution. Which is why here we point to other sections, such as Sec 3.1, that get into detailed differences. Some of these differences are better understood only after knowing how the density trees approach works - hence the use of a separate section.
> 6. Reg. length of the paper: since this is a new area (with only one other paper preceding us), the burden is on us to prove this technique reliably works. Note that *nearly 2/3rds of the paper consists of empirical results*: the theory is only from pg 1-13. We can move the Sec “Additional Applications” to the Appendix, if the reviewer recommends, but we believe the other details are relevant to the main paper.

---

### Review · Reviewer_7HcU · 2023-04-25

**Summary Of Contributions:**

In this study, the authors propose mitigation of the trade-off between interpretability and accuracy in machine learning models, specifically addressing the challenge of enhancing the accuracy of small, interpretable models by learning a training distribution. They propose a novel, model-agnostic algorithm for learning these training distributions, which is founded on optimizing parameters for an Infinite Beta Mixture Model based on a Dirichlet Process distribution with 7 parameters. To enable manageable sampling over a potentially large dataset space, the authors project the training data onto a single dimension: prediction uncertainty scores derived from a highly accurate pretrained (oracle) model. BO is used to optimize the parameters of the distribution.

Through empirical evaluation on multiple real-world datasets and an array of oracle and interpretable models, the authors observe substantial relative improvements in a variety of reasonable scores (the authors introduce many scores!). The proposed algorithm offers several notable advantages: it serves as a framework (multiple components that can be quite general) with 7 parameters for the flexible distribution which are constant throughout all instances of the framework, compatibility with models exhibiting non-differentiable training loss (e.g., Decision Trees), and the presence of reasonable defaults for the majority of parameters, making it user-friendly.

The authors also observe that the learned distribution does not necessarily match the test distribution. Furthermore, their method diminishes when the size of the model grows.

**Audience:**

Yes

**Claims And Evidence:**

Yes

**Requested Changes:**

Change A: Algorithm 1 is very abstract. What is \Phi? Can you give examples of such parameters? Furthermore "suggest" is vaguely defined. Do we know how it "inspects" the history to produce \Phi_t? I understand that you wanted to present a generic sketch of the algorithm, but it is confusing with so much generality at that stage. I believe concrete examples would be helpful so early in the paper.

Change B: I think the smoothening approach needs to be ablated and extensively compared with alternatives because to my understanding uniform sampling over the uncertainties would mean uniform sampling over the dataset. Of course, I may be wrong :).

Change C: A: Equation (1) seems to me to speak about the effect of "distillation" of the performance of the oracle model in curating the dataset. I would expect a discussion/ comparison with distillation.

Minor A: of well does <- of how well does

Minor B: In Figure 3, can you try log scale for the y-axis, or alternatively zoom in to the lower percentages of improvements? The overlap of many lines makes it hard to discern.


**Strengths And Weaknesses:**

Strength A: Distribution over uncertainties - that's interesting and potentially useful even for a wider range of tasks.

Strength B: The claims of the paper are thoroughly investigated empirically.

Weakness A: The paper is very difficult to read. There are many newly introduced scores in the paper which makes it hard to compare with existing approaches.

Weakness B: While the method is general, there are missing experiments with deep learning. For example, can your oracle be a neural network and the dataset be in computer vision? Can you try your framework on MNIST/ CIFAR-10? A discussion/ comparison with more recent interpretability frameworks, such as https://arxiv.org/abs/2202.00622, is missing.

Question A: What is the motivation to exclude the predictions of the oracle? I understand that since you are not using the predictions, you are different from distillation approaches, but why not include the predictions as well?

Question B: In Figure 2 it seems like the flattening makes the sampling roughly uniform across all uncertainties, which means that the sampling is uniform across the whole dataset (to my understanding). Isn't uniform sampling producing just the original dataset? Why do we expect to see gains from your method compared to a uniform sampling baseline?

---

> ### Author Response · Authors · 2023-04-27
> **Response to reviewer's comments**
>
>
> Thank you for your review and feedback! We provide our responses below:
> 1. Motivation to exclude labels of the oracle:
>     1. Primary reason: the intuition of this paper comes from the density trees paper which shows that information of the location of class boundaries, irrespective of class identities, is sufficient to learn an optimal distribution.We use the same central intuition (as mentioned in Section 3.1) and hence class identity is ignored.
>     2. We also want to minimize dependence on the oracle beyond using it as a tool for dimensionality reduction, to cheaply learn a distribution (further discussed in #2 of this list); a benefit of this is we don’t inherit many of the challenges of distillation, e.g., what causes good models to be not good teachers, etc.
>     3. This decreased dependence also allows us to frame the general approach as one of learning instance weights for future work (Section 6, #1), allowing us to get rid of the oracle entirely.
> 2. Reg. smoothing makes the sampling seem uniform, and experiments to show effectiveness: Smoothing spreads out the original uncertainty values, i.e, those provided by the oracle, in the range [0, 1]. However, the sampling distribution $\Phi$ is learned *over* these values (note the random variable for the Betas in line 7 of Algo. 2), and thus will not necessarily sample the original data uniformly.
> To provide an example, lets say we have only 3 training points with these uncertainty values: [0, 0.1, 0.15]. The smoothing step maps these respectively to new values such as  [0, 0.5, 1.0].  $\Phi$ is learned over these values. Let’s say it assigns sampling probabilities such as [0.3, 0.2, 0.5].  $\Phi$ effectively denotes: “points with uncertainty score=0 are to be sampled with probability 0.3, those with uncertainty score=0.5 are to be sampled with prob. of 0.2, and those with uncertainty score=1.0 get sampled with prob. of 0.5”. Note that:
>     1. We are still talking about sampling instances, but these instances are *identified* by their new uncertainty values. In a standard density function, we assign densities based on a points location $x \in \mathbb{R}^d$ (in general this makes the cost of representation proportional to at least the dimensionality $d$), but here we use its uncertainty value (a scalar) instead, as a cheap-to-compute proxy for the location (as a result, we only require an univariate density now).It is a lossy proxy but contains the information we need: proximity to a boundary.
>     2. This univariate density $\Phi$ is learned, implying the sampling probabilities need not be uniform.
>     3. An obvious question then is that if the uncertainty values are being merely used as identifiers (loosely speaking), why couldn’t we have just used the original values? This we explain in Section 3.6, but in short, while in theory this is fine, in practice, learning the distribution $\Phi$ over the original uncertainty scores can become hard based on their distribution. **Empirical evidence** that this smoothing step helps, appears in Section 5, Point 1 “Effect of Flattening”.
>   3. Reg. Algo. 2 being abstract: thank you for this feedback. We will add examples in a future version of the manuscript. Here are examples of the quantities mentioned:
>       1. $\Phi$ are the parameters of a distribution, e.g., for a Gaussian Mixture Model, this is the  number of components, means and the covariance matrix.
>       2. *suggest()* was meant to imply that, in general, the optimizer has access to all past evaluations. How this information is used depends on the optimizer, e.g., Gradient Descent uses only one past (most recent) evaluation while Bayesian Optimizers typically use all of it (for example, in a Gaussian Process (GP) based optimizer all of the history contributes to the form of the current GP).
> 4. Reg. comparisons to Deep Learning: this work exhaustively explores the case of tabular data, but we anecdotally show that this method might work in the case of DL too - this appears in Section 3.1, where the oracle is a Gated Recurrent Unit. We will make this scope clearer in our discussion of “Contributions” in Section 1.
> 5.  Reg. Eqns(1) look like a distillation set up:  we wanted to mention the oracle’s accuracy as an upper-bound (which we empirically validate in Section 5, #3 - see Fig 15 and Table 8) because if this bound doesn’t hold this would be cause for surprise esp. since we ignore critical information such as predicted labels, from the oracle. Thus, this upper bound was mentioned not to indicate that this is something we enforce, but that we intuitively expect to hold. We see that this distinction may not be clear - thank you for pointing out - we will make it clear in future versions.
> 6. We agree with the minor changes. For Fig. 3 we’ll add a zoomed version.
>
> Summary: We will incorporate changes A and C - discussed in items #3 and #5 here. We’ve provided an explanation in response to change B: #2 here. We will incorporate the minor changes A and B.

---

### Decision · Action_Editors · 2023-05-22

**Recommendation:** Reject

**Comment:**

The paper attempts to improve interpretability in ML models. In particular, the authors propose to adapt the training data for the capacity limited interpretable model using uncertainty estimates from the oracle. A lot of empirical study is carried out, but might not be comprehensive as pointed out by reviewers. Unfortunately majority of the reviews of the paper is leaning rejection as there are several significant concerns raised by the reviewers which remain unresolved. Reviewer 7HcU points out the difficulty in reading the paper due to the introduction of numerous scores and suggests including experiments with deep learning models to enhance the generalizability of the proposed method. Reviewer 8AEa raises concerns about the length of the paper and presentation, questions the interpretability focus, and emphasizes the sensitivity of results to the choice of oracle. Additionally, they highlight inaccuracies in the discussion of distillation and suggest shortening the paper. Possibly can be worth comparing to papers like Teaching with commentaries [Raghu et al ICLR 2021], Mentornet: Learning data-driven curriculum [Jiang et al ICML 2018], etc. Reviewer mM6u appreciates the clarity of Sections 1 through 3 but questions the artificial capping of dataset sizes even after the author response. Also the reviewer wants comparison to directly $M_{base}$. In summary, TMLR is not focused on novelty but definitely on clarity and comprehensiveness. As this is an early work in an emerging area which makes its clarity all the more important. Taking into account these concerns, it is clear that there are significant revisions required for this paper, which would warrant another round of reviews.

**Audience:**

Topic of the paper and some results are highly relevant to the broad TMLR community, but current presentation might limit the audience the paper appeals to.

**Claims And Evidence:**

At high level seems most of the claims seems backed by experiments, except for some discussion around related work which needs clarification. Unfortunately too many definitions and setups (e.g. scores) and length of the paper (40 pages) limits ability to check all the claims thoroughly.

---

> ### Author Response · Authors · 2023-06-05
> **Concerns with the review process**
>
> While we respect the decision of the reviewers, we would like to clarify some aspects mentioned in the final decision. Also, we are disappointed that despite detailed responses during the review process, in many cases there was no discussion (with the exception of Reviewer mM6u), leading to a non-constructive one-sided process.
>
> 1. Inaccuracies in the discussion of distillation: This is a strong claim, which is incorrect, as we have pointed out in our response to Reviewer 8AEa. We don’t use the labels from the oracle, and we cannot be said to distill a model if we don’t enforce alignment at this basic level  (point #4.1 in [1]). We’ve also mentioned that this work indicates that oracle may not be needed at all (point #4.2 in [1]). These are fundamental differences from KD. These arguments are also present in the paper. Reviewer 8AEa claims that we misunderstand KD based on a statement we make *in the Appendix* - and here too, we have clarified that this was unintended wording, which we’re willing to change (point #4.3 in [1]) and which does not affect the other arguments.
> 2. Lack of interpretability focus:  Reviewer 8AEa makes this claim based on a misunderstanding of our claims as well as results - that we compare performances *across* model families. We don’t, and this was clarified in our response (point #2 in [1]). In fact, in the paper we *explicitly call attention* to the fact that performances across model families are *not* similar (also mentioned in our response).
> 3. Although the AE doesn’t mention this in their final decision, it is concerning that Reviewer 8AEa also claims the paper is not novel since (a) an intermediate sampling step in our paper and prior paper look similar, and (b) the problem statements in our paper and prior paper look similar. Wrt (a), this is a standard sampling step (Blackwell-MacQueen) that’s not a contribution of either paper (it’s explanation of background theory - and why this was arbitrarily picked to compare instead of the actual contributions?), and wrt (b), we are confused as to why solving the same problem indicates derivative work. We have clarified this in our response (points #1.1 and #1.2 in [1]). Our larger concern is that this indicates a cursory review.
> 4. We are genuinely surprised that AE mentions size of the dataset at 10k points as a reason for rejection (Reviewer mM6u). Most interpretability research uses standard tabular datasets that are much smaller, e.g., heart (N=303), wine (N=178) , diabetes (N=768), mushroom(N=8124), dorothea (N=1950) - please see these recent papers on [linear models](https://link.springer.com/article/10.1007/s10994-015-5528-6), those [with interactions](https://www.tandfonline.com/doi/full/10.1080/10618600.2014.938812), and tree-based models ([A](https://ojs.aaai.org/index.php/AAAI/article/view/5711), [B](http://florent.avellaneda.free.fr/dl/AAAI20.pdf), [C](https://proceedings.mlr.press/v162/agarwal22b.html)).  We deliberately picked datasets that are significantly larger - 13 datasets with 10k instances.
> 5. Comparison to Deep Learning (Reviewer 7HcU) : this paper looks at tabular data, and we have mentioned this in our response. We have also mentioned that if this scope isn't clear, we would be happy to add this to the paper (point #4 in [2]).
> 6. Comparisons to $M_{base}$ (Reviewer mM6u): We have mentioned in our response (point #2 in [3]) that we present two sets of results - the second set uses the *same models as* $M_{base}$, and additionally, the model selection hyperparam. space and data available to the baselines match the models produced by our algorithm. This makes them suitable as control variables for statistical tests. We don’t shy away from reporting -ve results (see Table 3 in the paper). We can understand if the complaint were about an *extra* set of results; we are not sure why is there a lack indicated.
>
> More than the decision itself, it is extremely disheartening to see that despite our detailed responses, we have not been able to garner meaningful dialog, much less constructive feedback. A criticism like the length of the paper is easily addressed by relegating sections to the Appendix (**note**: the **theory part of the paper is 12 pages**, the rest is experiments, and many of the comments above pertain to these 12 pages), and it seems unfair that the other serious concerns that were already addressed by us, were not only left unacknowledged, but are also cited as reasons for rejection. It is easy to claim “this paper doesn’t have X” without any context, but a review process should be more than just that.
>
> [1] [Response to Reviewer 8AEa](https://openreview.net/forum?id=12QKTmXjZn&noteId=NMPl9sxCNn)
>
> [2] [Response to Reviewer 7HcU](https://openreview.net/forum?id=12QKTmXjZn&noteId=JLezdPTczR)
>
> [3] [Response to Reviewer mM6u](https://openreview.net/forum?id=12QKTmXjZn&noteId=Go9mr1Po26)